# UNLOCKING GUIDANCE FOR DISCRETE STATE-SPACE DIFFUSION AND FLOW MODELS

**Hunter Nisonoff**,[*] **Junhao Xiong**,[*] **Stephan Allenspach**,[*] **Jennifer Listgarten**
University of California, Berkeley
{hunter_nisonoff,junhao_xiong,stephall,jennl}@berkeley.edu

## ABSTRACT

Generative models on discrete state-spaces have a wide range of potential applications, particularly in the domain of natural sciences. In continuous state-spaces, controllable and flexible generation of samples with desired properties has been realized using *guidance* on diffusion and flow models. However, these guidance approaches are not readily amenable to discrete state-space models. Consequently, we introduce a general and principled method for applying guidance on such models. Our method depends on leveraging continuous-time Markov processes on discrete state-spaces, which unlocks computational tractability for sampling from a desired guided distribution. We demonstrate the utility of our approach, *Discrete Guidance*, on a range of applications including guided generation of small-molecules, DNA sequences and protein sequences.

## 1 DIFFUSION AND FLOW-BASED GENERATIVE MODELS FOR SCIENCE

Generative models based on diffusion (Sohl-Dickstein et al., 2015; Song et al., 2021; Ho et al., 2020), and more recently on flow matching (Liu et al., 2023; Lipman et al., 2023; Albergo & Vanden-Eijnden, 2023), have unlocked great potential not only in image applications (Rombach et al., 2022; Ma et al., 2024), but also increasingly in the sciences. For example, these models have been suggested for generating molecular conformations (Corso et al., 2023; Xu et al., 2022; Hoogeboom et al., 2022), protein backbone coordinates (Ingraham et al., 2023; Watson et al., 2023), and all-atom coordinates of small molecule-protein complexes (Krishna et al., 2024; Abramson et al., 2024; Qiao et al., 2024; Lu et al., 2024). In these scientific examples, the models are operating in real-valued state-spaces of 3D coordinates. However, in many scientific applications, not to mention natural language processing, the objects of interest lie in discrete state-spaces. In the sciences, discrete state-spaces emerge from biological sequences (DNA/RNA/protein) and molecular graphs (Wang et al., 2023), for example. Discrete state-spaces pose unique challenges for developing diffusion and flow-based models, which have been focused predominantly on real-valued spaces. In particular, continuous state-space diffusion processes rely at their core on gradients of probability densities with respect to variables in real-valued state spaces—the score function; consequently, when the state-space is discrete, one easily runs into problems. Flow matching, on the other hand, is typically implemented with a linear interpolation between a noise and target distribution, which has been shown to behave pathologically for discrete state-spaces, stimulating alternative approaches, such as one which operates on the continuous state-space of the probabilistic simplex (Stärk et al., 2024).

A number of discrete time, discrete state-space diffusion approaches have been proposed (Austin et al., 2021; Hoogeboom et al., 2021a; Vignac et al., 2023). On the other hand, Campbell and colleagues have developed frameworks of continuous-time diffusion and flow matching on discrete state-spaces by leveraging continuous-time Markov chains (CTMCs) (Campbell et al., 2022; 2024). In such formulations, one effectively learns a *denoising* neural network that approximates the instantaneous probability of transitioning between states—that is, a rate matrix—instead of a score or flow-generating vector field. However, one major ingredient missing from these approaches that operate on discrete state-spaces is the ability to construct conditional generative models by way of *guidance* in a principled way (Dhariwal & Nichol, 2021; Sohl-Dickstein et al., 2015; Ho & Salimans, 2021; Song et al., 2021). Indeed, arguably the most important ingredient enabling useful application

---

[*]Contributed equally: Author order is randomized and can be adjusted as needed for individual purposes.

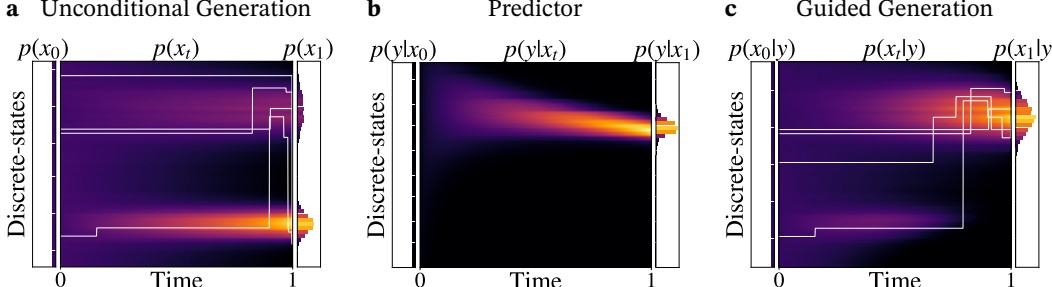

Figure 1: Illustration of guidance for continuous-time discrete state-space models. Color denotes the time-dependent density in each panel, with brighter as higher probability. **a**, A CTMC evolves samples (indicated by white lines) from an initial *noise* distribution, $p(x_0)$, through time, to generate samples from a desired target distribution, $p(x_1)$, by alternating between holding a state in time (horizontal lines), and jumping to new states (vertical lines). **b**, A predictor model trained to predict $y$ on time-dependent noisy samples of $x$. **c**, Discrete Guidance guides the unconditional model with the predictor to sample from $p(x_1|y)$.

of diffusion and flow models is that of conditioning the generative process on desired criteria. For example, in protein engineering, we may want to condition sequence generation on a backbone structure, binding affinity, or enzymatic activity (Hsu et al., 2024; Dauparas et al., 2022; Hsu et al., 2022; Li et al., 2023; Vanella et al., 2022). Conditioning of diffusion models is typically achieved by way of introducing guidance, either in a classifier-free way (Ho & Salimans, 2021), or by using a classifier (Dhariwal & Nichol, 2021). Classifier guidance, in particular, provides the key ability to leverage an already-trained unconditional model without re-training it, to generate conditional samples simply by modulating its generative process with guidance from a classifier (Du et al., 2023). Such a strategy is particularly important in the sciences where one frequently has orders of magnitude more unlabelled data than labeled data; one can therefore invest considerably in one unconditional model and then repurpose it for many different tasks (Ingraham et al., 2023).

Herein, for the first time, we introduce a principled approach, *Discrete Guidance*, to perform guidance for diffusion and flow models on discrete state-spaces (Figure 1). Discrete Guidance applies not only to continuous-time diffusion and flow models on discrete state-spaces, but also, in principle, to the broad class of generative models on discrete state-spaces realized through CTMCs. Our key insight is that, in *continuous time*, only a single dimension of the discrete state-space Markov chain can change at any point in time, making guidance *exact and tractable*. We empirically demonstrate the effectiveness of Discrete Guidance by applying it to a broad set of discrete state-space conditional generation tasks, including small-molecules, DNA sequences, and protein sequences. Because in the sciences, conditioning is often on real-valued properties rather than on satisfying a particular class, we refer to guidance more generally as *predictor guidance* and *predictor-free guidance*.

In summary, our contributions are:

1. We introduce an exact, theoretically rigorous, general framework, Discrete Guidance (DG), for applying both predictor guidance (PG) and predictor-free guidance (PFG) to generative models in discrete state-spaces based on CTMCs, such as diffusion and flow matching.
2. We develop an approximation to our guidance framework that leads to more efficient sampling while maintaining sample quality.
3. We empirically explore and demonstrate the potential utility of our guidance framework across a wide range of problem domains, using either diffusion or flow models.

## 2 CHALLENGES OF GUIDANCE IN DISCRETE STATE-SPACES

We first consider diffusion models. Roughly speaking, the generation of states, $x$ (*e.g.*, protein sequences), can be guided towards a desired property, $y$ (*e.g.*, binding affinity to a target), through application of Bayes' theorem so as to obtain the desired conditional. That is, the conceptual strategy is to take an unconditional density, $p(x)$, and a classifier- or regression-derived likelihood, $p(y|x)$, to obtain $p(x|y) = p(y|x)p(x)/p(y)$. However, because diffusion models on continuous

state-spaces learn the score, $\nabla_x \log p(x|y)$, the normalizing constant, $p(y)$, conveniently drops out as it does not depend on $x$. More specifically, in continuous state-space models, sampling $x \sim p(x|y)$ can be achieved by sampling from the reverse process at time $t$ with a conditional score function, $\nabla_{x_t} \log p_t(x_t|y)$, obtained by combining the unconditional score function, $\nabla_{x_t} \log p_t(x_t)$, and the score function of a predictor, $\nabla_{x_t} \log p_t(y|x_t)$ as follows, $\nabla_{x_t} \log p_t(x_t|y) = \nabla_{x_t} \log p_t(x_t) + \nabla_{x_t} \log p_t(y|x_t)$ (Sohl-Dickstein et al., 2015; Song et al., 2021; Dhariwal & Nichol, 2021). However, for discrete state-space models, these score functions are not defined, and such an approach cannot be readily adopted. While generalizations of the score function to discrete state-spaces have been proposed (Sun et al., 2023b; Meng et al., 2022; Lou et al., 2023), these developments do not address guidance. We sidestep the need for a score function by instead returning to Bayes' theorem, which, in the context of diffusion on discrete state-spaces, dictates that we must effectively compute

$$p(x_{t+dt}|x_t, y) = \frac{p(y|x_{t+dt}, x_t)p(x_{t+dt}|x_t)}{\sum\limits_{x'_{t+dt}} p(y|x'_{t+dt}, x_t)p(x'_{t+dt}|x_t)}, \tag{1}$$

where $x_{t+dt}$ and $x_t$ are separated by a infinitesimal time step $dt$, and $y$ is the desired property we wish to condition on. In a $D$-dimensional discrete state-space where each dimension has cardinality $S$, there are $S^D$ terms in the normalizing constant of Equation 1, rendering it generally intractable. Each term arises from the fact that, without constraints, any state can be reached from any other state. Intuitively, tractability can only be achieved if the number of terms is not exponential, thereby implying some constraints on the allowed state transitions. Later, we shall see how the continuous-time formulation of discrete state-space diffusion models enables us to tractably compute the normalizing constant by imposing such constraints without losing any model expressibility. As it turns out, the same underlying computations will also unlock flow-based guidance for us. To date, there has been limited work on incorporating guidance into flow models, even in continuous state-spaces (Dao et al., 2023; Zheng et al., 2023); although Stärk et al. (2024) perform flow matching on the probabilistic simplex to generate discrete samples, and explore guidance in the continuous state-space of the simplex.

## 3 BACKGROUND ON DISCRETE DIFFUSION AND FLOW MODELS

Continuous-time Markov Chains (CTMCs) are a class of continuous-time stochastic processes on discrete spaces (Norris, 1998). They have been used to build continuous-time discrete diffusion (CTDD) (Campbell et al., 2022), and discrete flow models (DFM) (Campbell et al., 2024). Intuitively, a CTMC can be described as a generative process wherein a state remains unchanged over some amount of *holding time*, which is exponentially distributed. Once the holding time is over, then the state must change (*jump*) to a new state, according to what is sometimes called the *jump probability* (Figure 1a). In practice, these two processes of holding, and jumping, are all jointly encoded in a (time-dependent) rate matrix, $R_t$, which describes the time derivative of the probability of transitioning from any state to any other state at time $t$. Consequently, the entries of this rate matrix are the key objects that determine both the frequencies and the destinations of the jumps. These rates, $R_t(x, \tilde{x})$, are related to the probability of an instantaneous transition occurring at time $t$ from a state $x$ to a state $\tilde{x}$ at time $t + dt$ through $p(x_{t+dt} = \tilde{x}|x_t = x) = \delta_{x,\tilde{x}} + R_t(x, \tilde{x})dt$. Transitions that change the state ($x \neq \tilde{x}$ so that $\delta_{x,\tilde{x}} = 0$) have positive rates (*i.e.*, $0 \leq R_t(x, \tilde{x})$ for $x \neq \tilde{x}$), while transitions from states to themselves ($x = \tilde{x}$ so that $\delta_{x,\tilde{x}} = 1$) have negative rates (*i.e.*, $R_t(x, x) \leq 0$).

Given these rates $R_t(x, \tilde{x})$ at every time point $t$, in addition to an initial distribution $p(x_{t=0})$, the generative process is fully defined and can be concretely used to simulate the corresponding CTMC through time, for example, by straightforward Euler integration (Campbell et al., 2024), but also with more sophisticated methods (Appendix B). Both the diffusion and flow-based models on discrete state-spaces mentioned above (CTDD and DFM) are generative models on discrete state-spaces that leverage CTMCs to define their generative processes.[1] Both can be thought of as at training time, progressively adding noise to the original data distribution (*e.g.*, by accumulating masked states), and then learning a denoising model to reverse that CTMC. Similarly to continuous state-space diffusion, the learned denoising model, $p^\theta(x_1|x_t)$, is trained to predict the initial (noise-free) data $x_1$ from noisy data $x_t$, where the specific loss function to do so depends on whether the model is diffusion- or flow-based. However, for discrete state-spaces, the corresponding denoising model induces a learned

---

[1]For all models, we follow the convention of flow-based models, where $t = 1$ is the target (training) distribution, and $t = 0$ is the noise distribution.

rate matrix (Appendix B), which can be used to generate samples. To do so, one starts with a sample from the known and easy-to-sample from noise distribution, then *moves* toward the desired target distribution by integrating over time with the time-dependent rate matrices. Similarly, for conditional generation of $x$ given a desired property $y$, one can in principle leverage conditional rate matrices $p(x_{t+dt} = \tilde{x} | x_t = x, y) = \delta_{x,\tilde{x}} + R_t(x, \tilde{x}|y)dt$ if one had access to these conditional rate matrices. One strategy to obtain these matrices would be to extract the conditional rates $R_t(x, \tilde{x}|y)$ directly from a learned conditional denoising model $p^\theta(x_1|x_t, y)$ (*e.g.*, using either CTDD (Campbell et al., 2022) or DFM (Campbell et al., 2024)). Such direct approaches to conditional modeling in diffusion models have typically been shown to be outperformed by guidance-based approaches (Dhariwal & Nichol, 2021; Ho & Salimans, 2021). Consequently, in the next section, we introduce Discrete Guidance to unlock guidance for these kinds of models.

## 4 UNLOCKING GUIDANCE IN DISCRETE STATE-SPACES

Both CTDD (Campbell et al., 2022) and DFM (Campbell et al., 2024) assume that the noising processes in each dimension are independent of one another (but not necessarily identical). Even in those models, without any attempt at guidance, this assumption has an important consequence: when the continuous-time noising process in each dimension is independent of the others, the probability that two or more dimensions of the current state jump (transition) at exactly the same time is zero (Campbell et al., 2022). Consequently, at any given time $t$, only $D \times (S - 1) + 1$ entries, $R_t(x, \cdot)$, in the rate matrix[2] are non-zero, making such models tractable. This assumption does not constrain model expressivity, because in any finite time interval, there is a non-zero probability to transition from one state to any other state.

It turns out that the same assumption which makes unguided CTDD and DFM tractable, also unlocks tractable guidance of these same models. Specifically, using the same reasoning, it follows that the sum in the denominator of Equation 1 is only over $D \times (S - 1) + 1$ terms instead of $S^D$. This occurs because we need only consider terms $p(x_{t+dt} = x'|x_t = x)$ computed from non-zero rates $R_t(x, x')$.

Here, we briefly present our main results and refer to Appendix C for their detailed derivation and to Appendix D for their minimal implementation in PyTorch. As both CTDD and DFM are formulated in terms of rates that are linked to the transition probabilities, we present our results in terms of rates. Consequently, we obtain Discrete Guidance by reformulating Bayes' theorem in Equation 1 in terms of rates. In doing so, we find, in analogy to predictor guidance in continuous state-space (Sohl-Dickstein et al., 2015; Song et al., 2021; Dhariwal & Nichol, 2021), an expression for the conditional rates, $R_t(x, \tilde{x}|y)$, in terms of the unconditional rates, $R_t(x, \tilde{x})$, and a (guiding) predictive distribution, $p(y|x, t)$, as

$$R_t(x, \tilde{x}|y) = \frac{p(y|\tilde{x}, t)}{p(y|x, t)} R_t(x, \tilde{x}), \tag{2}$$

for any state $\tilde{x} \neq x$. When $\tilde{x} = x$, owing to conservation of probability flow for any rate matrix, we have $R_t(x, x|y) = -\sum_{x' \neq x} R_t(x, x'|y)$. Note that for CTMCs defined by discrete state-space diffusion and flow models, tractability of the sum in the denominator of Bayes' theorem in Equation 1 is tantamount to tractability of the sum in $R_t(x, x|y)$, which is analogous to the *normalizing constant* of a probability distribution. The tractability arises because only $D \times (S - 1) + 1$ of the rates are non-zero (Campbell et al., 2022; 2024). As for the intuition behind Equation 2, we see that the conditional rates can be obtained by modulating the unconditional rates with a likelihood ratio describing how much the guidance signal increases (or decreases) in an infinitesimal time step.

As in guidance in continuous state-space (Song et al., 2021; Dhariwal & Nichol, 2021), one can tune the impact of guidance by raising the likelihood ratio term in Equation 2 by the guidance strength, $\gamma \in \mathbb{R}^+$, to obtain the *predictor-guided* (PG) rates $R_t^{(\gamma)}(x, \tilde{x}|y) = \left[\frac{p(y|\tilde{x},t)}{p(y|x,t)}\right]^\gamma R_t(x, \tilde{x})$ for $x \neq \tilde{x}$.

In analogy to predictor-free guidance in continuous state-spaces (Ho & Salimans, 2021), we too can derive predictor-free guidance, wherein one seeks to blend an unconditional model with a conditional model of $x$ given $y$. In analogy to predictor-free guidance in continuous state-spaces (Ho & Salimans, 2021), we find (Appendix C) that the predictor-guided rates can alternatively be obtained in terms of

---

[2]$(S - 1)$ transitions in each of the $D$ dimensions and a single identity transition.

conditional, $R_t(x, \tilde{x}|y)$, and unconditional, $R_t(x, \tilde{x})$, rates for $x \neq \tilde{x}$ in the form

$$R_t^{(\gamma)}(x, \tilde{x}|y) = R_t(x, \tilde{x}|y)^\gamma R_t(x, \tilde{x})^{1-\gamma}. \tag{3}$$

The rates in Equation 3 do not depend on any predictive distribution and we therefore call them *predictor-free-guided* (PFG) rates. In practice, predictor-free-guidance is achieved by learning both a conditional, $R_t^\varphi(x, \tilde{x}|y)$, and unconditional, $R_t^\theta(x, \tilde{x})$, rate matrix and combining them as described in Equation 3. Note that the guided rates, $R_t^{(\gamma)}(x, \tilde{x}|y)$, are a generalization of both the conditional $[R_t^{(\gamma=1)}(x, \tilde{x}|y) = R_t(x, \tilde{x}|y)]$ and unconditional (*i.e.*, fully unguided) $[R_t^{(\gamma=0)}(x, \tilde{x}|y) = R_t(x, \tilde{x})]$ rates. Prior work found that $\gamma > 1$ often produces higher-quality samples that satisfy the desired conditioning criterion (Dhariwal & Nichol, 2021).

Finally, we note that the rate adjustments just presented apply not only to continuous-time diffusion and flow models on discrete state-spaces, but also, in principle, to a broad class of generative models on discrete state-spaces realized through CTMCs (Appendix C.9).

### 4.1 EFFICIENT APPROXIMATIONS FOR PREDICTOR GUIDANCE

Although we just saw how to overcome the combinatorial complexity of computing the normalizing constant in Bayes' theorem for guidance in discrete state-spaces, computing the adjusted rates in Equations 2 may still be practically onerous in some cases. In particular, to generate a sample requires $D \times (S-1) + 1$ function calls of the predictor model, $p(y|x, t)$, for each step in the reverse sampling process. This is tractable for moderately-sized $D$ and $S$, but may be prohibitively expensive in some applications. To enable more efficient sampling for such cases, we draw insights from Grathwohl et al. (2021), where it was noted that even when trained on discrete $x$, a neural network model, $p^\phi(y|x, t)$, is a continuous function on $x \in \mathbb{R}^{D \times S}$, enabling a first-order Taylor series approximation. Thus the guidance log-likelihood ratio required in Equation 2 can be approximated as

$$\log \frac{p^\phi(y|\tilde{x}, t)}{p^\phi(y|x, t)} = \log p^\phi(y|\tilde{x}, t) - \log p^\phi(y|x, t) \approx (\tilde{x} - x)^T \nabla_x \log p^\phi(y|x, t), \tag{4}$$

requiring only one forward and one backward pass of the predictor model with the input $x$, instead of $D \times (S-1) + 1$ forward passes; this makes estimation of the guide-adjusted rates $\mathcal{O}(1)$ rather than $\mathcal{O}(D \times S)$. We refer to this as *Taylor-approximated guidance* (TAG). Note that Equation 4 is only ever evaluated at discrete values of $\tilde{x}$ and $x$. Although it is not obvious that such a gradient-based approximation can provide an accurate estimate of the true likelihood ratio, this idea builds on a large set of literature demonstrating empirical success in applying Taylor series approximations to neural networks on discrete data (Zhang et al., 2022; Kirjner et al., 2023; Sun et al., 2022; 2023a). Additionally, our own results substantiate these findings further—we systematically compared TAG to exact guidance in all our empirical investigations, finding that in most cases, TAG matches the sample quality of exact guidance (Section 6). Finally, note that although TAG bears superficial resemblance to DiGress (Vignac et al., 2023), the methods are quite distinct (Section 5), with correspondingly large difference in empirical results, generally showing TAG to be superior (Section 6).

## 5 RELATED WORK

Discrete Guidance builds on both the discrete flow model (DFM) (Campbell et al., 2024) and the continuous-time discrete diffusion model (CTDD) (Campbell et al., 2022), both of which operate on discrete spaces with continuous time. Those works extend prior work on diffusion in discrete spaces with discrete time (Hoogeboom et al., 2021b; Austin et al., 2021). Other approaches for continuous-time diffusion on discrete spaces also build on a CTMC formulation but employ alternative objectives to the ELBO (Sun et al., 2023b; Lou et al., 2023). More recent work generalizes the flow-matching framework in Campbell et al. (2024) to a more general family of probability paths (Gat et al., 2024). None of the aforementioned work propose a method to perform guidance; however, as each method defines the sampling process through a CTMC, each can be used naturally with Discrete Guidance.

In continuous state-spaces, Sohl-Dickstein et al. (2015) illustrate how an unconditional discrete-time diffusion model can be modified for conditioning. Song et al. (2021) and Dhariwal & Nichol (2021) further build on this result for controllable generation in a score-based continuous-time, continuous state-space framework by using a classifier, while Ho & Salimans (2021) shows how a similar result

can be achieved without the need for a classifier; instead, they blend score estimates of a conditional diffusion model with a jointly trained unconditional diffusion model.

Several methods sidestep the issue of performing guidance in discrete space by instead performing it on a learned, continuous embedding of the discrete state-space, thereby leveraging standard guidance for real-valued spaces (Gruver et al., 2024; Li et al., 2022). Such approaches require additional training and hyperparameter selection to obtain the required embedding. In contrast, Discrete Guidance enables easier re-purposing of already well-developed predictor models that work on the native, discrete input space (*e.g.*, see Section 6.3).

Stärk et al. (2024) introduce Dirichlet FM, enabling generation of samples in discrete spaces by flow matching on the continuous state-space of the probability simplex; they can perform both predictor-free guidance and predictor-guidance. In contrast, our approach directly operates on the native discrete state-space. Empirically, we find our predictor-guidance approach to yield better results than Dirichlet FM on the same task presented by Stärk et al. (2024) (Section 6.2).

The DiGress method of Vignac et al. (2023) performs approximate guidance in a discrete-time, discrete state-space diffusion framework. To do so, similarly to (yet distinctive of) our TAG approach, they use a Taylor series approximation of the predictor model operating on the discrete state-space. Although our TAG approximation bears resemblance to DiGress, DiGress is fundamentally anchored in discrete-time, which has been shown to be limiting in theory and in practice (Campbell et al., 2022; 2024; Lou et al., 2023; Gat et al., 2024), and further substantiated in our experimental results. In particular, because DiGress is based on a discrete-time formulation, the normalizing constant required for conditioning (Equation 1) is intractable, therefore requiring additional approximations beyond our TAG approximation (Appendix E). Moreover, with Discrete Guidance, we can perform tractable *exact* guidance, which at times provides a benefit over TAG, whereas DiGress can only perform approximate guidance. Finally, DiGress is specific to diffusion and not readily generalizable to flow matching. Further discussion of related work can be found in Appendix E.

## 6 EMPIRICAL INVESTIGATIONS

We deployed Discrete Guidance on three conditional generation tasks spanning problems on small molecules, DNA sequences and protein sequences. On the whole, we found unconditional flow matching training to be more stable than unconditional diffusion, so we trained our own unconditional flow-matching models to be used with guidance in all three experiments, following the training procedure in Campbell et al. (2024). To demonstrate that Discrete Guidance also readily applies to continuous-time discrete diffusion models, we also guided a pre-trained discrete diffusion model to produce class-conditional images (Appendix F.1). When predictor-guidance is used, a time-dependent predictor must be trained separately (Appendix C.5). The task-specific training procedures for these models are detailed in Appendix F.2-F.4. Note that in our flow matching experiments we used a stochasticity hyperparameter at sampling time, which Campbell et al. (2024) showed could improve sample quality.

Our goals were to investigate if Discrete Guidance worked as expected and to investigate potential benefits offered by our approach. For each task, we compared to the discrete-time, discrete diffusion guidance approach DiGress (Vignac et al., 2023) to illustrate the empirical advantage of Discrete Guidance. In all experiments, DiGress used the same architecture and hyperparameters as Discrete Guidance for both the unconditional and predictor models. For the DNA and protein tasks, we compared to additional task-specific baselines that could yield additional insights into the utility of our approach. Our intent in this section is not to claim optimal sample generation on any particular problem, rather, to demonstrate that Discrete Guidance can be successfully applied to multiple domains. Further details of all experiments can be found in Appendix F.1-F.4.

### 6.1 SMALL-MOLECULES GENERATION

First, we used Discrete Guidance with an unconditional generative model of small molecules for each of two properties: the number of rings ($N_r$) and the lipophilicity (LogP). The values of both chemical properties were obtained using RDKit (Landrum, 2010), where LogP is approximated using the Wildman-Crippen approach (Wildman & Crippen, 1999). Our dataset, consisting of 610,575 unique molecules with a weight below 750 Da, up to 7 number of rings, and lipophilicity in [-3,

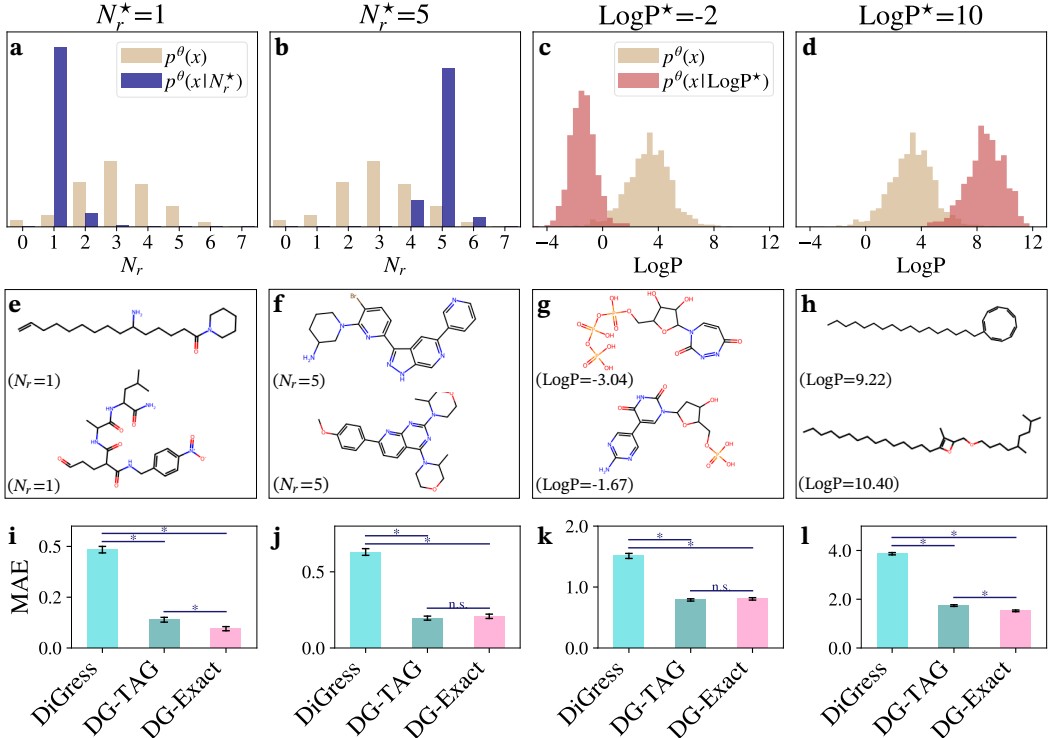

Figure 2: Predictor guidance for small molecule generation conditioned on number of rings ($N_r$) or lipophilicity (LogP). **a-d**, Property-histograms of 1,000 generated valid SMILES strings (*i.e.*, molecules) sampled from the unconditional model, $p^\theta(x)$, and by Discrete Guidance with predictor-guidance, $p^\theta(x|y^\star)$, where $y^\star = N_r^\star$, or LogP$^\star$. Guidance strength was $\gamma = 2$ and stochasticity was $\eta = 30$. **e-h**, First two molecules sampled from the predictor-guided model indicated above (in **a-d**), **i-l**, Mean absolute error (MAE) between specified target properties indicated respectively in panels **a-d**, and those of the 1,000 generated valid SMILES strings by each of three methods: DiGress (Vignac et al., 2023), Discrete Guidance with both exact (DG-Exact) and Taylor-approximated guidance (DG-TAG). Error bars correspond to standard errors. An asterisk ($\ast$) denotes statistical significance difference in MAE between the two models under the start and end point of the lines below it (two-sided Mann-Whitney U test, $p < 0.05$), while n.s. denotes lack of statistical significance.

10], is based on QMugs (Isert et al., 2021) and has been constructed as described in Appendix F.2.1. Using simplified molecular-input line-entry system (SMILES) (Weininger, 1988) strings as a discrete state-space representation of the molecules, we trained an unconditional masking flow model for discrete state spaces (Campbell et al., 2024), then guided it either with a number-of-rings model, or a lipophilicity model. Model architectures and training procedures are detailed in Appendix F.2.2.

We generated SMILES strings from both the unconditional model, $p^\theta(x)$, and also from the conditional model, $p^\theta(x|y^\star)$, obtained by guiding the unconditional model with a predictor model to the target property value $y^\star$ (Figure 2a-d). As generated SMILES strings are not guaranteed to be *valid* (*i.e.*, corresponding to molecules), we sampled until 1,000 valid SMILES strings were generated, using these as molecules to be assessed. We set the stochasticity hyperparameter of the unconditional model to $\eta = 30$, as this value maximized the validity of SMILES strings generation. Increasing the guidance strength, $\gamma$, led to a decrease in SMILES string validity and we selected $\gamma = 2$ for our experiments as the largest $\gamma$ resulting in a reasonable validity. Further details concerning generation, SMILES string validity, and choice of stochasticity can be found in Appendix F.2.3 and F.2.4.

As expected, Discrete Guidance shifted the histograms of sampled molecule property values compared to the unguided model, for conditioning on each of number of rings ($N_r^\star = 1, 5$ Figure 2a,b), and specified target lipophilicity (LogP$^\star = -2, 10$, Figure 2c,d). Inspecting the first two molecules generated with Discrete Guidance, we find molecules with $N_r$ matching the specified value $N_r^\star$ (Figure 2e,f) and molecules with LogP close to the specified LogP$^\star$ (Figure 2g,h). Further results on

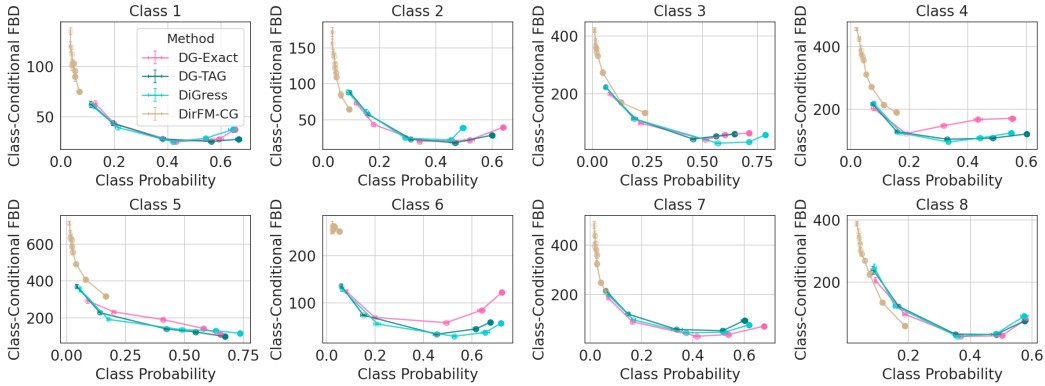

Figure 3: Cell type-conditioned DNA enhancer sequence generation for eight cell type classes. The better the model, the lower the *Class-Conditional FBD*, and the higher the average sample *Class Probability*. Thus, roughly speaking, better conditional generative models will appear in the lower, right-hand portion of each panel. Each point in each panel arises from 1,000 sampled sequences for one guidance strength. Mean and standard deviation of both metrics were estimated over five bootstrap replicates (the standard deviations are small and are not readily discernible). For each method, guidance strengths increase from the left to the right in each panel.

a wider range of target property values, discussion of chemical bonds and their relation to lipophilicity, and an assessment of diversity can be found in Appendix F.2.5.

Finally, we compared Discrete Guidance—using both exact and the Taylor-approximated guidance (TAG)—to DiGress, an approach for guided diffusion (Vignac et al., 2023). To compare methods, we used the mean absolute error (MAE) of the properties of molecules generated with both methods with the same guidance strength, $\gamma$, to each of specified property values (shown for $\gamma = 2$ in Figure 2i-l and for additional $N_r^\star$, LogP$^\star$, and $\gamma$ in Appendix F.2.6). Discrete Guidance was statistically significantly better able to generate molecules with properties closer to their specified values than was DiGress. Exact guidance with Discrete Guidance led to comparable or better results than using TAG.

## 6.2 ENHANCER DNA DESIGN

Following the experimental setup of Stärk et al. (2024), we next instantiated Discrete Guidance on a class-conditional DNA sequence generation task centered on enhancers. Enhancers are non-coding DNA sequences that regulate gene expression, often in a cell-specific manner. We used the same enhancer sequence dataset as Stärk et al. (2024), comprising 104k DNA sequences of length 500, each with one of 81 possible cell type class labels (Janssens et al., 2022; Taskiran et al., 2024). We compared our method to the Dirichlet FM (DirFM) conditional models of Stärk et al. (2024), namely both the classifier-free (DirFM-CFG) and classifier-guided (DirFM-CG) models, which use flow-matching in the (continuous) space of the probability simplex. For Discrete Guidance, we trained an unconditional flow-matching model, then used predictor guidance with either exact (DG-Exact) or Taylor-approximated guidance (DG-TAG). We also trained a conditional flow-matching model to be used with predictor-free guidance (DG-PFG). As predictor-free guidance for DirFM-CFG and DG-PFG behaved comparably (Appendix F.3.5), we focused here on predictor guidance. In addition, we compared to DiGress (Vignac et al., 2023). Further details are in Appendix F.3.2.

We used the same two evaluation metrics as Stärk et al. (2024), which depend on an *oracle* classifier model that gives the probability of each cell type class given a sequence; it was trained on only un-noised data. One evaluation metric was the Fréchet Biological Distance (FBD), used to compare generated samples to the training data, for each specified class—lower FBD is better (Stärk et al., 2024). The other metric was the probability of the desired target class under this same oracle classifier model—higher probability is better. We note that the FBD is a heuristic metric for measuring the distributional similarity between the generated samples and the training data, and that the target class probability is the metric which better reflects the effectiveness of guidance.

We evaluated the methods on a total of eight different cell type classes, including the three cell type classes chosen for evaluation in Stärk et al. (2024). These cell type classes were randomly

chosen among the classes where the oracle classifier had relatively good performance (Appendix F.3.4). We computed both metrics from 1,000 generated sequences conditioned on each of the eight target classes. For both DG-Exact and DG-TAG, sampling stochasticity, $\eta$, was set to 0 as this led to the best unconditional FBD. For each method, we varied the guidance strengths of each model to trace out how the two metrics changed (Figure 3). Originally, for DirFM-CG we used the same guidance strengths $\gamma \in \{1, 3, 6, 10, 20\}$ reported by Stärk et al. (2024). For Discrete Guidance and DiGress, we used similar guidance strengths $\gamma \in \{1, 2, 5, 10, 20\}$. However, we noticed that with the original guidance strengths, DirFM-CG was unable to match the performance of other methods, and so expanded the range of guidance strengths for DirFM-CG, which in some cases improved its results. Details of the sampling procedure can be found in Appendix F.3.3.

Across all cell type classes, DG-Exact, DG-TAG and DiGress perform comparably, while DirFM-CG performed more poorly by comparison (Figure 3). We hypothesize that DirFM-CG may suffer from the need to project the classifier score function onto the tangent plane of the simplex to perform guidance within the simplex. Additional results can be found in Appendix F.3.4.

## 6.3 GUIDING INVERSE-FOLDING MODELS FOR STABILITY

Inverse-folding models are conditional generative models of protein sequence, conditioned on a protein backbone structure. That is, one provides a structure as input, and can then sample different protein sequences that are likely to fold into that structure. These models play a key role in machine learning-driven protein engineering (Dauparas et al., 2022; Hsu et al., 2022; Sumida et al., 2024); consequently, any improvements we can make to them would be broadly valuable to the protein engineering community. Full experimental details are in Appendix F.4, with an overview here.

Herein, we explore whether guiding such conditional generative models with a protein stability predictive model might prove useful. In particular, our goal is to generate protein sequences that are just as likely to fold into the correct structure as a standard inverse folding model, while having improved predicted stability. To provide predictor guidance for Discrete Guidance, we trained a stability-predicting regression model from a large dataset of protein folding stability measurements on mini-proteins (Tsuboyama et al., 2023). Stability is reported in real-valued units of $\Delta\Delta G$, which reflects the difference in stability from the wild-type sequence. Herein, we define a stable protein as one that is at least as stable as the wild-type (*i.e.*, $\Delta\Delta G \geq 0$), although one might consider using other thresholds. Altogether then, we start with a standard (not stability-conditioned) inverse folding model, $p(\mathbf{x} \mid \mathbf{c})$, where $\mathbf{x}$ corresponds to a protein sequence and $\mathbf{c}$ is the structure condition. Then we apply Discrete Guidance to obtain the stability-conditioned model, $p(\mathbf{x} \mid \mathbf{c}, \Delta\Delta G > 0)$. Concretely, we first trained a flow-matching inverse folding model using data in Dauparas et al. (2022), which we refer as a Flow-Matching Inverse Folding (FMIF) model. We also trained a regression model, $p(\Delta\Delta G \mid \mathbf{x}, \mathbf{c})$, to use for guidance—this model showed good generalization on a holdout set (Appendix F.4).

We applied this set-up on eight randomly chosen proteins from the dataset provided by Tsuboyama et al. (2023). Here we show the summary of the results for all eight proteins, while detailed results for each individual protein can be found in Appendix F.4.6. We compared FMIF to both exact (FMIF-DG-Exact) and Taylor-approximated (FMIF-DG-TAG) predictor guidance, as well as to DiGress. We also included as a baseline, ProteinMPNN (Dauparas et al., 2022), a very widely-used approach for inverse folding, which does not explicitly model stability (Sumida et al., 2024). A stochasticity level, $\eta$, of 1 was used for sampling with the flow-matching models. For ProteinMPNN and FMIF we sampled from the models with structure-conditioned temperature values $1.0, 0.1$ and $0.01$. When guidance was applied, we evaluated with guidance strengths $\gamma \in \{1, 10, 100\}$ and used a structure-conditioned temperature value of $0.1$, as we observed that this temperature had much better RMSD values for the non-guided models (*i.e.*, ProteinMPNN and FMIF) than a temperature of $1.0$.

For each of the methods we sampled 100 sequences. The success criterion had two components: (1) the generated sequences should fold into the desired structure, $\mathbf{c}$, and (2) the generated sequences should be at least as stable as the wild-type sequence, $\Delta\Delta G \geq 0$. To evaluate how well a generated sequence folds onto a desired structure, we used AlphaFold 2 (Jumper et al., 2021) to predict the structure of the sequence and compare the RMSD of the predicted structure to the desired structure $\mathbf{c}$. Following previous evaluation criteria for inverse folding models, we consider a sequence to achieve the desired fold if the RMSD is less than or equal to 2Å (Campbell et al., 2024). To evaluate the folding stability of our sequences, we use our $\Delta\Delta G$ regression model.

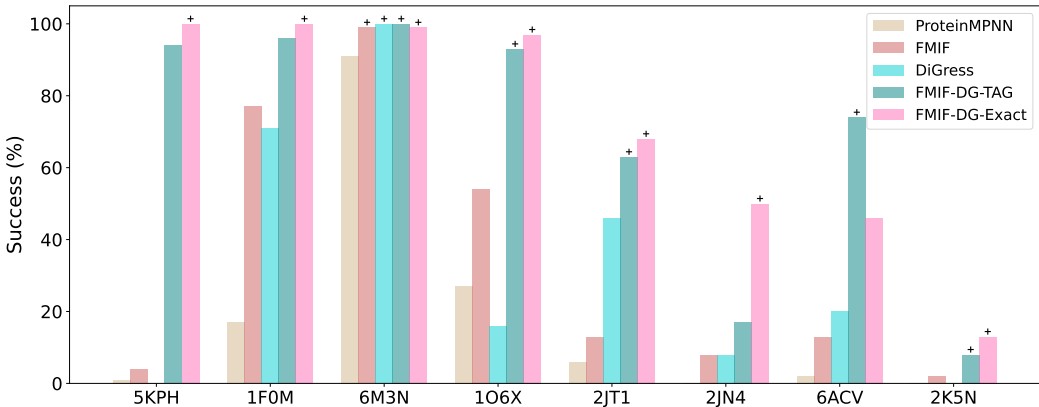

Figure 4: Stability-guided inverse-folding for eight randomly chosen proteins. Five generative models (in legend) were assessed by the success rate of their 100 generated samples, where success was defined as sequences that had correct predicted structure (RMSD $\leq 2$Å), and were stable ($\Delta\Delta G \geq 0$). For non-guided models (ProteinMPNN and FMIF) success was defined over the maximum of the three structure-conditioned temperature values, while for the guided methods (all others) success was defined over the maximum of the three guidance strength values. Top-performing methods are indicated by a plus sign (+), which denotes that each such model lacks a statistically significant difference from the one method with highest success rate (one-sided t-test, $p < 0.05$).

We observed that across all methods, FMIF-DG-Exact and FMIF-DG-TAG consistently had the highest success rates (Figure 4), with FMIF-DG-Exact outperforming or matching all other methods on seven out of eight proteins. There was one protein, 6ACV, for which FMIF-DG-TAG actually performed better than FMIF-DG-Exact. We hypothesize that this is a result of the time-dependant predictor model used in guidance being miscalibrated with respect to the original oracle predictor. In such instances, exact guidance may actually result in worse sample quality compared to approximate methods since the guidance can be led astray by the miscalibrated predictor. Interestingly among the two non-guided models, FMIF consistently matched or outperformed ProteinMPNN, suggesting possible benefits to using flow-matching models for inverse folding over order-agnostic autoregressive models. Altogether, these results suggest that stability-guided inverse folding with Discrete Guidance is a promising direction for updating protein generative models with experimental stability data.

## 7 DISCUSSION

We introduced and empirically explored a principled and general approach to perform guidance on discrete state-spaces. Our approach, Discrete Guidance, is applicable to a broad class of generative models on discrete state-spaces realized through CTMCs, including continuous-time diffusion and flow models. We evaluated our approach empirically by applying it to guided conditional generation tasks in multiple domains, including small-molecules, DNA sequences and protein sequences.

While our work demonstrated the effectiveness of Discrete Guidance in a variety of applications, it also leaves some avenues for further investigations and improvements. Although we have shown that Taylor-approximated guidance works well empirically, it lacks the theoretical guarantees offered by exact guidance, and further investigation into the potential trade-off between efficiency and accuracy would be of interest to the community. Also, guidance requires training predictors on noised samples, and it is unclear what training strategies would lead to the most effective guided generation (Klarner et al., 2024). Finally, it would also be interesting and potentially fruitful to explore Discrete Guidance for controllable text generation of language models.

Our work illustrates that guided conditional generation in discrete state-spaces has the potential to be leveraged across the natural sciences. We expect that future work will more fully realize the potential of guided generation in these and other domains.

## ETHICS STATEMENT

Our work focuses on improving conditional generative modeling. There are many potential positive impacts of this work. Conditional generative modeling is increasingly being used in drug discovery and protein engineering creating positive impacts for human health. However, there is also a risk that such models could be used for malicious purposes like bioweapons. They also can amplify biases present in the training data that can lead to unfair treatment of certain groups.

## REPRODUCIBILITY STATEMENT

We have included detailed derivations of the results we presented in Appendix C. Appendix D provides minimal PyTorch implementations of Discrete Guidance and source code is available at https://github.com/hnisonoff/discrete_guidance. Detailed descriptions of each of the experiments are included in Appendix F.

## ACKNOWLEDGEMENTS

We thank H. Jiang, J. Bowden, M. Giammar, K. Atz, and H. Stärk for helpful discussions. This work was funded in part by DTRA under award HDTRA1036045 and supported by the Office of Naval Research (ONR) under grant N00014-23-1-2587. Additional support was provided by the U.S. Department of Energy, Office of Science, Office of Biological and Environmental Research, Lawrence Livermore National Laboratory BioSecure SFA within the Secure Biosystems Design program (SCW1710). S. A. was supported by the Swiss National Science Foundation (grant no. P500PT_214430).

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

# APPENDIX

## A ORGANIZATION OF APPENDIX

The Appendix is organized as follows. Appendix B provides additional backgrounds on CTMCs and the models based on CTMCs. Appendix C contains detailed derivations of the main results. Appendix D provides minimal PyTorch implementations of Discrete Guidance, as well as the derivations for an extension of DFM for states with fixed components. Appendix E contains further discussion of related work, including a detailed comparison with other approaches for guidance. Appendix F provides further details and results for the experiments.

## B ADDITIONAL BACKGROUND

In this section, we provide more detailed backgrounds on Continuous-time Markov Chains (CTMCs), as well as continuous-time discrete diffusion models (CTDD) and discrete flow models (DFM), which are based on CTMCs.

First, recall that the rates, $R_t(x, \tilde{x})$, are related to the probability of an instantaneous transition occurring at time $t$ from a state $x$ to a state $\tilde{x}$ at time $t + dt$ through:

$$p_{t+dt|t}(x_{t+dt} = \tilde{x}|x_t = x) = \delta_{x,\tilde{x}} + R_t(x, \tilde{x})dt.$$

This can be derived from the Kolmogorov forward equation:

$$\partial_t p_t(x_t = x) = \sum_{\tilde{x} \neq x} R_t(\tilde{x}, x)p_t(x_t = \tilde{x}) - \sum_{\tilde{x} \neq x} R_t(x, \tilde{x})p_t(x_t = x).$$

Intuitively, the Kolmogorov forward equation states that the difference between the incoming and outgoing probability mass of any given state is the time derivative of the marginal (Campbell et al., 2024), ensuring that the total flow is conserved globally. Also, note that for generative modeling, the CTMCs are not homogeneous, because we specifically want the process to correspond to regimes with more or less noise, hence the $t$ subscript on the rate matrix.

In CTDD (Campbell et al., 2022), the user specifies time-dependent rate matrices $\bar{R}_t(x_t, \cdot)$ (referred to as the *forward process* for diffusion models) that govern the noise process evolving from $t = 1$ to $t = 0$, where $t = 0$ is pure noise. The model is trained on an ELBO loss, which is a continuous-time analogy of the discrete-time ELBO derived in Austin et al. (2021). Minimizing this ELBO can be intuitively thought of as minimizing the KL-divergence between the forward process posterior, and learned reverse process, as the time step size goes to zero. By minimizing the ELBO, we learn a denoising model, $p_{1|t}^\theta(x_1|x_t)$, which in turn induces a backward-in-time rate matrix, $R_t^\theta(x_t, \tilde{x}_t)$. In the notation of flow-based models, this learned backward-in-time rate matrix can be rewritten as

$$R_t^\theta(x_t, \tilde{x}_t) = \bar{R}_t(\tilde{x}_t, x_t) \sum_{x_1} \frac{p_{t|1}(\tilde{x}_t|x_1)}{p_{t|1}(x_t|x_1)} p_{1|t}^\theta(x_1|x_t).$$

In DFM (Campbell et al., 2024), rather than specifying a forward rate matrix such as in CTDD, the user specifies a data-conditional flow $p_{t|1}(\cdot|x_1)$, which induces a family of data-conditional rate matrices, $\bar{R}_t(x_t, \cdot|x_1)$, that each generate the relevant conditional flow (by satisfying the Kolmogorov equations). Commonly used conditional flows include those that linearly interpolate towards $x_1$ from a uniform prior or fully-masked state $M$ (Campbell et al., 2024), and can also be generalized to other types of probability paths (Gat et al., 2024). We primarily used the masking noising process introduced in Campbell et al. (2024) in our experiments: $p_{t|1}^{\text{mask}}(x_t|x_1) = \text{Cat}(t\delta\{x_1, x_t\} + (1 - t)\delta\{M, x_t\})$.

Campbell et al. (2024) show that the rates used in generative sampling can be obtained by training a neural network with parameters $\theta$, $p_{1|t}^\theta(x_1|x_t)$, to approximate the true denoising distribution using the standard cross-entropy loss:

$$\mathcal{L}_{ce} = \mathbb{E}_{p_{\text{data}}(x_1)\mathcal{U}(t;0,1)p_{t|1}(x_t|x_1)} \left[ \log p_{1|t}^\theta(x_1|x_t) \right] \tag{5}$$

where $\mathcal{U}(t; 0, 1)$ is a uniform distribution on $[0, 1]$. $x_t$ can be sampled from $p_{t|1}(x_t|x_1)$. As shown in Campbell et al. (2024), the cross entropy loss can be understood as a simplification to the ELBO used to train CTDDs. Given such a trained denoising model $p_{1|t}^\theta(x_1|x_t)$, the unconditional rate matrix used in sampling is defined as:

$$R_t^\theta(x_t, \cdot) = \mathbb{E}_{p_{1|t}^\theta(x_1|x_t)}\Big[\bar{R}_t(x_t, \cdot|x_1)\Big] \tag{6}$$

where $\bar{R}_t(x_t, \cdot|x_1)$ is induced by $p_{t|1}(x_t|x_1)$. DFM generalizes CTDD by enabling more flexible choices of $\bar{R}_t(x_t, \cdot|x_1)$ at sampling-time, since there is a family of data-conditional rate matrices $\bar{R}_t(x_t, \cdot|x_1)$ that all generate $p_{t|1}(x_t|x_1)$ by maintaining detailed balance (Campbell et al., 2024).

There are many methods for sampling from CTMCs given the rates and an initial distribution. One simple approach is to use $R_t^\theta(x_t, \cdot)$ to estimate the transition probability $p_{t+\Delta t|t}^\theta(x_{t+\Delta t}|x_t)$ via a finite Euler step of step size $\Delta t$. Alternatively, the Gillespie's Algorithm (Gillespie, 1977) can simulate a time-homogeneous CTMC exactly by alternating between sampling a holding time and sampling the next state, where both the holding time and the state transition probabilities depend on the rates. The Gillespie's Algorithm can be adjusted for a time-inhomogeneous CTMC with the modified next reaction method, which involves numerical integration of the time-dependent rate matrices (Anderson, 2007). Finally, $\tau$-leaping (Gillespie, 2001; Campbell et al., 2022) is an approximate simulation method to allow transitions in multiple dimensions to be applied simultaneously, which recovers the exact simulation in the limit as $\tau \to 0$.

## C GUIDANCE IN CTMCS

### C.1 UNCONDITIONAL GENERATION WITH CTMCS

In this subsection, we provide an overview of the work of Refs. (Campbell et al., 2022; 2024) that formulate discrete state-space diffusion and flow models as CTMCs. We denote $x_t$ as the random-variable describing the state of the system at time $t$ that can take any state-value $x$ in the state-space $\mathcal{X}$. As described in Section 3, the dynamics of a CTMC is specified by an initial distribution $p(x_0)$ at time $t = 0$ and the rates $R_t(x_t, x_{t+\Delta t})$ that are related to the infinitesimal transition probabilities $p(x_{t+\Delta t}|x_t)$ through

$$\begin{aligned}
p(x_{t+\Delta t}{=}\tilde{x}|x_t{=}x) &= \delta_{x,\tilde{x}} + R_t(x, \tilde{x})\Delta t + \mathcal{O}(\Delta t^{1+\epsilon}) \\
&= \delta_{x,\tilde{x}} + \delta_{x,\tilde{x}}R_t(x, \tilde{x})\Delta t + (1 - \delta_{x,\tilde{x}})R_t(x, \tilde{x})\Delta t + \mathcal{O}(\Delta t^{1+\epsilon}) \\
&= \delta_{x,\tilde{x}}\big[1 + R_t(x, x)\Delta t\big] + (1 - \delta_{x,\tilde{x}})R_t(x, \tilde{x})\Delta t + \mathcal{O}(\Delta t^{1+\epsilon}),
\end{aligned} \tag{7}$$

for a transition from a state $x_t$ (with value $x$) to a state $x_{t+\Delta t}$ (with value $\tilde{x}$) at time $t$, where we define $R_t(x, \tilde{x}) \equiv R_t(x_t{=}x, x_{t+\Delta t}{=}\tilde{x})$. In comparison to the exposition in the main paper, here we explicitly write these transition probabilities as an expansion in $\Delta t$ following Campbell et al. (2022), while implicitly assuming $\Delta t \to 0$ for Equation 7 and in this entire section. We use $\mathcal{O}(\Delta t^{1+\epsilon})$ to denote all terms that tend to zero faster than $\Delta t$ (i.e., with a positive $\epsilon \ll 1$).

In the following, we assume that the rates, $R_t(x, \tilde{x})$ are known; for example they could be induced by a learned probability distribution $p_{1|t}^\theta(x_1|x_t)$ with parameters $\theta$ as discussed in Appendix B. Using these rates, a known initial distribution $p(x_0)$ can be propagated in time to $p(x_t)$ for any $t \geq 0$ using Equation 7 along with some integrator in time, such as those described above.

As $p(x_{t+\Delta t}|x_t)$ is a probability distribution, it must fulfill, for all $x_{t+\Delta t}$ and $x_t$,
(i) $0 \leq p(x_{t+\Delta t}|x_t)$ and (ii) $\sum_{x_{t+\Delta t}} p(x_{t+\Delta t}|x_t) = 1$. From (i) with Equation 7 it follows that $0 \leq R_t(x, \tilde{x})$ for all $x' \neq x$, and from (ii) with Equation 7, it follows that

$$\sum_{\tilde{x}} R_t(x, \tilde{x}) = 0, \tag{8}$$

which can also be written in the alternative form

$$R_t(x, x) = -\sum_{\tilde{x} \neq x} R_t(x, \tilde{x}). \tag{9}$$

Note that $R_t(x, x) \leq 0$ because $0 \leq R_t(x, \tilde{x})$ for all $\tilde{x} \neq x$. Thus, if $R_t(x, \tilde{x})$ is known for all $\tilde{x} \neq x$, then one can derive $R_t(x, x)$ from these values.

## C.2 CONDITIONAL CTMCS

The unconditional CTMC framework can be adapted for conditional generation given a property $y$ by learning $p^\varphi_{1|t}(x_1|x_t, y)$ that induces the conditional rates $R_t(x, \tilde{x}|y)$ as in the unconditional scenario discussed in Appendix B. These conditional rates then specify the infinitesimal transition probabilities in the conditional form of Equation 7:

$$
\begin{aligned}
p(x_{t+\Delta t}\!=\!\tilde{x}|x_t\!=\!x, y) &= \delta_{x,\tilde{x}} + R_t(x, \tilde{x}|y)\Delta t + \mathcal{O}(\Delta t^{1+\epsilon}) \\
&= \delta_{x,\tilde{x}}\big[1 + R_t(x, x|y)\Delta t\big] + (1 - \delta_{x,\tilde{x}})R_t(x, \tilde{x}|y)\Delta t + \mathcal{O}(\Delta t^{1+\epsilon})
\end{aligned}
\tag{10}
$$

As in the unconditional case, the rates must fulfill $0 \le R_t(x, \tilde{x}|y)$ for all $\tilde{x} \ne x$ and $\sum_{\tilde{x}} R_t(x, \tilde{x}|y) = 0$ or equivalently $R_t(x, x|y) = -\sum_{\tilde{x}} R_t(x, \tilde{x}|y)$.

After learning the conditional rates during training from data $p_{\text{data}}(x, y)$, one can sample $x|y$ from $p^\varphi(x_1|y)$ by generating samples from a (conditional) noise-distribution $p(x_0|y)$ and propagating them to $t = 1$ using Equation 10.

Conditional generation in this form requires either $y$-labeled data during training, or fine-tuning of an unconditional model (trained on an unlabeled set) by transfer-learning on another (potentially smaller) labeled set. Note that, as in the continuous state-space case, conditional training does lack the modularity offered by guidance (Du et al., 2023).

## C.3 PREDICTOR GUIDANCE IN CTMCS

We will show now that instead of obtaining the conditional rates $R_t(x, \tilde{x}|y)$ from a directly-learned conditional model, $p^\varphi_{1|t}(x_1|x_t, y)$, one can instead leverage Bayes' theorem to construct them by combining the unconditional rates $R_t(x, \tilde{x})$ with a $y$-predictive distribution, $p(y|x_{t+\Delta t}\!=\!\tilde{x}, x_t\!=\!x)$. In particular, we can rewrite the left-hand side of Equation 10 as

$$
\begin{aligned}
p(x_{t+\Delta t}\!=\!\tilde{x}|x_t\!=\!x, y) &= \frac{p(y|x_{t+\Delta t}\!=\!\tilde{x}, x_t\!=\!x)p(x_{t+\Delta t}\!=\!\tilde{x}|x_t\!=\!x)}{p(y|x_t\!=\!x)} \\
&= \frac{p(y|x_{t+\Delta t}\!=\!\tilde{x}, x_t\!=\!x)p(x_{t+\Delta t}\!=\!\tilde{x}|x_t\!=\!x)}{\sum_{x'} p(y|x_{t+\Delta t}\!=\!x', x_t\!=\!x)p(x_{t+\Delta t}\!=\!x'|x_t\!=\!x)} \\
&= \frac{q(\tilde{x})p(x_{t+\Delta t}\!=\!\tilde{x}|x_t\!=\!x)}{\sum_{x'} q(x')p(x_{t+\Delta t}\!=\!x'|x_t\!=\!x)}
\end{aligned}
\tag{11}
$$

Where we have defined $q(x^\star) \equiv p(y|x_{t+\Delta t}\!=\!x^\star, x_t\!=\!x)$ in the last step. Using the expression of $p(x_{t+\Delta t}\!=\!\tilde{x}|x_t\!=\!x)$ in Equation 7 we can rewrite Equation 11 as

$$
\begin{aligned}
p(x_{t+\Delta t}\!=\!\tilde{x}|x_t\!=\!x, y) &= \frac{q(\tilde{x})p(x_{t+\Delta t}\!=\!\tilde{x}|x_t\!=\!x)}{\sum_{x'} q(x')p(x_{t+\Delta t}\!=\!x'|x_t\!=\!x)} \\
&= \frac{q(\tilde{x})\Big\{\delta_{x,\tilde{x}}\big[1+R_t(x, x)\Delta t\big] + \big[1-\delta_{x,\tilde{x}}\big]R_t(x, \tilde{x})\Delta t + \mathcal{O}(\Delta t^{1+\epsilon})\Big\}}{\sum_{x'} q(x')\Big\{\delta_{x,x'}\big[1+R_t(x, x)\Delta t\big] + \big[1-\delta_{x,x'}\big]R_t(x, x')\Delta t + \mathcal{O}(\Delta t^{1+\epsilon})\Big\}} \\
&= \frac{\delta_{x,\tilde{x}}q(x)\big[1+R_t(x, x)\Delta t\big] + \big[1-\delta_{x,\tilde{x}}\big]q(\tilde{x})R_t(x, \tilde{x})\Delta t + \mathcal{O}(\Delta t^{1+\epsilon})}{q(x)\big[1+R_t(x, x)\Delta t\big] + \sum_{x'\ne x} q(x')R_t(x, x')\Delta t + \mathcal{O}(\Delta t^{1+\epsilon})} \\
&= \frac{\delta_{x,\tilde{x}}\frac{q(x)}{q(x)}\big[1+R_t(x, x)\Delta t\big] + \big[1-\delta_{x,\tilde{x}}\big]\frac{q(\tilde{x})}{q(x)}R_t(x, \tilde{x})\Delta t + \mathcal{O}(\Delta t^{1+\epsilon})}{\frac{q(x)}{q(x)}\big[1+R_t(x, x)\Delta t\big] + \sum_{x'\ne x} \frac{q(x')}{q(x)}R_t(x, x')\Delta t + \mathcal{O}(\Delta t^{1+\epsilon})} \\
&= \frac{\delta_{x,\tilde{x}}\big[1+R_t(x, x)\Delta t\big] + \big[1-\delta_{x,\tilde{x}}\big]R^{(y)}_t(x, \tilde{x})\Delta t + \mathcal{O}(\Delta t^{1+\epsilon})}{1+R_t(x, x)\Delta t + \sum_{x'\ne x} R^{(y)}_t(x, x')\Delta t + \mathcal{O}(\Delta t^{1+\epsilon})}
\end{aligned}
\tag{12}
$$

where we have defined

$$R_t^{(y)}(x, x^*) \equiv \frac{q(x^*)}{q(x)} R_t(x, x^*) = \frac{p(y|x_{t+\Delta t}=x^*, x_t=x)}{p(y|x_{t+\Delta t}=x, x_t=x)} R_t(x, x^*) \tag{13}$$

for any $x^* \neq x$ in the last step.

Next, we use the Taylor expansion around $v = 0$ of $1/(1+v) = 1 - v + \mathcal{O}(v^2)$ to transform the denominator in Equation 12 to

$$\frac{1}{1 + \left\{ R_t(x,x)\Delta t + \sum_{x' \neq x} R_t^{(y)}(x,x')\Delta t + \mathcal{O}(\Delta t^{1+\epsilon}) \right\}}$$

$$= 1 - \left\{ R_t(x,x)\Delta t + \sum_{x' \neq x} R_t^{(y)}(x,x')\Delta t + \mathcal{O}(\Delta t^{1+\epsilon}) \right\} + \mathcal{O}\left( \left\{ \dots \right\}^2 \right)$$

$$= 1 - \left\{ R_t(x,x)\Delta t + \sum_{x' \neq x} R_t^{(y)}(x,x')\Delta t + \mathcal{O}(\Delta t^{1+\epsilon}) \right\} + \mathcal{O}(\Delta t^2)$$

$$= 1 - R_t(x,x)\Delta t - \sum_{x' \neq x} R_t^{(y)}(x,x')\Delta t - \mathcal{O}(\Delta t^{1+\epsilon}) + \mathcal{O}(\Delta t^2)$$

$$= 1 - R_t(x,x)\Delta t - \sum_{x' \neq x} R_t^{(y)}(x,x')\Delta t + \mathcal{O}(\Delta t^{1+\epsilon})$$

and insert this back into Equation 12

$$p(x_{t+\Delta t}=\tilde{x}|x_t=x, y) = \frac{\delta_{x,\tilde{x}}\left[1+R_t(x,x)\Delta t\right] + \left[1-\delta_{x,\tilde{x}}\right]R_t^{(y)}(x,\tilde{x})\Delta t + \mathcal{O}(\Delta t^{1+\epsilon})}{1+R_t(x,x)\Delta t + \sum_{x' \neq x} R_t^{(y)}(x,x')\Delta t + \mathcal{O}(\Delta t^{1+\epsilon})}$$

$$= \left\{ \delta_{x,\tilde{x}}\left[1+R_t(x,x)\Delta t\right] + \left[1-\delta_{x,\tilde{x}}\right]R_t^{(y)}(x,\tilde{x})\Delta t + \mathcal{O}(\Delta t^{1+\epsilon}) \right\}$$

$$\times \left\{ 1 - R_t(x,x)\Delta t - \sum_{x' \neq x} R_t^{(y)}(x,x')\Delta t - \mathcal{O}(\Delta t^{1+\epsilon}) \right\}$$

$$= \delta_{x,\tilde{x}}\left[ 1 + \cancel{R_t(x,x)\Delta t} - \cancel{R_t(x,x)\Delta t} - \sum_{x' \neq x} R_t^{(y)}(x,x')\Delta t \right]$$

$$+ \left[1-\delta_{x,\tilde{x}}\right]R_t^{(y)}(x,\tilde{x})\Delta t + \mathcal{O}(\Delta t^{1+\epsilon})$$

$$= \delta_{x,\tilde{x}}\left[ 1 - \sum_{x' \neq x} R_t^{(y)}(x,x')\Delta t \right] + \left[1-\delta_{x,\tilde{x}}\right]R_t^{(y)}(x,\tilde{x})\Delta t + \mathcal{O}(\Delta t^{1+\epsilon}). \tag{14}$$

As the left-hand sides of Equation 10 and Equation 14 are the same, we can compare terms in $(1 - \delta_{x,\tilde{x}})\Delta t$ to deduce

$$R_t(x, \tilde{x}|y) = R_t^{(y)}(x, \tilde{x}) \equiv \frac{p(y|x_{t+\Delta t}=\tilde{x}, x_t=x)}{p(y|x_{t+\Delta t}=x, x_t=x)} R_t(x, \tilde{x}), \tag{15}$$

where, in the second step, we have inserted the definition of $R_t^{(y)}(x, \tilde{x})$ (Equation 13). We find $0 \leq R_t(x, \tilde{x}|y)$ for all $\tilde{x} \neq x$, as required for rates, because $0 \leq R_t(x, \tilde{x})$ for all $\tilde{x} \neq x$ and $0 \leq p(y|x_{t+\Delta t} = x^*, x_t = x)$ for any $y$, $x$, $x^*$, because $p(y|x_{t+\Delta t} = x^*, x_t = x)$ is a probability distribution.

Comparing the terms in $\delta_{x,\tilde{x}}\Delta t$ in Equation 10 and Equation 14, we deduce

$$R_t(x, x|y) = - \sum_{x' \neq x} R_t^{(y)}(x, x') = - \sum_{x' \neq x} R_t(x, x'|y) \tag{16}$$

where, in the second step, we have used Equation 15. Note that Equation 16 is analogous to Equation 9 and is equivalent to $\sum_{x'} R_t(x, x'|y) = 0$ (in analogy to Equation 8) as required for rates.

Equation 15 illustrates that conditionality can be introduced to an unconditional CTMC with rates $R_t(x, x')$ using the predictive distribution $p(y|x_{t+\Delta t} = x', x_t = x)$. As a predictive distribution is used here to adjust the rates, this adjustment can be considered a realization of *predictor guidance* for CTMCs in analogy to continuous state-space diffusion (Sohl-Dickstein et al., 2015; Dhariwal & Nichol, 2021).

The sum in Equation 16 is theoretically over the entire (exponentially large) state space (that is $S^D - 1$ many states). However, in CTMCs defined by discrete state-space diffusion and flow models only $D \times (S - 1) + 1$ of the unconditional rates $R_t(x, \tilde{x})$ (Campbell et al., 2022; 2024) [and thus according to Equation 15 of the guide-adjusted rates $R_t^{(y)}(x, \tilde{x})$] are non-zero. As a consequence, the sum in Equation 16 must only be computed over $D \times (S - 1)$ terms in practice.

## C.4 SIMPLIFYING PREDICTOR GUIDANCE IN CTMCS

The predictor-guide-adjusted rates defined in Equation 15 involve the ratio of predictive distributions of the form $p(y|x_{t+\Delta t} = x^*, x_t = x)$. We will show now how these distributions can be transformed to a simpler form thereby simplifying the form of the predictor-guide-adjusted rates. In analogy to predictor guidance for continuous state-space diffusion in Dhariwal & Nichol (2021), we use Bayes' theorem to find

$$p(y|x_{t+\Delta t} = x^*, x_t = x) = \frac{p(x_t = x|x_{t+\Delta t} = x^*, y)p(y|x_{t+\Delta t} = x^*)}{p(x_t = x|x_{t+\Delta t} = x^*)}. \tag{17}$$

In predictor guidance, the noising process ($x_{t+\Delta t} \to x_t$) is independent of the guide property $y$ so that $x_t \perp\!\!\!\perp y|x_{t+\Delta t}$ and therefore $p(x_t = x|x_{t+\Delta t} = x^*, y) = p(x_t = x|x_{t+\Delta t} = x^*)$. Inserting this into Equation 17 we find

$$\begin{aligned} p(y|x_{t+\Delta t} = x^*, x_t = x) &= \frac{p(x_t = x|x_{t+\Delta t} = x^*, y)p(y|x_{t+\Delta t} = x^*)}{p(x_t = x|x_{t+\Delta t} = x^*)} \\ &= \frac{\cancel{p(x_t = x|x_{t+\Delta t} = x^*)}p(y|x_{t+\Delta t} = x^*)}{\cancel{p(x_t = x|x_{t+\Delta t} = x^*)}} \\ &= p(y|x_{t+\Delta t} = x^*) \end{aligned} \tag{18}$$

and Equation 15 can be simplified to

$$R_t(x, \tilde{x}|y) = \frac{p(y|x_{t+\Delta t} = \tilde{x}, x_t = x)}{p(y|x_{t+\Delta t} = x, x_t = x)} R_t(x, \tilde{x}) = \frac{p(y|x_{t+\Delta t} = \tilde{x})}{p(y|x_{t+\Delta t} = x)} R_t(x, \tilde{x}) = \frac{p(y|\tilde{x}, t)}{p(y|x, t)} R_t(x, \tilde{x}). \tag{19}$$

where, in the last step, we express $p(y|x_{t+\Delta t} = x^*)$ (with an implicit time-dependence) as $p(y|x^*, t)$ (with an explicit time dependence) using $t$ instead of $t + \Delta t$ in the limit of $\Delta t \to 0$. The intuition behind Equation 19 is that conditioning is achieved by moving to states $\tilde{x}$ that have higher predictive likelihood for the desired guidance property ($y$) than does the originating state, $x$, as quantified by the likelihood ratio raised to the power of the guidance strength.

## C.5 TRAINING A NOISY PREDICTOR MODEL

In practice, and following Song et al. (2021); Dhariwal & Nichol (2021), one can parameterize the *noisy predictor* model $p(y|x^*, t)$ with a learnable distribution $p^\phi(y|x^*, t)$. The parameters of this distribution, $\phi$, are then obtained by maximizing

$$\mathcal{L}(\phi) = \mathbb{E}_{\substack{(x_1, y) \sim p_{\text{data}}(x_1, y) \\ t \sim \mathcal{U}(0, 1) \\ x^* \sim p(x_t|x_1)}} \left[ \log p^\phi(y|x^*, t) \right]$$

using a labeled data distribution $p_{\text{data}}(x_1, y)$, a uniform distribution $\mathcal{U}(0, 1)$ and the unconditional noising (*i.e.*, forward) process $p(x_t|x_1)$. So in words; one samples a state $x_1$ together with its label $y$ from the data distribution, samples a time $t$, samples a noised state $x^*$ at time $t$ based on $x_1$, and evaluates the log-probability for $y$, $x^*$, and $t$ at the current model parameters $\phi$. There are also alternative strategies for approximating $p(y|x^*, t)$ that does not involve training a noisy predictor model as discussed in Appendix E.2.

## C.6 TEMPERATURE SAMPLING

Instead of sampling from

$$
\begin{aligned}
p(x_{t+\Delta t}\!=\!\tilde{x}|x_t\!=\!x,y) &= \delta_{x,\tilde{x}} + R_t(x,\tilde{x}|y)\Delta t + \mathcal{O}(\Delta t^{1+\epsilon}) \\
&= \delta_{x,\tilde{x}} + \frac{p(y|\tilde{x},t)}{p(y|x,t)} R_t(x,\tilde{x})\Delta t + \mathcal{O}(\Delta t^{1+\epsilon})
\end{aligned}
$$

(using Equation 19 to obtain the second from the first line) one can introduce a *guidance temperature* $T$, or equivalent an inverse guidance temperature $\gamma = 1/T$, which we call the *guidance strength*, and sample from

$$
\begin{aligned}
p^{(\gamma)}(x_{t+\Delta t}\!=\!\tilde{x}|x_t\!=\!x,y) &= \delta_{x,\tilde{x}} + \left[\frac{p(y|\tilde{x},t)}{p(y|x,t)}\right]^{\gamma} R_t(x,\tilde{x})\Delta t + \mathcal{O}(\Delta t^{1+\epsilon}) \\
&= \delta_{x,\tilde{x}} + R_t^{(\gamma)}(x,\tilde{x}|y)\Delta t + \mathcal{O}(\Delta t^{1+\epsilon})
\end{aligned} \tag{20}
$$

where we have defined

$$
R_t^{(\gamma)}(x,\tilde{x}|y) \equiv \left[\frac{p(y|\tilde{x},t)}{p(y|x,t)}\right]^{\gamma} R_t(x,\tilde{x}). \tag{21}
$$

for $x \neq \tilde{x}$.

This procedure is known as temperature sampling (Ackley, David H. and Hinton, Geoffrey E. and Sejnowski, Terrence J., 1985) and is for example used in continuous state-space diffusion models (Dhariwal & Nichol, 2021; Ho & Salimans, 2021). Lowering the temperature enables one to tune the generative process from fully unconditional ($R_t^{(\gamma=0)}(x,\tilde{x}|y) = R_t(x,\tilde{x})$) to conditional ($R_t^{(\gamma=1)}(x,\tilde{x}|y) = R_t(x,\tilde{x}|y)$). One can even go *beyond* the exact conditioning, making the guidance signal stronger still ($1 < \gamma$ or equivalently $T < 1$). We note that effectively sampling from the true temperature-annealed distribution of diffusion models remains an open problem (Ingraham et al., 2023; Du et al., 2023).

## C.7 PREDICTOR-FREE GUIDANCE IN CTMCS

We will show now that Equation 20 can also be reformulated in terms of rates $R_t(x,\tilde{x})$ of an unconditional CTMC and $R_t(x,\tilde{x}|y)$ of a conditional CTMC. This step is in direct analogy to predictor-free guidance in continuous state-space diffusion (Ho & Salimans, 2021) that has been derived as an alternative to predictor-guidance in continuous state-space diffusion (Sohl-Dickstein et al., 2015; Dhariwal & Nichol, 2021). This result will, as in the continuous state-space case, allow us to apply the controllability of predictor guidance but without the dependence on any predictor, and therefore corresponds to a realization of *predictor-free guidance* for CTMCs.

Using Equation 17, Equation 18, and Bayes' theorem, we rewrite $R_t^{(\gamma)}(x,\tilde{x}|y)$ defined in Equation 21 for $\tilde{x} \neq x$ as

$$
\begin{aligned}
R_t^{(\gamma)}(x,\tilde{x}|y) &= \left[\frac{p(y|\tilde{x},t)}{p(y|x,t)}\right]^{\gamma} R_t(x,\tilde{x}) = \left[\frac{p(y|x_{t+\Delta t}\!=\!\tilde{x},x_t\!=\!x)}{p(y|x_{t+\Delta t}\!=\!x,x_t\!=\!x)}\right]^{\gamma} R_t(x,\tilde{x}) \\
&= \left[\frac{p(x_{t+\Delta t}\!=\!\tilde{x}|x_t\!=\!x,y)\,\cancel{p(y|x_t\!=\!x)}\,p(x_{t+\Delta t}\!=\!x|x_t\!=\!x)}{p(x_{t+\Delta t}\!=\!x|x_t\!=\!x,y)\,\cancel{p(y|x_t\!=\!x)}\,p(x_{t+\Delta t}\!=\!\tilde{x}|x_t\!=\!x)}\right]^{\gamma} R_t(x,\tilde{x}) \\
&= \left[\frac{p(x_{t+\Delta t}\!=\!\tilde{x}|x_t\!=\!x,y)\,p(x_{t+\Delta t}\!=\!x|x_t\!=\!x)}{p(x_{t+\Delta t}\!=\!x|x_t\!=\!x,y)\,p(x_{t+\Delta t}\!=\!\tilde{x}|x_t\!=\!x)}\right]^{\gamma} R_t(x,\tilde{x}).
\end{aligned} \tag{22}
$$

Using expressions for the relationships between transition probabilities and rate matrices (unconditional in Equation 7 and conditional in Equation 10), we can further expand Equation 22 as follows

(assuming $\tilde{x} \neq x$, which will be used to transform the second to the third line),

$$
\begin{aligned}
R_t^{(\gamma)}(x,\tilde{x}|y) &= \left[ \frac{p(x_{t+\Delta t}=\tilde{x}|x_t=x,y)p(x_{t+\Delta t}=x|x_t=x)}{p(x_{t+\Delta t}=x|x_t=x,y)p(x_{t+\Delta t}=\tilde{x}|x_t=x)} \right]^{\gamma} R_t(x,\tilde{x}) \\
&= \left[ \frac{\left\{ \delta_{x,\tilde{x}} + R_t(x,\tilde{x}|y)\Delta t + \mathcal{O}(\Delta t^{1+\epsilon}) \right\} \times \left\{ \delta_{x,x} + R_t(x,x)\Delta t + \mathcal{O}(\Delta t^{1+\epsilon}) \right\}}{\left\{ \delta_{x,x} + R_t(x,x|y)\Delta t + \mathcal{O}(\Delta t^{1+\epsilon}) \right\} \times \left\{ \delta_{x,\tilde{x}} + R_t(x,\tilde{x})\Delta t + \mathcal{O}(\Delta t^{1+\epsilon}) \right\}} \right]^{\gamma} R_t(x,\tilde{x}) \\
&= \left[ \frac{\left\{ 0 + R_t(x,\tilde{x}|y)\Delta t + \mathcal{O}(\Delta t^{1+\epsilon}) \right\} \times \left\{ 1 + R_t(x,x)\Delta t + \mathcal{O}(\Delta t^{1+\epsilon}) \right\}}{\left\{ 1 + R_t(x,x|y)\Delta t + \mathcal{O}(\Delta t^{1+\epsilon}) \right\} \times \left\{ 0 + R_t(x,\tilde{x})\Delta t + \mathcal{O}(\Delta t^{1+\epsilon}) \right\}} \right]^{\gamma} R_t(x,\tilde{x}) \\
&= \left[ \frac{R_t(x,\tilde{x}|y)\Delta t + \mathcal{O}(\Delta t^{1+\epsilon})}{R_t(x,\tilde{x})\Delta t + \mathcal{O}(\Delta t^{1+\epsilon})} \right]^{\gamma} R_t(x,\tilde{x}) \\
&= \left[ \frac{R_t(x,\tilde{x}|y)}{R_t(x,\tilde{x})} \right]^{\gamma} \left[ \frac{1 + \mathcal{O}(\Delta t^{\epsilon})}{1 + \mathcal{O}(\Delta t^{\epsilon})} \right]^{\gamma} R_t(x,\tilde{x}).
\end{aligned}
$$

Then we again use the Taylor expansion around $v = 0$ of $1/(1+v) = 1 - v + \mathcal{O}(v^2)$ for $\frac{1}{1+\mathcal{O}(\Delta t^{\epsilon})}$ to simplify

$$
\left[ \frac{1 + \mathcal{O}(\Delta t^{\epsilon})}{1 + \mathcal{O}(\Delta t^{\epsilon})} \right]^{\gamma} = \left[ \left\{ 1 + \mathcal{O}(\Delta t^{\epsilon}) \right\} \times \left\{ 1 - \mathcal{O}(\Delta t^{\epsilon}) \right\} \right]^{\gamma} = [1 + \mathcal{O}(\Delta t^{\epsilon})]^{\gamma} = 1 + \mathcal{O}(\Delta t^{\epsilon}),
$$

so that

$$
R_t^{(\gamma)}(x,\tilde{x}|y) = \left[ \frac{R_t(x,\tilde{x}|y)}{R_t(x,\tilde{x})} \right]^{\gamma} [1 + \mathcal{O}(\Delta t^{\epsilon})] R_t(x,\tilde{x}) = R_t(x,\tilde{x}|y)^{\gamma} R_t(x,\tilde{x})^{1-\gamma} + \mathcal{O}(\Delta t^{\epsilon}).
$$
(23)

Inserting this expression of $R_t^{(\gamma)}(x,\tilde{x}|y)$ into the temperature-modulated version of the conditional transition probabilities (Equation 20), we obtain

$$
\begin{aligned}
p^{(\gamma)}(x_{t+\Delta t}=\tilde{x}|x_t=x,y) &= \delta_{x,x'} + R_t^{(\gamma)}(x,\tilde{x}|y)\Delta t + \mathcal{O}(\Delta t^{1+\epsilon}) \\
&= \delta_{x,\tilde{x}} + \left\{ R_t(x,\tilde{x}|y)^{\gamma} R_t(x,\tilde{x})^{1-\gamma} + \mathcal{O}(\Delta t^{\epsilon}) \right\} \Delta t + \mathcal{O}(\Delta t^{1+\epsilon}) \quad (24) \\
&= \delta_{x,\tilde{x}} + R_t(x,\tilde{x}|y)^{\gamma} R_t(x,\tilde{x})^{1-\gamma} \Delta t + \mathcal{O}(\Delta t^{1+\epsilon}).
\end{aligned}
$$

Thus, instead of adjusting the unconditional rates $R_t(x,\tilde{x})$ employing *predictor guidance* at an inverse guidance temperature $\gamma = 1/T$, which we call the guidance strength, with a predictive distribution $p(y|x_{t+\Delta t}, x_t)$ *via* Equation 20, we can equivalently use *predictor-free guidance* at guidance strength $\gamma$ using the unconditional rates and the conditional rates $R_t(x,\tilde{x}|y)$ *via* Equation 24. Note that in practice, one would learn both a conditional, $R_t^{\varphi}(x,\tilde{x}|y)$, and an unconditional, $R_t^{\theta}(x,\tilde{x})$, rate matrix (induced by a conditional, $p_{1|t}^{\varphi}(x_1|x_t,y)$, and an unconditional model, $p_{1|t}^{\theta}(x_1|x_t)$, respectively) and combine them to employ *predictor-free guidance* with $R_t^{(\gamma)}(x,\tilde{x}|y) = R_t^{\varphi}(x,\tilde{x}|y)^{\gamma} R_t^{\theta}(x,\tilde{x})^{1-\gamma}$.

### C.8 SUMMARY OF GUIDANCE IN CTMCS

Summarizing our results, we have demonstrated that, in the limit $\Delta t \to 0$, one can sample the next state $x_{t+\Delta t} = \tilde{x}$ (starting in a state $x_t = x$) at guidance strength (corresponding to the inverse guidance temperature) $\gamma = 1/T$ from

$$
p^{(\gamma)}(x_{t+\Delta t}=\tilde{x}|x_t=x,y) = \delta_{x,\tilde{x}} + R_t^{(\gamma)}(x,\tilde{x}|y)\Delta t + \mathcal{O}(\Delta t^{1+\epsilon}).
$$

using either the *predictor-guided* (PG) version of the rates (Equation 19)

$$
R_t^{(\gamma)}(x,\tilde{x}|y) = \left[ \frac{p(y|\tilde{x},t)}{p(y|x,t)} \right]^{\gamma} R_t(x,\tilde{x})
$$
(25)

with (learnable) predictor distribution $p(y|x^*, t)$, or the *predictor-free-guided* (PFG) version of the rates (Equation 23)

$$R_t^{(\gamma)}(x, \tilde{x}|y) = R_t(x, \tilde{x}|y)^\gamma R_t(x, \tilde{x})^{1-\gamma} \tag{26}$$

for $\tilde{x} \neq x$ in both Equations 25 and 26. When $\tilde{x} = x$, as mentioned earlier based on conservation of flow for rate matrices, we use

$$R_t^{(\gamma)}(x, x|y) = -\sum_{x' \neq x} R_t^{(\gamma)}(x, x'|y), \tag{27}$$

for both PG and PFG. For CTMCs realized with CTDD (Campbell et al., 2022) and DFM (Campbell et al., 2024) only one component of the state can change in infinitesimal time, so that only $D \times (S - 1) + 1$ rates $R_t(x, x')$ are non-zero. Thus, only the corresponding $D \times (S - 1) + 1$ rates $R_t^{(\gamma)}(x, x'|y)$ are non-zero, therefore ensuring that the *identity* rate $R_t^{(\gamma)}(x, x|y)$ (Equation 27) is tractable.

### C.9 GENERALITY OF OUR RESULTS

The results summarized in Equations 25, 26 and 27 assumed only that the rates of a general (unguided) CTMC are known. In practice, one requires tractability of the identity rate (*i.e.*, the sum in Equation 27) to apply guidance to the CTMC. However, similar to CTMCs realized with CTDD (Campbell et al., 2022) and DFM (Campbell et al., 2024), tractability of the unconditional identity rate

$$R_t(x, x) = -\sum_{x' \neq x} R_t(x, x')$$

or the conditional identity rate

$$R_t(x, x|y) = -\sum_{x' \neq x} R_t(x, x'|y),$$

where $R_t(x, x')$ and $R_t(x, x'|y)$ are induced by an unconditional and a conditional, respectively, (*e.g.*, denoising) model in any (unguided) CTMC (as discussed in Appendix B), should guarantee tractability of the corresponding identity rate (Equation 27) in a guided CTMC (Appendix C.8).

Note that we have only instantiated Discrete Guidance for CTDD (Campbell et al., 2022) and DFM (Campbell et al., 2024). Conceptually, however, our framework applies generally to any generative model that can be realized by a CTMC for which transitions are independent across dimensions. It does not matter (for Discrete Guidance) how these rate matrices have been determined (*e.g.*, within a continuous-time discrete space diffusion or flow-model framework, or some other approach not yet invented). We are currently only aware of CTMCs used for generative modeling in discrete-state spaces realized with diffusion (Campbell et al., 2022; Lou et al., 2023; Sun et al., 2023b) and flow matching (Campbell et al., 2024; Gat et al., 2024), but one can envision that other frameworks might be developed for this purpose in the future.

## D IMPLEMENTATION DETAILS

In this section we provide implementation details of Discrete Guidance, both as algorithmic summary and PyTorch code. First, we describe the procedure of training a discrete state-space flow models (DFM) under a masking process as detailed in Campbell et al. (2024) (Algorithm 1). Then, we describe the procedure for obtaining the guide adjusted rates (Algorithm 2), both with exact calculations (Listing 1) and Taylor-approximated guidance (TAG) (Listing 2). Finally, we also provide the minimal implementations for how Discrete Guidance can be integrated with the sampling of a DFM with masking process (Algorithm 3; predictor guidance in Listing 3; predictor-free guidance in Listing 4). We note that the algorithmic summaries are only meant for illustrative purpose and we refer the readers to the PyTorch implementations for more detailed information.

The implementations for DFM sampling are adapted from the DFM GitHub repository.[3] We also provide a derivation for extending the DFM to work for states with fixed components (Appendix D.4).

---

[3]https://github.com/andrew-cr/discrete_flow_models

## D.1 TRAINING DISCRETE STATE-SPACE FLOW MODELS

The procedure for training a discrete state-space flow models (DFM) under a masking process is detailed in Campbell et al. (2024) and we summarized it in Algorithm 1.

---

**Algorithm 1** Training Flow Matching Model with Masking Process

---

**Require:** Training dataset $\{x_1\}$, denoising model $p_{1|t}^{\theta}(x_1|x_t)$, learning rate $\beta$
**Ensure:** Trained model parameters $\theta$
1: **for** each batch $x_1$ in dataset **do**
2:      Sample uniform time $t \sim \mathcal{U}(0,1)$ for batch
3:      Sample $x_t \sim p_{t|1}^{\text{mask}}(x_t|x_1) = \text{Cat}(t\delta\{x_1, x_t\} + (1-t)\delta\{M, x_t\})$        ▷ Forward process
4:      Get logits from $p_{1|t}^{\theta}(x_1|x_t)$        ▷ Shape $(B, D, S)$
5:      $\mathcal{L}_{ce}^{\theta} = -\mathbb{E}_{x_1, t, x_t}[\log p_{1|t}^{\theta}(x_1|x_t)]$        ▷ Cross entropy loss
6:      $\theta \leftarrow \theta - \beta\nabla_{\theta}\mathcal{L}_{ce}^{\theta}$        ▷ Gradient update
7: **end for**

---

## D.2 CALCULATION OF GUIDE-ADJUSTED RATES

The procedure for obtaining the guide adjusted rates is summarized in Algorithm 2. PyTorch implementations are provided for exact calculations (Listing 1) and Taylor-approximated guidance (TAG) (Listing 2).

---

**Algorithm 2** Computing Guide-Adjusted Rates

---

**Require:** Predictor model $p(y|x, t)$, current state $x_t$, time $t$, target property $y$, unconditional rates $R_t(x_t, \cdot) \in \mathbb{R}^{D \times S}$ (with dimension size $D$ and state space size $S$), guidance strength $\gamma$
**Ensure:** Guide-adjusted rates $R_t^{(\gamma)}(x_t, \cdot) \in \mathbb{R}^{D \times S}$
1: **if** using exact guidance **then**
2:      $\log p_{x_t} \leftarrow \log p(y|x_t, t)$
3:      $\{\tilde{x}\} \leftarrow \texttt{get\_next\_states}(x_t)$        ▷ All possible next states
4:      **for** each possible next state $\tilde{x}$ **do**
5:          $\log p_{\tilde{x}} \leftarrow \log p(y|\tilde{x}, t)$
6:          $\alpha(x_t, \tilde{x}) \leftarrow \log p_{\tilde{x}} - \log p_{x_t}$        ▷ Log ratio for this next state
7:      **end for**
8:      Collate $\alpha(x_t, \tilde{x})$ for all $\tilde{x}$ into $\alpha(x_t, \cdot)$
9: **else**        ▷ using TAG
10:      $\mathbf{x_t} \leftarrow \texttt{one\_hot\_encode}(x_t)$        ▷ $\mathbf{x_t} \in \mathbb{R}^{D \times S}$ is the one-hot encoding of $x_t$
11:      $g \leftarrow \nabla_{\mathbf{x_t}} \log p(y|\mathbf{x_t}, t)$        ▷ $\nabla_x \log p(y|x, t)$ at $\mathbf{x_t}$
12:      $\alpha(x_t, \cdot) \leftarrow g - (\mathbf{x_t} \odot g)\mathbf{1}$        ▷ Taylor approximation for all next states, where $\mathbf{1} = \{1\}^{S \times S}$
13: **end if**
14: $\alpha(x_t, \cdot) \leftarrow \gamma\alpha(x_t, \cdot)$        ▷ Apply guidance strength
15: $R_t^{(\gamma)}(x_t, \cdot) \leftarrow R_t(x_t, \cdot) \odot \exp(\alpha(x_t, \cdot))$        ▷ Adjust rates
16: **return** $R_t^{(\gamma)}(x_t, \cdot)$

---

Listing 1: Compute the exact guide-adjusted rates

```python
import torch
import torch.nn.functional as F

def get_guided_rates_exact(pred_model, xt, t, y_guide, S, R_t, guide_temp
    ):
    """
    Obtain the exact guide-adjusted rates for predictor guidance (PG)

    Variables: B for batch size, D for number of dimensions, S for state-
        space size

    Assumes we are given a trained noisy predictor `pred_model` that
        takes as input xt of shape (B, D) and time t of shape (B,).
        Assumes `pred_model` implements a function`get_log_prob(xt, t, y)
        ` which takes inputs xt, t and guidance values y of shape (B,)
        and outputs log p(y | xt, t) of shape (B,).

    Also assumes we are given a function `get_all_jump_transitions(xt)`
        that takes as input xt of shape (B, D) and outputs xt_jumps of
        shape (B*D*S, D) with all states after an identity transition,
        plus all possible states after 1 jump from xt.

    Args:
        model (nn.Module): Trained noisy predictor.
        xt (torch.tensor): Noised input, shape (B, D).
        t (torch.tensor): Time values used to obtain xt, shape (B,).
        y_guide (torch.tensor): Guidance values we want to condition on,
            shape (B,).
        S (int): Size of the input state-space of xt.
        R_t (torch.tensor): The learned unconditional rates used for
            generative sampling, shape (B, D, S).
        guide_temp (float): Guidance temperature.

    Returns:
        (torch.tensor): The guide-adjusted rates used for sampling with
            guidance, shape (B, D, S).
    """
    B, D = xt.shape
    # Get log prob of xt
    log_prob_xt = model.get_log_prob(xt, t, y_guide)

    # Get log prob of all possible states after 1 jump
    xt_jumps = get_all_jump_transitions(xt)
    log_prob_xt_jumps = model.get_log_prob(
        xt_jumps, t.repeat(1, D*S).flatten(),
        y_guide.repeat(1, D*S).flatten()
    ).view(B, D, S)

    # Compute log likelihood ratios
    log_prob_ratio = log_prob_xt_jumps - log_prob_xt.view(B, 1, 1)

    # Temperature adjustment
    log_prob_ratio /= guide_temp

    # Adjust the unconditional rates
    prob_ratio = torch.exp(log_prob_ratio)
    R_t = R_t * prob_ratio
    return R_t
```

Listing 2: Compute the guide-adjusted rate matrices with Taylor-approximated guidance (TAG)

```python
import torch
import torch.nn.functional as F

def get_guided_rates_tag(model, xt, t, y_guide, S, R_t, guide_temp):
    """
    Obtain the guide-adjusted rates for predictor guidance (PG) with
        Taylor-approximated guidance (TAG)

    Variables: B for batch size, D for number of dimensions, S for state-
        space size

    Assumes we are given a trained noisy predictor `model` that takes as
        input xt of shape (B, D) and time t of shape (B,). Assumes `model
        ` implements a function `get_log_prob(xt, t, y)` which takes
        inputs xt, t and guidance values y of shape (B,) and outputs log
        p(y | xt, t) of shape (B,).

    Note this function assumes the data is categorical, not ordinal.

    Args:
        model (nn.Module): Trained noisy predictor.
        xt (torch.tensor): Noised input, shape (B, D).
        t (torch.tensor): Time values used to obtain xt, shape (B,).
        y_guide (torch.tensor): Guidance values we want to condition on,
            shape (B,).
        S (int): Size of the input state-space of xt.
        R_t (torch.tensor): The learned unconditional rates used for
            generative sampling, shape (B, D, S).
        guide_temp (float): Guidance temperature.

    Returns:
        (torch.tensor): The guide adjusted rates with TAG used for
            sampling with guidance, shape (B, D, S)
    """
    B, D = xt.shape
    # One-hot encode xt, then compute log prob of xt with gradient
        calculated with autograd
    xt = F.one_hot(xt.long(), num_classes=S).to(torch.float)
    with torch.enable_grad():
        xt.requires_grad_(True)
        log_prob_xt = model.get_log_prob(xt, t, y_guide)
        log_prob_xt.sum().backward()
        grad_log_prob_xt = xt.grad
    # Compute likelihood ratios with TAG
    log_prob_ratio = grad_log_prob_xt - (xt * grad_log_prob_xt).sum(dim
        =-1, keepdim=True)

    # Temperature adjustment
    log_prob_ratio /= guide_temp

    # Adjust the unconditional rates
    prob_ratio = torch.exp(log_prob_ratio)
    R_t = R_t * prob_ratio
    return R_t
```

### D.3 DFM SAMPLING WITH GUIDANCE

The procedure to perform guided sampling with a discrete flow model with a masking process is illustrated in Algorithm 3. PyTorch implementation for predictor guidance is in Listing 3 and predictor-free guidance is in Listing 4.

---

**Algorithm 3** Guided Discrete Flow Matching Sampling with Masking Process

---

**Require:** Denoising model $p_{1|t}^{\theta}(x_1|x_t)$, predictor model $p^{\phi}(y|x_t, t)$ or conditional model $p_{1|t}^{\varphi}(x_1|x_t, y)$, target property $y$, guidance strength $\gamma$, step size $dt$

**Ensure:** Generated sequence $x_1$

 1: Initialize $x_t$ with all mask tokens except fixed positions
 2: $t \leftarrow 0$
 3: **while** $t < 1$ **do**
 4:     **if** using predictor guidance **then**
 5:         Get logits from $p_{1|t}^{\theta}(x_1|x_t)$
 6:         Compute unconditional rates $R_t^{\theta}(x_t, \cdot)$                $\triangleright$ Equation 6 in Appendix B
 7:         $R_t^{(\gamma)}(x_t, \cdot) \leftarrow \texttt{compute\_guided\_rates}\big(p^{\phi}(y|x_t, t), x_t, t, y, R_t^{\theta}(x_t, \cdot), \gamma\big)$
 8:     **else if** using predictor-free guidance **then**
 9:         Get unconditional logits from $p_{1|t}^{\theta}(x_1|x_t)$ and conditional logits from $p_{1|t}^{\varphi}(x_1|x_t, y)$
10:         Compute $R_t^{\theta}(x_t, \cdot)$ and $R_t^{\varphi}(x_t, \cdot|y)$        $\triangleright$ Equation 6 in Appendix B
11:         $R_t^{(\gamma)}(x_t, \cdot) \leftarrow R_t^{\varphi}(x_t, \cdot|y)^{\gamma} R_t^{\theta}(x_t, \cdot)^{1-\gamma}$
12:     **end if**
13:     Adjust diagonal rates and normalize
14:     Sample next $x_t$ from transition probabilities derived from $R_t^{(\gamma)}(x_t, \cdot)$
15:     $t \leftarrow t + dt$
16: **end while**
17: **return** $x_t$

---

Listing 3: DFM sampling loop with the masking process, with predictor guidance (PG)

```python
import torch
import torch.nn.functional as F
from torch.distributions.categorical import Categorical

"""
Variables: B for batch size, D for number of dimensions, S for state-
    space size. We specify the guide values `y_guide` of shape (B,) that
    we want to condition on

Assumes we have a trained denoising model `denoise_model` that takes as
    input xt of shape (B, D) and time t of shape (B,) and outputs x1
    prediction logits of shape (B, D, S)

Also assumes we are given a trained noisy predictor `pred_model` that
    takes as input xt of shape (B, D) and time t of shape (B,). Assumes `
    pred_model` implements a function `get_log_prob(xt, t, y)` which
    takes inputs xt, t and guidance values y of shape (B,) and outputs
    log p(y | xt, t) of shape (B,).
"""

dt = 0.001  # Euler step size
noise = 0  # DFM stochasticity
guide_temp = 1.0  # Guidance temperature
x1_temp = 1  # Temperature for the x1 prediction logits
mask_token_id  = S - 1  # Mask token id

model.eval()
xt = mask_token_id * torch.ones((B, D), dtype=torch.long)
t = 0.0
```

```
22  mask_one_hot = torch.zeros((S,))
23  mask_one_hot[mask_token_id] = 1.0
24
25  while t < 1.0:
26      # Get p(x1 | xt), scaled by temperature, shape (B, D, S)
27      logits = model(xt, t * torch.ones((B,)))
28      pt_x1_probs = F.softmax(logits / x1_temp, dim=-1)
29
30      # Compute the rates and the probabilities
31      # When the current state is masked, compute rates for unmasking
32      xt_is_mask = (xt == mask_token_id).view(B, D, 1).float()
33      R_t = xt_is_mask * pt_x1_probs * ((1 + noise * t) / (1 - t))
34      # When the current state is not a mask, compute rates for remasking
35      remask_rates = (1 - xt_is_mask) * mask_one_hot.view(1, 1, -1) * noise
36      R_t += remask_rates
37
38      # Perform predictor guidance by adjusting the unconditional rates
39      R_t = get_guided_rates(pred_model, xt, t, y_guide, S, R_t, guide_temp
              )
40
41      # Adjust the diagonals of the rates to negative row sum
42      R_t.scatter_(-1, xt[:, :, None], 0.0)
43      R_t.scatter_(-1, xt[:, :, None], (-R_t.sum(dim=-1, keepdim=True)))
44
45      # Obtain probabilities from the rates
46      step_probs = (R_t * dt).clamp(min=0.0, max=1.0)
47      step_probs.scatter_(-1, xt[:, :, None], 0.0)
48      step_probs.scatter_(-1, xt[:, :, None], (1.0 - torch.sum(step_probs,
              dim=-1, keepdim=True)).clamp(min=0.0))
49      step_probs = torch.clamp(step_probs, min=0.0, max=1.0)
50
51      # Sample the next xt, shape (B, D)
52      xt = torch.distributions.Categorical(step_probs).sample()
53
54      t += dt
```

Listing 4: DFM sampling loop with the masking process, with predictor-free guidance (PFG)

```
1   import torch
2   import torch.nn.functional as F
3   from torch.distributions.categorical import Categorical
4
5   """
6   Variables: B for batch size, D for number of dimensions, S for state-
        space size
7   We specify the guide values `y_guide` of shape (B,) that we want to
        condition on
8
9   Assumes we have a trained denoising model `model` that takes as input xt
        of shape (B, D), time of shape (B,) and the guide values `y_guide` of
         shape (B,) that we want to condition on, and outputs x1 prediction
        logits of shape (B, D, S)
10
11  Also assumes we are given `no_cls_token_id`, the input to the denoising
        model that represents the input for the unconditional case
12  """
13
14  dt = 0.001  # Euler step size
15  noise = 0  # DFM stochasticity
16  guide_temp = 1.0  # Guidance temperature
17  x1_temp = 1  # Temperature for the x1 prediction logits
18  mask_token_id  = S - 1  # Mask token id
19  eps = 1e-9  # For numerical stability in the log
20
21  model.eval()
```

```python
22  xt = mask_token_id * torch.ones((B, D), dtype=torch.long)
23  t = 0.0
24  mask_one_hot = torch.zeros((S,))
25  mask_one_hot[mask_token_id] = 1.0
26
27  while t < 1.0:
28      # Get p(x1 | xt) for the unconditional prediction, scaled by
              temperature
29      cls_input_uncond = (no_cls_token_id * torch.ones((B,))).long()
30      t_input = t * torch.ones((B,))
31      logits = model(xt, t_input, cls_input_uncond) # (B, D, S)
32      pt_x1_probs = F.softmax(logits / x1_temp, dim=-1)
33
34      # Also get the conditional prediction for the class we want to guide
              towards: p(x1 | xt, y)
35      logits_cond = model(xt, t_input, y_guide) # (B, D, S)
36      pt_x1_probs_cond = F.softmax(logits_cond / x1_temp, dim=-1)
37
38      # When the current state is masked, compute rates for unmasking
39      xt_is_mask = (xt == mask_token_id).view(B, D, 1).float()
40      # Compute unconditional rates, shape (B, D, S)
41      R_t = xt_is_mask * pt_x1_probs * ((1 + noise * t) / (1 - t))
42      # Compute conditional rates, shape (B, D, S)
43      R_t_cond = xt_is_mask * pt_x1_probs_cond * ((1 + noise * t) / (1 - t)
              )
44
45      # When the current state is not a mask, compute rates for remasking
46      remask_rates = (1 - xt_is_mask) * mask_one_hot.view(1, 1, -1) * noise
47      # Add remask rates to unconditional rates
48      R_t += remask_rates
49
50      # Perform predictor-free guidance by using both the unconditional and
              conditional rates
51      # First add the remask rates to the conditional rates
52      R_t_cond += remask_rates
53      # Perform rate adjustment, note that we scale by the inverse
              temperature
54      # If inverse_guide_temp = 0 (guide_temp = inf), equivalent to
              unconditional
55      # If inverse_guide_temp = 1, (guide_temp = 1), equivalent to
              conditional
56      inverse_guide_temp = 1 / guide_temp
57      R_t = torch.exp(inverse_guide_temp * torch.log(R_t_cond + eps) + (1 -
              inverse_guide_temp) * torch.log(R_t + eps))
58
59      # Adjust the diagonals of the rates to negative row sum
60      R_t.scatter_(-1, xt[:, :, None], 0.0)
61      R_t.scatter_(-1, xt[:, :, None], (-R_t.sum(dim=-1, keepdim=True)))
62
63      # Obtain probabilities from the rates
64      step_probs = (R_t * dt).clamp(min=0.0, max=1.0)
65      step_probs.scatter_(-1, xt[:, :, None], 0.0)
66      step_probs.scatter_(-1, xt[:, :, None], (1.0 - torch.sum(step_probs,
              dim=-1, keepdim=True)).clamp(min=0.0))
67      step_probs = torch.clamp(step_probs, min=0.0, max=1.0)
68
69      # Sample the next xt, shape (B, D)
70      xt = torch.distributions.Categorical(step_probs).sample()
71
72      t += dt
```

## D.4 Discrete Flow Matching for States with Fixed Components

In some of our experiments, we work with padded sequences, where padded tokens stay padded during the noising process. The noising processes discussed in Campbell et al. (2024) for DFM are noising to either (i) a fully masked or (ii) a uniform stationary-state at $t = 0$. These processes do not directly include the case where the component of a state (*e.g.*, the token of a sequence) is fixed (*e.g.*, padded) during noising. Here, we will show how to augment the masking process (i) for states with a fixed component and derive the rates for this scenario in analogy to the derivation of the rates for the masking process in Campbell et al. (2024) whose notation we adopt here for this derivation. We will refer to the $d$-th component of a state, $x_t$, as the *component-state $x_t^d$*.

The data-conditional (noising) flow factorizes over the components $d \in \{1, \ldots, D\}$

$$p_{t|1}(x_t^{1:D}|x_1^{1:D}) = \prod_{d=1}^{D} p_{t|1}(x_t^d|x_1^d)$$

where the data-conditional flow of the masking process in component $d$, $p_{t|1}(x_t^d|x_1^d)$, is given by (Campbell et al., 2024)

$$p_{t|1}(x_t^d|x_1^d) = t\delta_{x_t^d,x_1^d} + (1-t)\delta_{x_t^d,x_M^d} \equiv p_{t|1}^{\mathrm{mask}}(x_t^d|x_1^d) \tag{28}$$

using $x_M^d$ to denote the masked component-state of component $d$.

Here, we introduce an additional *fixed* component-state $x_F^d$ (for an arbitrary component $d$) that remains unchanged during noising (*i.e.*, masking). In analogy to Equation 28, the data-conditional (denoising) flow in component $d$ for a masking process including a fixed component-state can be written as

$$
\begin{aligned}
p_{t|1}(x_t^d|x_1^d) &= \delta_{x_1^d,x_F^d} + (1-\delta_{x_1^d,x_F^d})p_{t|1}^{\mathrm{mask}}(x_t^d|x_1^d) \\
&= \delta_{x_1^d,x_F^d} + (1-\delta_{x_1^d,x_F^d})\Big[t\delta_{x_t^d,x_1^d} + (1-t)\delta_{x_t^d,x_M^d}\Big].
\end{aligned} \tag{29}
$$

The intuition behind this choice is that the data-conditional flow leaves the component-state $x_1^d$ unchanged for all times, $t$, if it corresponds to the fixed component-state $x_F^d$ at $t = 1$. Otherwise, it masks the component-state as in the standard masking case (Equation 28). The time derivative of Equation 29 is given by

$$\frac{\partial p_{t|1}(x_t^d|x_1^d)}{\partial t} = (1-\delta_{x_1^d,x_F^d})\Big[\delta_{x_t^d,x_1^d} - \delta_{x_t^d,x_M^d}\Big]$$

and can be used to determine the data-conditional rates

$$
\begin{aligned}
\bar{R}_t^{*,d}(x_t^d, \tilde{x}^d|x_1^d) &= \frac{\max\Big(0, \frac{\partial p_{t|1}(\tilde{x}^d|x_1^d)}{\partial t} - \frac{\partial p_{t|1}(x_t^d|x_1^d)}{\partial t}\Big)}{\mathcal{Z}_t[x_1^d]p_{t|1}(x_t^d|x_1^d)} \\
&= \frac{(1-\delta_{x_1^d,x_F^d}) \cdot \max\Big(0, \Big[\delta_{\tilde{x}^d,x_1^d} - \delta_{\tilde{x}^d,x_M^d}\Big] - \Big[\delta_{x_t^d,x_1^d} - \delta_{x_t^d,x_M^d}\Big]\Big)}{\mathcal{Z}_t[x_F^d]\delta_{x_1^d,x_F^d} + (1-\delta_{x_1^d,x_F^d})\mathcal{Z}_t[x_1^d]\Big[t\delta_{x_t^d,x_1^d} + (1-t)\delta_{x_t^d,x_M^d}\Big]} \\
&= (1-\delta_{x_1^d,x_F^d})\frac{\delta_{x_t^d,x_M^d}\delta_{\tilde{x}^d,x_1^d}}{1-t}.
\end{aligned} \tag{30}
$$

following the analogous derivation for standard masking in Campbell et al. (2024). We note that $\mathcal{Z}_t[x_1^d] \equiv |\{\bar{x}_t^d : p_{t|1}(\bar{x}_t^d|x_1^d) > 0\}|$ is the number of component-states with finite probability at time $t$ that is $\mathcal{Z}_t[x_F^d] = 1$ for all $t$ and $\mathcal{Z}_t[x_1^d] = 2$ for $x_1^d \neq x_F^d$ state for $t \in (0, 1)$. The data-conditional rates in Equation 30 are zero for $x_1^d = x_F^d$ and have the same expression as in the masking process (Campbell et al., 2024) for $x_1^d \neq x_F^d$.

Following Campbell et al. (2024), one can add additional rates that satisfy detailed balance. Because we do not allow transitions from or to the fixed component-state $x_F^d$, the detailed balance data-conditional rates of component $d$ are given by

$$\bar{R}_t^{\mathrm{DB},d}(x_t^d, \tilde{x}^d|x_1^d) = (1-\delta_{x_1^d,x_F^d})\left[\frac{\eta t}{1-t}\delta_{x_t^d,x_M^d}\delta_{\tilde{x}^d,x_1^d} + \eta\delta_{x_t^d,x_1^d}\delta_{\tilde{x}^d,x_M^d}\right] \tag{31}$$

with stochasticity $\eta$. The rate in Equation 31 is similar to the corresponding rate in the standard masking (Campbell et al., 2024), with the exception that we only allow transitions for $x_1^d \neq x_F^d$ here.

We can combine the data-conditional rates, $\bar{R}_t^{*,d}(x_t^d, \tilde{x}^d | x_1^d)$, with these detailed balance data-conditional rates, $\bar{R}_t^{\text{DB},d}(x_t^d, \tilde{x}^d | x_1^d)$, obtaining

$$
\bar{R}_t^{\text{tot},d}(x_t^d, \tilde{x}^d | x_1^d) = \bar{R}_t^{*,d}(x_t^d, \tilde{x}^d | x_1^d) + \bar{R}_t^{\text{DB},d}(x_t^d, \tilde{x}^d | x_1^d)
$$
$$
= (1 - \delta_{x_1^d, x_F^d}) \left[ \frac{1 + \eta t}{1 - t} \delta_{x_t^d, x_M^d} \delta_{\tilde{x}^d, x_1^d} + \eta \delta_{x_t^d, x_1^d} \delta_{\tilde{x}^d, x_M^d} \right]
$$

that has the same expression as in the standard masking case (Campbell et al., 2024) for $x_1^d \neq x_F^d$. Finally, following the derivation in Campbell et al. (2024), we find the rates, induced by the denoising model $p_{1|t}^\theta(x_1 | x_t)$, for the masking process including fixed component-states

$$
R_t^{\theta,d}(x_t^{1:D}, \tilde{x}^d) = \underset{p_{1|t}^\theta(x_1^d | x_t^{1:D})}{\mathbb{E}} \left[ \bar{R}_t^{\text{tot},d}(x_t^d, \tilde{x}^d | x_1^d) \right] = \sum_{x_1^d} p_{1|t}^\theta(x_1^d | x_t^{1:D}) \bar{R}_t^{\text{tot},d}(x_t^d, \tilde{x}^d | x_1^d)
$$
$$
= \sum_{x_1^d} p_{1|t}^\theta(x_1^d | x_t^{1:D})(1 - \delta_{x_1^d, x_F^d}) \left[ \frac{1 + \eta t}{1 - t} \delta_{x_t^d, x_M^d} \delta_{\tilde{x}^d, x_1^d} + \eta \delta_{x_t^d, x_1^d} \delta_{\tilde{x}^d, x_M^d} \right]
$$
$$
= (1 - \delta_{\tilde{x}^d, x_F^d}) p_{1|t}^\theta(x_1^d = \tilde{x}^d | x_t^{1:D}) \frac{1 + \eta t}{1 - t} \delta_{x_t^d, x_M^d}
$$
$$
+ (1 - \delta_{x_t^d, x_F^d}) \underbrace{p_{1|t}^\theta(x_1^d = x_t^d | x_t^{1:D})}_{=1 - \delta_{x_t^d, x_M^d}} \eta \delta_{\tilde{x}^d, x_M^d}
$$
$$
= (1 - \delta_{\tilde{x}^d, x_F^d}) \underbrace{p_{1|t}^\theta(x_1^d = \tilde{x}^d | x_t^{1:D}) \frac{1 + \eta t}{1 - t} \delta_{x_t^d, x_M^d}}_{= R_{t,\text{unmask}}^{\theta,d}(x_t^{1:D}, \tilde{x}^d)}
$$
$$
+ (1 - \delta_{x_t^d, x_F^d}) \underbrace{\eta \delta_{\tilde{x}^d, x_M^d} (1 - \delta_{x_t^d, x_M^d})}_{= R_{t,\text{remask}}^{\theta,d}(x_t^{1:D}, \tilde{x}^d)}
$$
$$
= (1 - \delta_{\tilde{x}^d, x_F^d}) R_{t,\text{unmask}}^{\theta,d}(x_t^{1:D}, \tilde{x}^d) + (1 - \delta_{x_t^d, x_F^d}) R_{t,\text{remask}}^{\theta,d}(x_t^{1:D}, \tilde{x}^d),
$$

(32)

where we have used that $p_{1|t}^\theta(x_1^d = x_t^d | x_t^{1:D}) = 0$ for $x_t^d = x_M^d$ and $p_{1|t}^\theta(x_1^d = x_t^d | x_t^{1:D}) = 1$ for $x_t^d \neq x_M^d$, so that $p_{1|t}^\theta(x_1^d = x_t^d | x_t^{1:D}) = 1 - \delta_{x_t^d, x_M^d}$ as discussed in Campbell et al. (2024). We have, in the last step of Equation 32, defined $R_{t,\text{remask}}^{\theta,d}(x_t^{1:D}, \tilde{x}^d)$ and $R_{t,\text{unmask}}^{\theta,d}(x_t^{1:D}, \tilde{x}^d)$ as the induced unmasking and remasking rates, respectively, that have the same expression as in the standard masking case (Campbell et al., 2024). Intuitively, Equation 32 illustrates that we can neither unmask to a fixed component-state nor remask a fixed component-state. Thus, fixed component-states become isolated from the unmasking/remasking operations during denoising and remain unchanged from $t = 0$ to $t = 1$ as demanded.

In summary, when noising (*i.e.*, masking) the data, we follow Equation 29 and thus sample $x_t$ using the masking rates $p_{t|1}^{\text{mask}}(x_t^d | x_1^d)$ followed by a restoration of the fixed component-state for all components that have $x_1^d = x_F^d$. This noising procedure is used to learn $p_{1|t}^\theta(x_1^d | x_t^{1:D})$ as in the standard masking case (Campbell et al., 2024). During denoising (*i.e.*, generation), we first determine the unmasking, $R_{t,\text{unmask}}^{\theta,d}(x_t^{1:D}, \tilde{x}^d)$, and remasking, $R_{t,\text{remask}}^{\theta,d}(x_t^{1:D}, \tilde{x}^d)$, rates induced by $p_{1|t}^\theta(x_1^d | x_t^{1:D})$. In a second step, all unmasking or remasking rates are set to zero for which $\tilde{x}^d = x_F^d$ (*i.e.*, unmasking to the fixed component-state is forbidden) or $x_t^d = x_F^d$ (*i.e.*, remasking the fixed component-state is forbidden), respectively.

# E  FURTHER DISCUSSION OF RELATED WORK

In this section, we include additional discussions of related work (Appendix E.1) and a detailed comparisons among various guidance approaches (Appendix E.2).

## E.1 ADDITIONAL DISCUSSION

There are other approaches for constructing continuous-time discrete diffusion models besides Campbell et al. (2022) using alternative objectives to the ELBO (Sun et al., 2023b; Meng et al., 2022; Lou et al., 2023; Santos et al., 2023). Sun et al. (2023b) extend the original insight from Hyvärinen (2007) to train on a ratio-matching loss. Meng et al. (2022) propose concrete score as the generalization of the (Stein) score for discrete settings, which are defined by the rate of change of the likelihood with respect to local directional changes of the input, and learns these scores using a $L^2$-norm based loss. Lou et al. (2023) further refine this approach to learn these data ratios using the score entropy. We note that although these works also involve the notion of data likelihood ratios, they are trained via a score matching objective and do not introduce guidance. Our approach of predictor guidance can also be viewed as using the likelihood ratios of the predictor model to modulate the sampling process, analogous to how the score of the predictor model is used to modulate the sampling process in continuous state-space diffusion.

Another line of works proposed to model discrete data by embedding the data into continuous state-spaces, either on a probabilistic simplex (Richemond et al., 2022; Floto et al., 2023; Avdeyev et al., 2023) or on a continuous latent space (Li et al., 2022; Dieleman et al., 2022; Han et al., 2023; Strudel et al., 2022; Gong et al., 2023; Chen et al., 2023; Gulrajani & Hashimoto, 2024). As an example, Frey et al. (2024) generate samples in discrete state-spaces by way of Langevin MCMC in a continuous latent space, using ideas from empirical Bayes (Robbins, 1992; Saremi & Hyvärinen, 2019), but do not specify how to perform guidance. Some of the approaches proposed to apply guidance to diffusion models on the continuous state-space representations of discrete state-space objects in order to utilize the guidance approaches for continuous state-space diffusion. Li et al. (2022) model text by first embedding the discrete text tokens into a continuous word embeddings, perform standard Gaussian diffusion on the embeddings, then decode the embedding back to discrete tokens. This approach requires specialized modeling choices *e.g.*, reparameterization of the loss function and clamping the predictions to the nearest word embedding, which might be challenging to adapt to domains other than natural languages. In an alternative approach, Gruver et al. (2024) (NOS) propose to perform guidance in the hidden layers of the unconditional diffusion model. However, NOS requires additional training procedures and hyperparameters, *e.g.*, jointly training the unconditional denoising model and the classifier on one hidden layer of the denoising model, potentially limiting the flexibility and adaptability of such approach. We note that approaches which perform guidance in continuous latent spaces lose the discrete structure of the data during guidance, which can be important for settings where the discrete structures contain information that are useful for guidance (Campbell et al., 2024; Qin et al., 2023).

Guidance has also been explored in the framework of discrete-time, discrete state-space diffusion (Vignac et al., 2023). In particular, Vignac et al. (2023) (DiGress) use discrete time, discrete state-space diffusion for graph generation. They propose to perform approximate guidance using gradient on the one-hot representation of the discrete input. Even though the original presentation of guidance in DiGress addresses only graph inputs, we note that guidance with DiGress shares a similar form to Taylor-approximated guidance (Equation 4). However, since DiGress uses a fixed number of time steps at training time, the gradient approximation is applied to all possible transition states. On the other hand, we present a continuous-time formulation which allows guidance to be achieved exactly by adjusting the rates, and derived Taylor-approximated guidance to specifically approximate the likelihood ratios of only the transitions that change a single dimension. Furthermore, Discrete Guidance allows the user to adjust the number of time steps at inference time and choose among different sampling algorithms (*e.g.*, those discussed in Appendix B) for the best performance on downstream tasks. Wang et al. (2024) use a similar guidance approach to DiGress in a diffusion language model for protein sequences.

For autoregressive models, Yang & Klein (2021) propose an approach (FUDGE) of applying guidance by training a predictor on each step of the decoding process and adjusting the decoding probabilities accordingly. This can be viewed as being similar to how a predictor can be trained at various noise scale for predictor guidance in diffusion. In an alternative approach (PPLM), Dathathri et al. (2020) steer the output of a pre-trained language model towards desired attributes by applying gradient ascent on the hidden states using an attribute model. There is also extensive literature on fine-tuning a pre-trained autoregressive models for controllable generations (Schulman et al., 2017; Rafailov et al., 2023), and some of these approaches have been recently applied to tasks in the natural

Table 1: Comparisons between approaches for guidance in continuous and discrete state-spaces, under either a continuous or discrete time formulation.

| Space \ Time | Discrete | Continuous |
|---|---|---|
| Continuous | Sohl-Dickstein et al. (2015) | Song et al. (2021) |
| Discrete | Vignac et al. (2023) | Discrete Guidance (this work) |

sciences (Widatalla et al., 2024; Chennakesavalu et al., 2024). We note that a key advantage of guidance over these approaches is that guidance can leverage a already-trained unconditional model without further fine-tuning, which can be prohibitively expensive. Furthermore, predictor guidance also allows for different predictors to be combined in a modular fashion.

### E.2 COMPARISONS WITH OTHER GUIDANCE APPROACHES

In this section, we make a more detailed comparisons among the approaches for guidance in continuous and discrete state-spaces, under either a continuous or discrete time formulation. We will start by reviewing the derivation from Sohl-Dickstein et al. (2015) for constructing a general conditional reverse diffusion process in a discrete-time continuous state-space diffusion framework and the approximations they introduced. Then, we show that the guidance procedure in DiGress (Vignac et al., 2023) corresponds to making two similar approximations, which we refer to as the *time-discretization approximation* and the *gradient approximation*. The time-discretization approximation is made exact in the limit of continuous-time, which corresponds to Taylor-approximated guidance (Equation 4). The gradient approximation is made exact by explicitly calculating the likelihood ratios of all $D \times (S-1)$ possible transitions, corresponding to exact guidance (Equation 2). In the case of continuous state-spaces, no gradient approximation is required and the only approximation error is incurred through time-discretization, which is made exact in the continuous-time, score-based diffusion framework (Song et al., 2021). The relationships between these approaches are summarized in Table 1.

Sohl-Dickstein et al. (2015) note that to condition an unconditional reverse diffusion process given a property label $y$, it suffices to sample each transition according to Equation 11, which we repeat in part here for clarity:

$$
\begin{aligned}
p(x_{t+\Delta t}=\tilde{x}|x_t=x, y) &= \frac{p(y|x_{t+\Delta t}=\tilde{x})p(x_{t+\Delta t}=\tilde{x}|x_t=x)}{p(y|x_t=x)} \\
&= \frac{p(y|x_{t+\Delta t}=\tilde{x})p(x_{t+\Delta t}=\tilde{x}|x_t=x)}{\sum_{x'} p(y|x_{t+\Delta t}=x')p(x_{t+\Delta t}=x'|x_t=x)}
\end{aligned}
\tag{33}
$$

where we used the simplification $p(y|x_{t+\Delta t}=x^*, x_t=x) = p(y|x_{t+\Delta t}=x^*)$ shown in Appendix C.4. It is typically intractable to sample from this distribution exactly due to the intractability of the denominator, which in discrete state-spaces involves summing over all possible states, and in continuous state-spaces corresponds to integrating over all possible states.

Now, suppose the reverse process follows a Gaussian distribution:

$$
p(x_{t+\Delta t}=\tilde{x}|x_t=x) = \mathcal{N}(\mu_t, \Sigma_t)
$$

$$
\log p(x_{t+\Delta t}=\tilde{x}|x_t=x) = -\frac{1}{2}(\tilde{x}-\mu_t)^T \Sigma_t^{-1}(\tilde{x}-\mu_t) + C_t
$$

where $C_t$ is the normalizing constant. Sohl-Dickstein et al. (2015) approximate $\log p(y|x_{t+\Delta t}=\tilde{x})$ using a Taylor expansion around $x_{t+\Delta t} = \mu_t$:

$$
\log p(y|x_{t+\Delta t}=\tilde{x}) \approx \log p(y|x_{t+\Delta t}=\mu_t) + (\tilde{x}-\mu_t)^T \nabla_{x_{t+\Delta t}} \log p(y|x_{t+\Delta t})\big|_{x_{t+\Delta t}=\mu_t}
$$

With this approximation, the conditional transition distribution can be approximated by the following Gaussian distribution with a shifted mean:

$$
p(x_{t+\Delta t}=\tilde{x}|x_t=x, y) \approx \mathcal{N}(\mu_t + \Sigma_t \nabla_{x_{t+\Delta t}} \log p(y|x_{t+\Delta t})\big|_{x_{t+\Delta t}=\mu_t}, \Sigma_t)
$$

Sohl-Dickstein et al. (2015) noted that this approximation is valid under the assumption that $\log p(y|x_{t+\Delta t}=\tilde{x})$ has low curvature compared to $\Sigma_t$. As noted in Dhariwal & Nichol (2021),

this assumption, and correspondingly the approximation, is correct in the limit of infinite diffusion steps, which corresponds to $\Delta t \to 0$ (*i.e.*, in the continuous-time limit), where $\|\Sigma_t\| \to 0$. However, in discrete-time, the error of the approximation depends on the number of time steps chosen at training time and can be large. This approximation is made exact in Song et al. (2021) by leveraging the connection between score matching and denoising diffusion models. In particular, they show that to sample from the conditional distribution in Equation 33, it suffices to sample using the following score function

$$\nabla_{x_{t+\Delta t}} \log p(x_{t+\Delta t}|x_t, y) = \nabla_{x_{t+\Delta t}} \log p(x_{t+\Delta t}|x_t) + \nabla_{x_{t+\Delta t}} \log p(y|x_{t+\Delta t}) \qquad (34)$$

that is obtained by taking a gradient of the first line of Equation 33 with respect to $x_{t+\Delta t}$. Equation 34 is written in terms of conditional score functions, while the scores in Song et al. (2021) are marginal scores of the form $s(x_t) = \nabla_{x_t} \log p(x_t)$, resulting in

$$\nabla_{x_t} \log p_t(x_t|y) = \nabla_{x_t} \log p_t(x_t) + \nabla_{x_t} \log p_t(y|x_t), \qquad (35)$$

which is the marginal version of Equation 34.

In discrete state-spaces, the score function is not defined. We now provide an alternative view of predictor guidance that might provide more intuition on the connection between guidance in discrete and continuous state-spaces. First, we note that the first line of Equation 33 can equivalently be written in log-space as:

$$\log p(x_{t+\Delta t}=\tilde{x}|x_t=x, y) = \log p(x_{t+\Delta t}=\tilde{x}|x_t=x) + [\log p(y|x_{t+\Delta t}=\tilde{x}) - \log p(y|x_t=x)]$$

From this, we can deduce that the quantities we need to estimate in order to perform predictor guidance are indeed the likelihood ratios: $\log p(y|x_{t+\Delta t}=\tilde{x}) - \log p(y|x_t=x)$, for all $\tilde{x} \neq x$. Now, in order to sample from $p(x_{t+\Delta t}=\tilde{x}|x_t=x, y)$, a normalized probability is needed. Therefore, in discrete state-spaces, we still need to estimate this quantity for all $S^D$ possible states. However, as we have shown in Appendix C.3, by working with the rates in CTMCs, we only need to estimate $D \times (S-1) + 1$ of these likelihood ratios to make all the rate adjustments that are needed to sample from the conditional distribution exactly. Furthermore, when $p(y|x_t=x)$ is a continuous function on $x \in \mathbb{R}^{D \times S}$ (*e.g.*, a neural network), we showed that we can also leverage a first-order Taylor series approximation to achieve more efficient sampling:

$$\log p(y|x_{t+\Delta t}=\tilde{x}) - \log p(y|x_t=x) \approx (\tilde{x} - x)^T \nabla_{x_t} \log p(y|x_t)|_{x_t=x}$$

Vignac et al. (2023) (DiGress) uses an approximation of a similar form for guided sampling in a discrete-time, discrete state-space diffusion formulation. They directly introduce a gradient approximation to Equation 33:

$$p(x_{t+\Delta t}=\tilde{x}|x_t=x, y) \approx \frac{\exp\left[g(\tilde{x})\right]p(x_{t+\Delta t}=\tilde{x}|x_t=x)}{\sum_{x'} \exp\left[g(x')\right]p(x_{t+\Delta t}=x'|x_t=x)}$$

where $g(x^*) \equiv (x^* - x)^T \nabla_{x_{t+\Delta t}} \log p(y|x_{t+\Delta t})\big|_{x_{t+\Delta t}=x}$. This approximate reverse distribution is still intractable to sample from due to the intractability of the denominator. To bypass this, DiGress assumes that dimensions factorize and samples each dimension independently. Since DiGress operates in discrete time, this introduces an additional time-discretization approximation, which only becomes exact as $\Delta t \to 0$. In discrete time, however, the error introduced by this approximation depends on the number of time steps fixed during training.

There is another related line of work which performs predictor guidance without explicitly training a time-dependent noisy predictor (Chung et al., 2023; Song et al., 2023). We note that the procedure to compute the guide-adjusted rate matrix in Equation 2 is the same regardless of how one obtains the estimate for $p(y|x_t)$. As noted in Chung et al. (2023), the true distribution $p(y|x_t)$ is intractable, and the closer one's approximation is to the true distribution, the closer the generated samples would be to being from the true posterior distribution $p(x|y)$. Concretely, for continuous state-spaces, one approach for approximating $p(y|x_t)$ is by training a predictor that takes in noisy input, $p^\phi(y|x_t)$, where $x_t$ is the "noisy" version of the input $x$ at time step $t$ in the noising process. Alternatively, in Chung et al. (2023), one estimates $p(y|x_t)$ as $p(y|\hat{x}_1)$, where $\hat{x}_1 = \mathbb{E}_{x_1 \sim p(x_1|x_t)}[x_1]$ and $p(y|x_1)$ is a "clean" predictor trained only on unnoised data. It's important to note that both training a noisy predictor and the approach in Chung et al. (2023) are approximations to the true distribution $p(y|x_t)$.

The training procedure of the predictor can be modified to better match the true $p(y|x_t)$, such as those we detailed in Appendix F.2-F.4, while the approach in Chung et al. (2023) has been shown to perform sub-optimally without additional corrections, *e.g.*, via sequential Monte Carlo (SMC) as suggested by Wu et al. (2024). It is not immediately obvious how to extend these training-free approaches to discrete state-spaces. For example, one potential issue is that when the input space is discrete, $\hat{x}_1 = \mathbb{E}_{x_1 \sim p(x_1|x_t)}[x_1]$ can lie on the probabilistic simplex for one-hot encoded states $x_1$, while $p(y|x_1)$ only takes discrete input (*i.e.*, in the corners of the simplex), although there are potential ways to bypass this issue that are beyond the scope of this work.

## F    EXPERIMENTAL DETAILS

This section contains detailed descriptions and results of each of the experiments: images (Appendix F.1), small molecules (Appendix F.2), enhancer DNA (Appendix F.3) and proteins (Appendix F.4).

### F.1    IMAGE MODELING

Following Campbell et al. (2022), we modeled a CIFAR-10 image dataset as discrete pixels. Campbell et al. (2022) used this dataset to demonstrate successful unconditional modeling on discrete state-spaces, while we used this dataset to demonstrate that Discrete Guidance can readily leverage a pre-trained discrete diffusion models, and can be applied to a high-dimensional problem like images to generate samples with desired properties. Details of model architecture and training can found in Appendix F.1.1 and detailed results are in Appendix F.1.2.

#### F.1.1    MODEL ARCHITECTURE AND TRAINING

We used the pre-trained unconditional denoising network from Campbell et al. (2022), which is a standard U-net architecture. The weights of the pre-trained model are taken from the official repository.[4] The classifier architecture is the downsampling trunk of the same U-net model with a global average pooling layer. We trained the noisy classifier for 200 epochs on the same training set as the denoising model, with noise added using the same forward process used to train the denoising model. We used accuracy evaluated on the validation set for early stopping. We used the Adam optimizer (Kingma & Ba, 2015) with a learning rate of 0.0002 with 5,000 linear warmup steps. We set the dropout rate of the network at 0.1. The input images have random horizontal flips applied to them during training. We trained on one RTX 6000A GPU. Training the classifier for 200 epochs takes 2 hours.

#### F.1.2    RESULTS

For sampling, we used $\tau$-leaping with $\tau = 0.001$ without predictor-corrector steps. To generate class conditional images, we used predictor guidance with Taylor-approximated guidance. Randomly chosen class-conditional samples for all classes are shown for guidance strength $\gamma = 3$ (Figure 5) and $\gamma = 2$ (Figure 6). By visual inspection, we see that guidance produced expected class-conditional images with reasonable quality.

More quantitatively, we also compared the unconditional model used by Campbell et al. (2022), CTDD, and that model with Discrete Guidance using PG (CTDD-DG) on the quality of *unconditional* samples on two metrics: Inception Score (IS) (Salimans et al., 2016) and Fréchet Inception Distance (FID) (Heusel et al., 2017) (by marginalizing out the class labels, as done in Dhariwal & Nichol (2021)). We emphasize that the goal of this comparison was not to claim improved performance on unconditional generation, as unconditional generation is not the goal of guidance, but as a more quantitative check for how well guidance covers the distributions over all target classes. To calculate the Inception Score and FID values in Table 2, we used the same library as Campbell et al. (2022).[5] Intuitively, Inception Score rewards samples that can be well-classified, while the FID aims to better capture nuances and diversity. We observed that by marginalizing out the class labels, CTDD-PG can match, or in some cases exceed the performance of CTDD on unconditional generation metrics, suggesting that guidance produces class-conditional images with comparable sample quality and

---

[4] https://github.com/andrew-cr/tauLDR
[5] https://github.com/w86763777/pytorch-gan-metrics

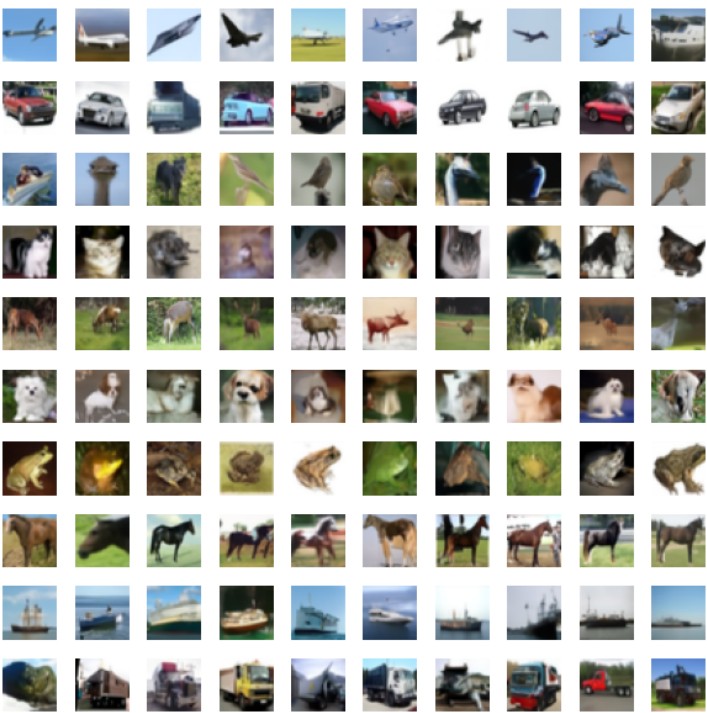

Figure 5: Randomly chosen class-conditional image samples with guidance strength $\gamma = 3$. Each row corresponds to a class. From top to bottom, the classes are: airplane, automobile, bird, cat, deer, dog, frog, horse, ship, truck.

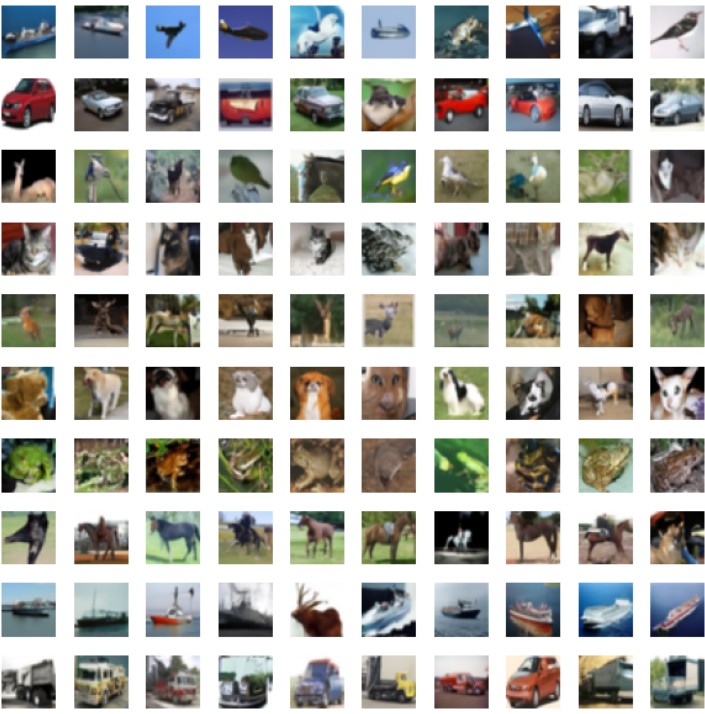

Figure 6: Randomly chosen class-conditional image samples with guidance strength $\gamma = 2$. Each row corresponds to a class. From top to bottom, the classes are: airplane, automobile, bird, cat, deer, dog, frog, horse, ship, truck.

Table 2: Sample quality metrics of unconditional samples generated both with and without guidance. The Inception Score (IS) and Fréchet Inception Distance (FID) were calculated using 50,000 samples. CTDD refers to the baseline model from Campbell et al. (2022), while CTDD-DG are predictor-guided version of that model with Discrete Guidance for each of two guidance strengths ($\gamma$).

| Method | IS ($\uparrow$) | FID ($\downarrow$) |
|---|---|---|
| CTDD | 8.74 | 8.10 |
| CTDD-DG ($\gamma = 2$) | 8.91 | 7.86 |
| CTDD-DG ($\gamma = 3$) | 9.09 | 9.04 |

diversity as its unconditional counterpart. In addition, we observed that increasing the guidance strength from 2 to 3 improved the Inception Score further, while worsening the FID due to decreased sample diversity, suggesting that varying guidance strength might allow one to trade-off sample quality and diversity, aligning with previous observations by Dhariwal & Nichol (2021).

## F.2 MOLECULE GENERATION

Here, we will present the details concerning dataset construction (Appendix F.2.1), model architectures and training (Appendix F.2.2), SMILES string generation (Appendix F.2.3), SMILES string validity (Appendix F.2.4), property-histograms across a wide range of specified molecular property values (Appendix F.2.5), and a comparison of Discrete Guidance to DiGress (Vignac et al., 2023) (Appendix F.2.6).

### F.2.1 DATASET CONSTRUCTION

We constructed our dataset as a subset of QMugs (Isert et al., 2021) that itself contains drug-like molecules extracted from the ChEMBL database (Gaulton et al., 2011). Using RDKit (Landrum, 2010) and the ChEMBL Structure pipeline[6] we preprocessed the molecules of QMugs by (i) standardizing them (removing remaining fragments), (ii) removing stereo-chemical information, and (iii) transforming them to canonical simplified molecular-input line-entry system (SMILES) strings (Weininger, 1988). In the following, we will refer to canonical SMILES strings as SMILES strings if not stated otherwise. 630,508 unique molecules remained after these preprocessing steps.

Next, we removed molecules with a molecular weight of more than 750 Da that corresponds to 1.5 times the upper boundary of 500 Da proposed heuristically by Lipinski's rule of five (Lipinski, 2004) for drug-likeness. We have selected this upper boundary for the weight as a trade-off between number of included molecules and drug-likeness of the molecules. Figure 7a compares the weight-histogram of the unfiltered molecules and of the molecules with weight below 750 Da. After filtering by weight, we removed molecules with a SMILES length above, more than 7 rings, or a LogP value outside [-3, 10]. The (Wildman-Crippen) LogP values have been determined for all molecules with RDKit (including hydrogen atoms in the computation) following the method presented in Wildman & Crippen (1999). As shown in Figure 7b-d, these boundaries (corresponding to the edges of the *Included* area) were chosen to filter molecules in the tails (*Tail-filters*) of the corresponding histograms and did not dilute our dataset. After these filtering steps, 610,575 unique molecules remained in the dataset that was then randomly-split into a train- and a holdout-set with a ratio of 4:1. We used SMILES strings, padded to a length of 100 tokens, as discrete-state representation $x$, resulting in discrete-state space specified by $D = 100$ and a number of $S = 32$ possible tokens (including one pad and one mask token).

### F.2.2 MODEL ARCHITECTURE AND TRAINING

We trained three different fully-connected neural networks (FCNNs); (i) an unconditional discrete flow model with masking (Campbell et al., 2024) $p^\theta(x)$, (ii) a number-of-rings ($N_r$) predictor $p^{\phi_1}(N_r|x,t)$ and (iii) a lipophilicity (LogP) predictor $p^{\phi_2}(\text{LogP}|x,t)$.

The unconditional discrete flow model with masking (Campbell et al., 2024) were parameterized by

$$p^\theta(x_1|x_t) = \text{Categorical}\Big(x_1\Big|p = \text{SoftMax}\big[f^\theta(x_t)\big]\Big)$$

---

[6]https://github.com/chembl/ChEMBL_Structure_Pipeline

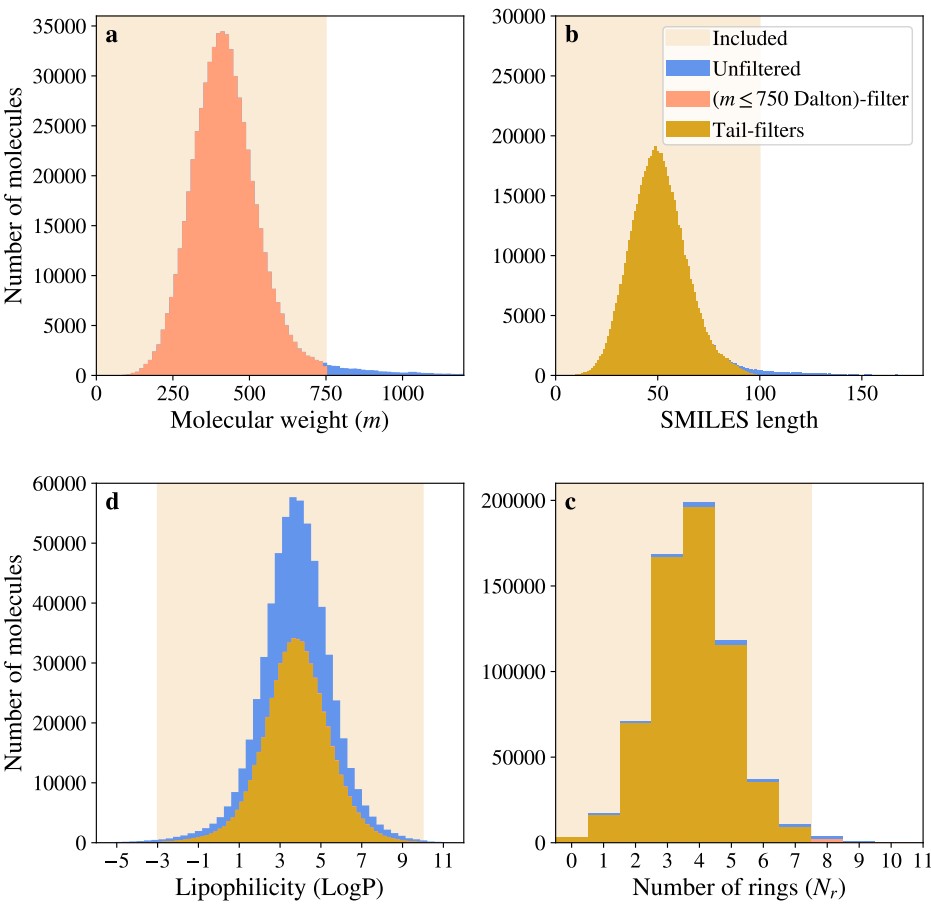

Figure 7: Influence of the filter-operations on the property-distribution of the molecules in the dataset. The *Tail-filter* in panels **b-d** corresponded to the exclusion of molecules with a SMILES length larger than 100 tokens, $\mathrm{LogP}$ values outside [-3, 10] and more than 7 number of rings.

for one-hot-encoded $x_t$ where $f^\theta : \mathbb{R}^{D \times S} \to \mathbb{R}^{D \times S}$ was an FCNN consisting of two hidden layers each containing 20,000 units using ReLU-activation. The $\mathrm{SoftMax}$ function was applied over the $S$ states for each of the $D$ components. When using masking as noising process, the time $t$ is implicitly included in a noisy sample $x_t$ as the average number of mask tokens in $x_t$ is proportional to $t$. In agreement with this intuition, we found that it is sufficient to only pass $x_t$ (without requiring to explicitly pass the time) as input to the unconditional denoising model and also to the predictor models discussed in the next paragraph.

We have used two normal likelihoods as predictor models for $y \in \{N_r, \mathrm{LogP}\}$ in the form

$$p^\phi(y|x_t, t) = \mathrm{Normal}\big(y\big|\mu^\chi(x_t), \sigma^y(t)\big) \tag{36}$$

for one-hot-encoded $x_t$, where the mean $\mu^\chi : \mathbb{R}^{D \times S} \to \mathbb{R}$ was parameterized by an FCNN with learnable parameters $\chi$, with one FCNN for $y = N_r$ and another for $y = \mathrm{LogP}$. Both FCNNs had the same architecture of one hidden layer containing 1,000 units using ReLU-activation. The standard deviation was parameterized by

$$\sigma^y(t) = t\sigma_1^y + (1 - t)\sigma_0^y$$

where $\sigma_1^y$ and $\sigma_0^y$ are the standard deviation at time $t = 1$ (unnoised) and $t = 0$ (fully noised). We treated $\sigma_1^y$ as learnable parameter so that the total set of learnable parameters was given by $\phi = (\chi, \sigma_1^y)$. For $\sigma_0^y$ we used an empirical estimate that we will now briefly motivate. In the fully noised state at $t = 0$, all tokens of all SMILES strings will be in the masked state so that $\mu^\chi$ will output the same value for all molecules and thus most likely learn to predict the mean of training set's property-distribution. $\sigma_0^y$ can therefore be approximated with the empirical standard deviation of the property-distribution over the entire train corresponding to $\sigma_0^{N_r} = 1.24$ for the $N_r$ predictor

and $\sigma_0^{\mathrm{LogP}} = 1.66$ for the LogP predictor. As $N_r$ is an ordinal and not continuous quantity, we have also investigated the use of a discretized normal distribution for $N_r$-prediction finding a comparable performance to the (simpler) continuous normal distribution in Equation 36, and selected the later for our experiments.

We found best performance on the holdout set when not using any dropout for the unconditional discrete flow model and a dropout probability of 0.1 for both predictors. All models have been trained with the Adam optimizer (Kingma & Ba, 2015) and a learning rate of 0.0001 on an RTX 6000A GPU for 100 epochs in case of the unconditional model (corresponding to 263 minutes) and 50 epochs in case of the predictors (corresponding to 8 minutes each).

### F.2.3 SMILES STRING GENERATION

During noising (*i.e.*, masking), we have *fixed* the pad tokens meaning that they have not been masked during the noising process as discussed in Appendix D.4. As a consequence, the denoising process also fixed the pad tokens so that pad tokens could be neither added nor removed during denoising. To generate a molecule $x_1$, we started with a fully masked SMILES string $x_0 = (<\mathrm{MASK}>, <\mathrm{MASK}>, \ldots, <\mathrm{MASK}>)$. We then sampled the number of to be applied pad-tokens, $N_{<\mathrm{PAD}>}$, from a distribution over the number of pads $p(N_{<\mathrm{PAD}>})$. This distribution $p(N_{<\mathrm{PAD}>})$ was obtained empirically from the distribution of SMILES lengths of the molecules in the training set. The fully masked $x_0$ is then padded with the sampled number of pads so that

$$[x_0]_d = \begin{cases} <\mathrm{MASK}> & d \in \{1, \ldots, D - N_{<\mathrm{PAD}>}\} \\ <\mathrm{PAD}> & d \in \{D - N_{<\mathrm{PAD}>} + 1, \ldots, D\}. \end{cases}$$

The padded tokens stayed padded during the entire denoising process, persisting to $x_1$. This approach was inspired by a similar procedure that has been employed for diffusion models operating on graphs as for example in Hoogeboom et al. (2022) where the number of atoms has been sampled at $t = 0$.

Here, we present the generation settings that were used to generate the molecules shown in Fig. 2 in Section 6.1 and in the following of this Appendix section. We employed Euler integration with a step size of 0.001 up to a maximal time of 0.98 followed by an argmax operation, and used Taylor-approximated guidance for generation with a guidance strength of $\gamma \in \{0.2, 1, 2\}$ and a stochasticity value of $\eta = 30$. For both unconditional and conditional generation, we did not set a temperature for the unconditional model (corresponding mathematically to setting the temperature of the unconditional model to 1).

### F.2.4 SMILES STRING VALIDITY

Only a small fraction of the possible $S^D$ SMILES strings are *valid* and correspond to an actual molecule (*i.e.*, a chemical structure), which is a general problem in methods generating SMILES strings (Gómez-Bombarelli et al., 2018; Krenn et al., 2019). Here, we verify the validity of a generated SMILES string by trying to construct a molecule from it using RDKit. As SMILES string validity is crucial for efficient generation of molecules, we have selected the stochasticity ($\eta = 30$) that resulted in the highest validity (on average). With the aforementioned generator settings and this optimal stochasticity, we find a validity of $(12\pm3)\%$ for unconditional generation. For conditional generation we find validity values ranging from $(4\pm2)\%$ (for $N_r^\star = 7$) up to $(19\pm3)\%$ (for $N_r^\star = 0$) where values and uncertainties correspond to the means and standard deviations, respectively, of the validity fractions of 100 SMILES string simultaneously generated in a batch. Intuitively, this makes sense as one cause for invalidity are non-closing rings. Guiding to zero rings discourages the unmasking of dimensions to tokens related to rings during generation thereby increasing validity, In contrast, guiding to a higher number of rings increases the chance to generate SMILES strings containing non-closing rings thereby potentially decreasing validity. Another direction for future research would be to implement Discrete Guidance for generation of molecular graph-structures that might naturally increase validity. For molecular generation presented in Section 6.1 and in this Appendix section (Appendix F.2.5 and F.2.6), we sample SMILES strings until we have generated the requested number of valid SMILES strings (*i.e.*, molecules) that was 1,000 here.

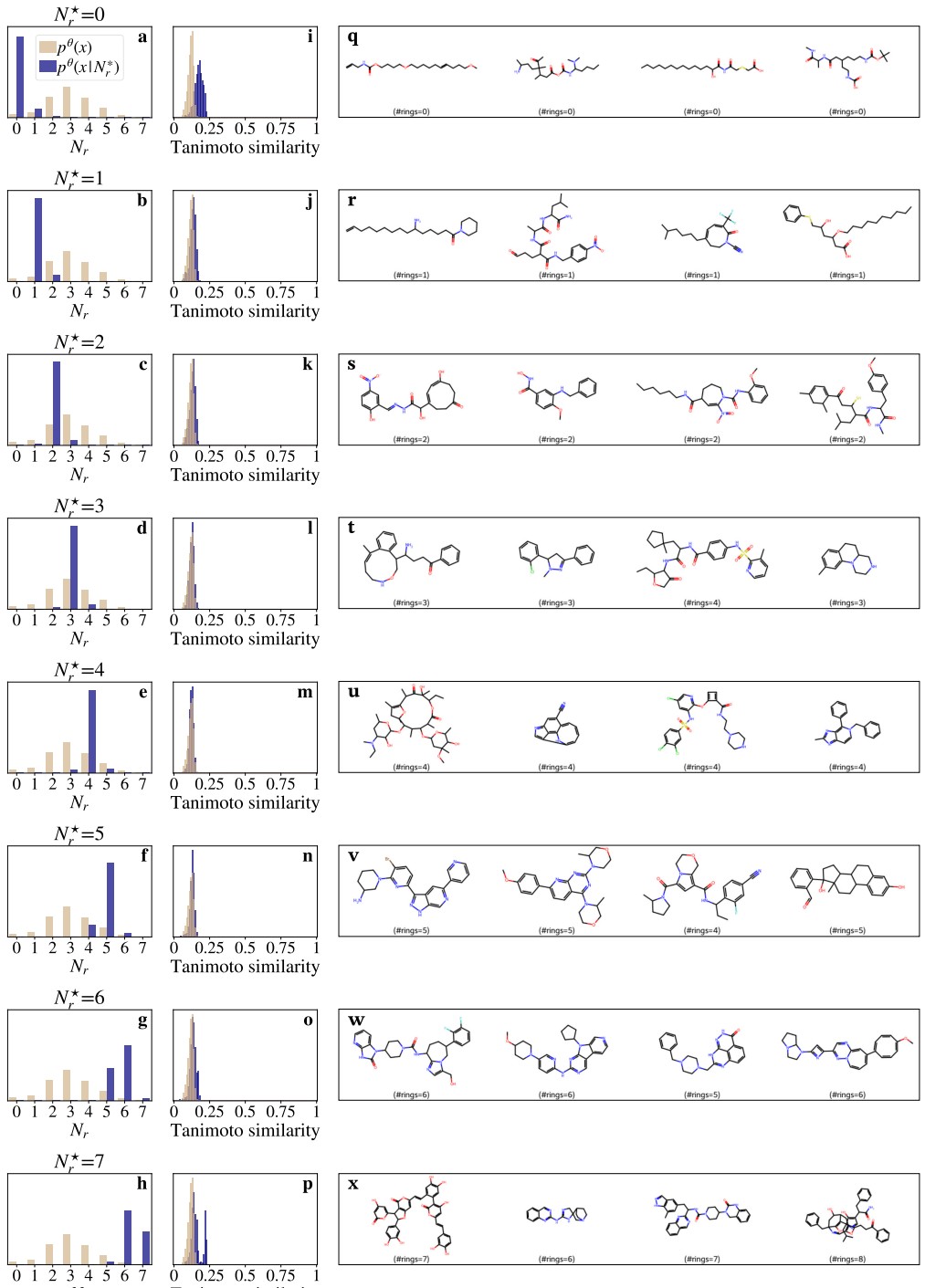

Figure 8: **a-h**, Number of rings ($N_r$) histograms of molecules $x$ generated with an unconditional flow model $p^\theta(x)$ and predictor-guided model $p^\theta(x|N_r^\star)$ for various target $N_r^\star$ values. For both unconditional and guided generation 1,000 molecules (*i.e.*, valid SMILES strings) have been sampled. To obtain $p^\theta(x|N_r^\star)$, $p^\theta(x)$ has been guided with the predictor $p^\phi(N_r^\star|x)$ using a guidance strength of $\gamma = 2$ with Discrete Guidance. **i-p** Distribution of the average Tanimoto similarity (Tanimoto, 1958) of each generated molecule to the others. **q-x**, First four molecules sampled from $p^\theta(x|N_r^\star)$ in the corresponding row in panels **a-h**, where we have added their $N_r$ (=#rings) values as determined by RDKit (Landrum, 2010).

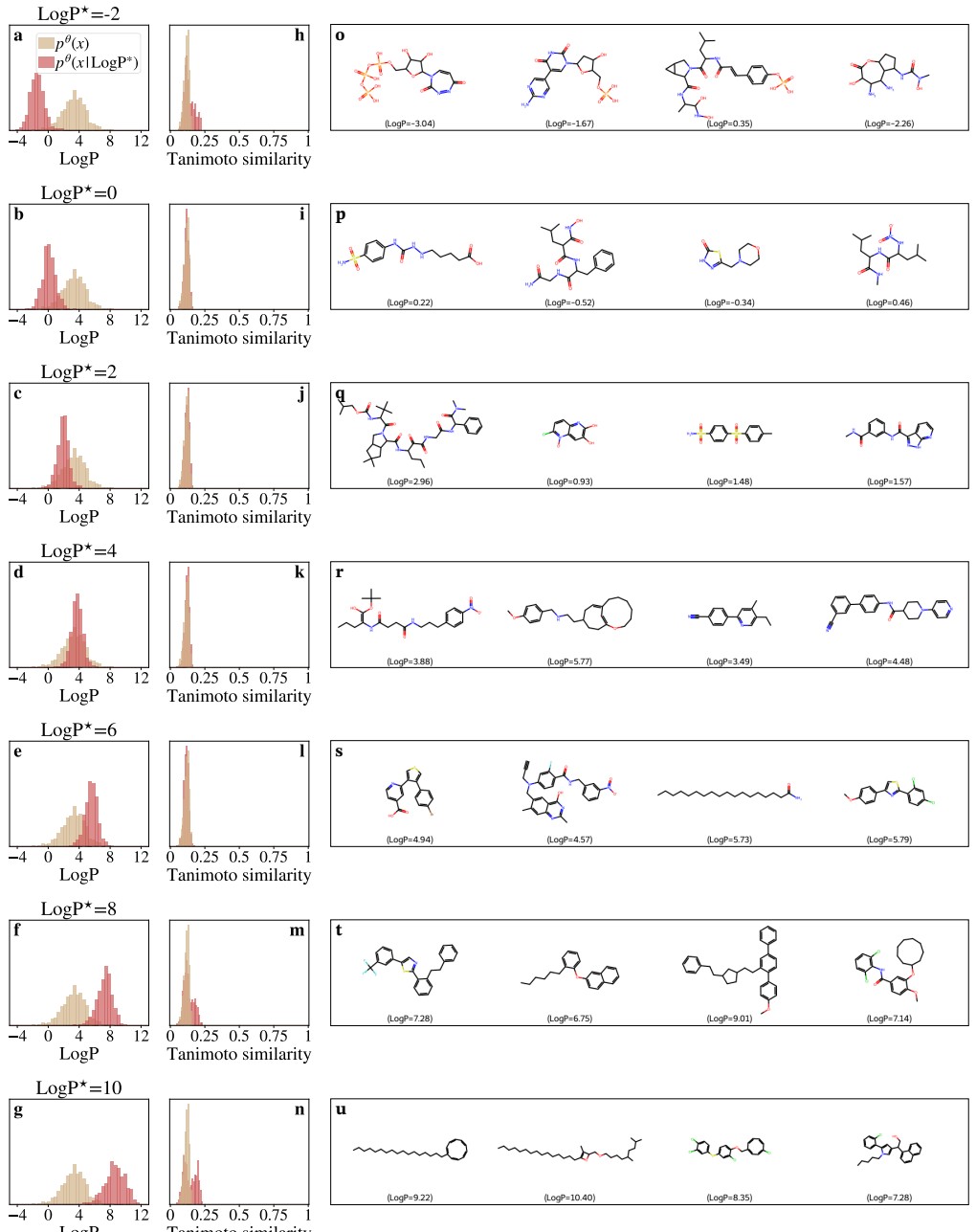

Figure 9: **a-g**, Lipophilicity (LogP) histograms of molecules $x$ generated with an unconditional flow model $p^\theta(x)$ and predictor-guided model $p^\theta(x|\text{LogP}^\star)$ for various target $\text{LogP}^\star$ values. For both unconditional and guided generation 1,000 molecules (*i.e.*, valid SMILES strings) have been sampled. To obtain $p^\theta(x|\text{LogP}^\star)$, $p^\theta(x)$ has been guided with the predictor $p^\phi(\text{LogP}^\star|x)$ using a guidance strength of $\gamma = 2$ with Discrete Guidance. **h-n** Distribution of the average Tanimoto similarity (Tanimoto, 1958) of each generated molecule to the others. **o-u**, First four molecules sampled from $p^\theta(x|\text{LogP}^\star)$ in the corresponding row in panels **a-g**, where we have added their LogP values as determined by RDKit (Landrum, 2010).

## F.2.5 WIDE RANGE CONDITIONAL GENERATION

In Figure 2 of Section 6.1, we only showed two selected target values for the number of rings ($N_r$) and lipophilicity (LogP). Here, we present the property-histogram of molecules generated using predictor guidance $p^\theta(x|y)$ for a wide range of target values for $N_r$ (Figure 8) and for LogP (Figure 9). As

in Figure 2, we compared the property-histogram obtained by guidance to the property-histogram of molecules generated with the unconditional model $p^\theta(x)$. 1,000 valid SMILES strings (*i.e.*, molecules) were generated with a guidance strength of $\gamma = 2$ using Discrete Guidance. We observed a clear shift of the distribution according to the demanded target property values (Figures 8 and 9). The $N_r$ property-histograms are sharp for low $N_r^\star$ and broadens as $N_r^\star$ is increased (Figure 8). The LogP distribution is broad for $\text{LogP}^\star$ values in the tails and sharpens for $\text{LogP}^\star$ in the vicinity of the mode of the unconditional distribution (Figure 9).

We assess the diversity of the generated molecules by determining the Tanimoto similarity (Tanimoto, 1958) of their Morgan fingerprints (Morgan, 1965; Rogers & Hahn, 2010) (with radius 2 and length 1024) constructed with RDKit (Landrum, 2010). Note that a Tanimoto similarity value of 1 is obtained for identical fingerprints, while a value of 0 indicates no similarity between two fingerprints. In both Figure 8 and 9, we display the distribution of the average Tanimoto similarity of each generated molecule to all the others. The average Tanimoto similarity distributions of molecules obtained by unconditional sampling and guided sampling overlap for most specified target properties. However, we observed a shift towards higher values (*i.e.*, slightly less diverse) for guided generation towards properties in the tails of the property distributions (see Figure 8i,o,p and Figure 9h,m,n).

In both Figure 8 and 9, we display the first four generated molecules with their ground truth values as extracted from RDKit (Landrum, 2010). In analogy to the discussion in Section 6.1, we can visually inspect the generated molecules with respect to the target properties. While the majority of the shown molecules contain the specified number of rings, a minority has one ring more or less than requested. Note that this small impurity is already reflected in the property-histograms shown in Figure 8a-h and could be counteracted by increasing the guidance strength. In Figure 9, we observe a transition from molecules generated with primarily polar bonds (*e.g.*, carbon-oxygen, carbon-nitrogen, or oxygen-hydrogen) to primarily non-polar (*e.g.*, carbon-carbon) bonds as we increase $\text{LogP}^\star$, which is indeed what we observed (Figure 2g,h). This observation is in agreement with the intuition that the lipophilicity of a molecule tends to be inversely related to the polarity of its bonds.

### F.2.6 COMPARING DISCRETE GUIDANCE TO DiGRESS

As discussed in Appendix E, DiGress has been developed by Vignac et al. (2023) as framework to guide discrete-time discrete-space diffusion models for graph generation. Here, we compare our method Discrete Guidance to DiGress. For this comparison, we use DiGress to guide a discrete-time discrete-space diffusion model with two discrete-time prediction models with the same architecture (and trained in the same way) as their continuous-time counterparts described in Appendix F.2.2. For this comparison, we generated 1,000 molecules guided with the different frameworks—DiGress and our method Discrete Guidance with exact as well as Taylor-approximated guidance (TAG)—towards specified property values in the number of rings ($N_r^\star$) and lipophilicity ($\text{LogP}^\star$). As a metric for the performance of this guided generation, we chose the mean absolute error (MAE) between the specified and RDKit-determined (Landrum, 2010) property values of the generated molecules and used its standard error as measure of uncertainty. Moreover, we determined the statistical significance of the difference in central tendency of the distribution of molecular properties between frameworks with a two-sided Mann-Whitney U test (Mann & Whitney, 1947) using its implementation in the *stats* module of the *scipy* python package.[7]

We find that Discrete Guidance achieves statistically significantly better results across a wide range of six different specified property values ($N_r^\star \in \{1, 3, 5\}$ and $\text{LogP} \in \{-2, 4, 10\}$) and for three different guidance strengths $\gamma \in \{0.2, 1, 2\}$ (Figure 10). The exact version (DG-exact) of Discrete Guidance performed either (statistically significant) better than or equally well as the Taylor-approximated version (DG-TAG) across these specified property and guidance strengths. From these results, we believe that there are two factors at play: (1) sampling with flow models (using Discrete Guidance) allows the unmasked tokens to be corrected at later stages of the denoising process, which could be advantageous for guided generation of SMILES strings. (2) the approximation error of the Taylor-approximation might guide generation towards wrong property values in certain scenarios.

---

[7]https://docs.scipy.org/doc/scipy/reference/generated/scipy.stats.mannwhitneyu.html

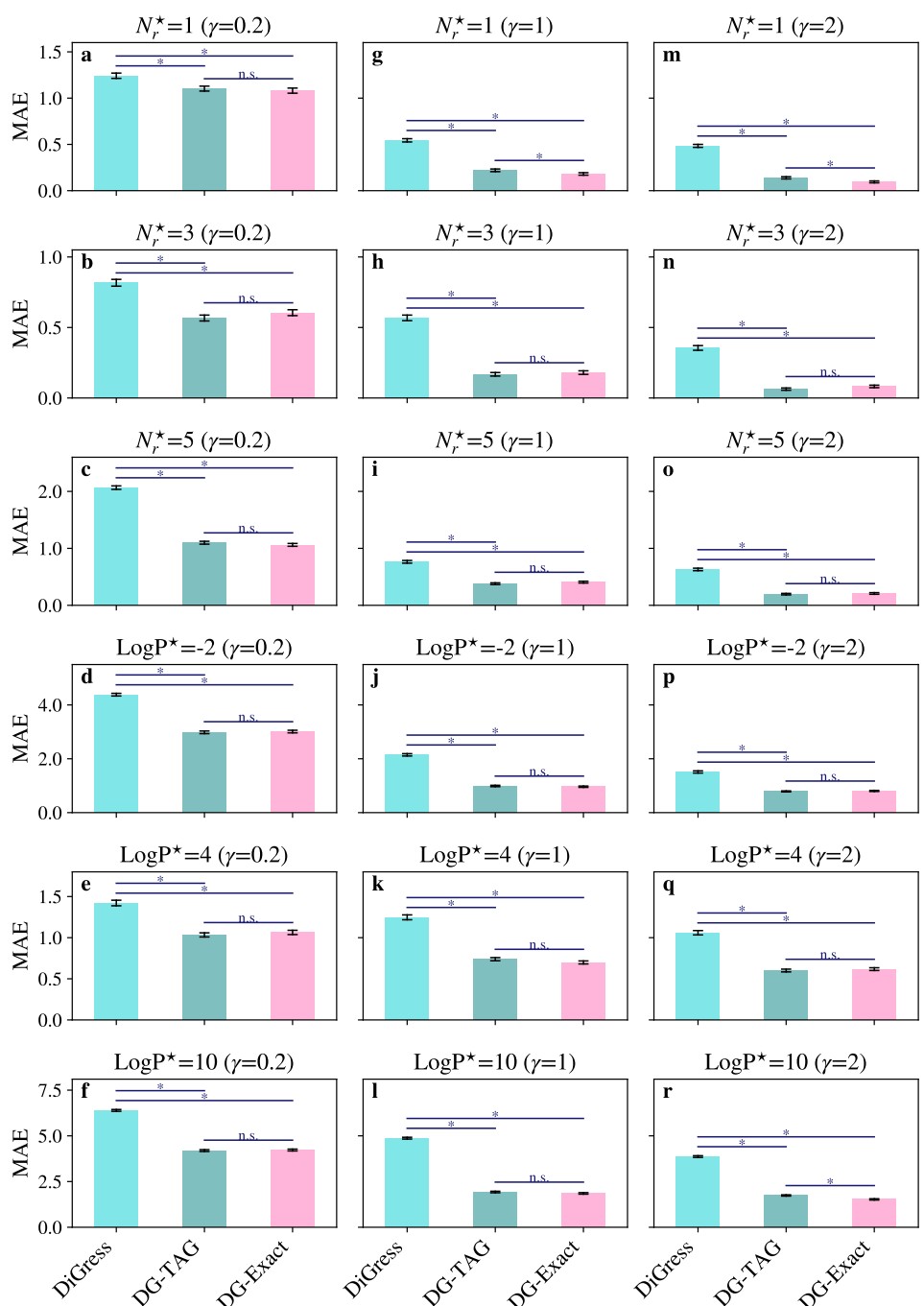

Figure 10: Mean absolute error (MAE) between specified target properties (number of rings $N_r^\star$ and lipophilicity LogP$^\star$), and properties of molecules generated by guiding to these values. DiGress was used to guide a discrete-time discrete diffusion model. DG-Exact and DG-TAG correspond to a continuous-time masking discrete flow-matching model guided applying Discrete Guidance (DG) with exact guidance and Taylor approximated guidance (TAG), respectively. Guidance has been performed at a guidance strength of $\gamma = 0.2$ (**a-f**), $\gamma = 1$ (**g-l**), and $\gamma = 2$ (**m-r**). Bar heights and error bars correspond to MAEs and their standard error, respectively, evaluated over 1,000 generated molecules. An asterisk ($\ast$) denotes statistical significance difference in MAE between the two models under the start and end point of the lines below it (two-sided Mann-Whitney U test, $p < 0.05$), while n.s. denotes lack of statistical significance.

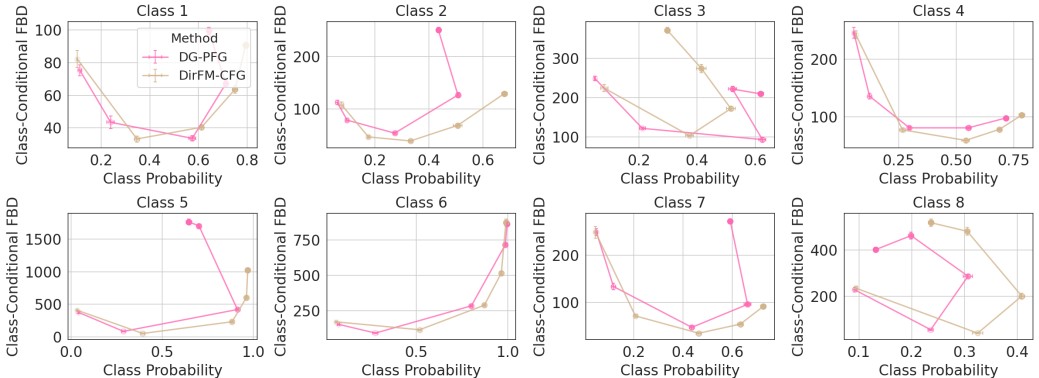

Figure 11: Classifier-free guided cell type conditioned enhancer generation on eight cell type classes. For the class-conditional FBD conditioned on the target class, lower is better, while for the average target Class Probability, higher is better. Each point in each panel arises from 1,000 sampled sequences for one guidance strength. For DirFM-CFG, we used the same guidance factors used in that work, namely $\gamma \in \{1, 3, 6, 10, 20\}$. DG-PFG used guidance strength $\gamma \in \{1, 2, 5, 10, 20\}$. Mean and standard deviation of both metrics were estimated over five bootstrap replicates (the standard deviations are small and are not readily discernible). For each method, guidance strengths increase from the left to the right in each panel.

## F.3 Enhancer DNA Design

In this section, we include the setup and additional results of the enhancer DNA experiment, including a description of the usage of Dirichlet FM (Appendix F.3.1), model architectures and training procedures (Appendix F.3.2), sampling procedures (Appendix F.3.3), additional results on predictor guidance (Appendix F.3.4) and results on predictor-free guidance (Appendix F.3.5).

### F.3.1 Dirichlet FM

For Dirichlet FM with both classifier guidance (DirFM-CG) and classifier-free guidance (DirFM-CFG), we used the model checkpoints provided in the official library.[8] We built on their codebase to evaluate the class-conditional FBD and target class probability, using the same *oracle* classifier checkpoint provided from the library for evaluation. For each target class, at each guidance strength, we sampled 1,000 sequences for evaluation. We also experimented with sampling 2,000, 5,000 or 10,000 sequences and observed similar results for the class-conditional FBD and the average target class probability. We observed the variances of the class-conditional FBD and the average target class probability between samples produced with different random seeds to be small. We verified with the authors to ensure that the models were used and evaluated correctly. The reproduced results closely match those reported in Stärk et al. (2024). The small discrepancies are potentially due to: different sampling random seeds; different number of generated sequences were used for evaluation (we used 1,000 sequences, they used 10,000); different number of sequences from the training set were used as the reference set for calculating the class-conditional FBD (we used all sequences of the target class from the training set, whereas they first randomly sampled 10,000 sequences from the training set, then subsetted to those from the target class).

### F.3.2 Model Architecture and Training

For both the denoising model and the classifier model for Discrete Guidance, we used the same architectures as Dirichlet FM. The denoising model consists of 20 layers of 1D convolutions interleaved with time embedding layers, with hidden dimension size 128. The classifier model has a similar architecture with 5 layers of 1D convolutions, and the final representations are mean-pooled and fed into a 2-layer feed-forward network. For Discrete Guidance, we used similar training hyperparameters as Dirichlet FM. For the unconditional denoising model, we trained for 400 epochs and used FBD on the validation set for early stopping, with a learning rate of 0.0005 and 500 linear warmup steps. For

---

[8]https://github.com/HannesStark/dirichlet-flow-matching

the conditional denoising network used in PFG, we trained with a conditioning ratio (the fraction of times the model is trained with a class label as input instead of the no-class token as input) of 0.7 for 300 epochs and used validation loss for early stopping. For DiGress, we trained a discrete-time, discrete diffusion model using the same training hyperparameters as the flow-matching models used in Discrete Guidance, using 100 discrete time steps.

Since Stärk et al. (2024) perform guidance on the continuous space of the simplex, whereas both Discrete Guidance and DiGress work in the native, discrete DNA sequence space, the classifier models used for guidance for these approaches cannot be identical. Consequently, we used the classifier model of Stärk et al. (2024) for guidance of DirFM-CG, and we trained separate classifiers for Discrete Guidance and DiGress, but adopting the same architecture as the classifier model of Stärk et al. (2024) as detailed above. In all cases, these classifiers take in a time and a correspondingly *noised* sample, where the noise levels correspond to those used to train the paired unconditional model. Notably, the same oracle classifier was used for all methods for evaluation. In DirFM-CG, the classifier takes in a sample from the probability simplex, whereas for Discrete Guidance and DiGress, the classifier takes in a sequence that is partially masked. We initially observed a discrepancy between the noisy classifier and the clean classifier trained on un-noised data. To mitigate this, we found it helpful to train the noisy classifier using samples generated by the unconditional model and labeled by the clean classifier. We trained the noisy classifier for 400 epochs, using 1 million sequences sampled from the unconditional model and labeled by the clean classifier. For DiGress, we trained a noisy classifier with the same architecture on the same data under a discrete-time forward noising process. We used a learning rate of 0.001 with the Adam optimizer (Kingma & Ba, 2015) and a batch size of 256 for all models. We trained on one RTX 6000A GPU. Training each model for 400 epochs takes 5 hours.

### F.3.3 SAMPLING

For sampling from Discrete Guidance, we used Euler integration with a step size of $0.01$. We used a stochasticity level, $\eta$, of 0 in sampling as it leads to the best unconditional FBD. For Dirichlet FM, we used the same number of integration steps (100) as Stärk et al. (2024). For DiGress, sampling uses the same number of discrete time steps as in training, which is 100.

As for guidance strength, for Dirichlet FM, we used the same guidance strength used in Stärk et al. (2024): $\gamma \in \{1, 3, 6, 10, 20\}$. For DirFM-CG, we additionally used stronger guidance strengths, $\gamma \in \{50, 100, 200, 500, 1000, 2000\}$ in an attempt to improve its performance further from the results presented in Stärk et al. (2024). We increased the guidance strengths until DirFM-CG became unstable and unable to produce samples. For DG-PG, DG-PFG and DiGress, we used guidance strengths $\gamma \in \{1, 2, 5, 10, 20\}$.

### F.3.4 PREDICTOR GUIDANCE

We evaluated on a total of eight cell type class, including the three classes selected for evaluation in Stärk et al. (2024). These classes were randomly sampled among the classes where the oracle classifier has relatively good performances, as measured by AUROC and AUPR on the test set (Figure 13). The classes are labeled as $\{16, 5, 4, 2, 33, 68, 9, 12\}$ in the dataset. For predictor guidance, we observed that DG-Exact, DG-TAG and DiGress perform comparably across the cell type classes, with each exhibiting slight advantage over others on specific classes and metrics, while outperforming DirFM-CG (Figure 3). We also note that in practice, one metric that might be informative is the proportion of generated samples that has the target class as the most likely predicted class, along with the diversity of these samples. We observed that DG-Exact, DG-TAG and DiGress also perform comparably under these metrics (Figure 12).

### F.3.5 PREDICTOR-FREE GUIDANCE

We compared Discrete Guidance with predictor-free guidance (DG-PFG) with Dirichlet FM with classifier-free guidance (DirFM-CFG) on the same eight cell type classes as in the predictor guidance evaluation (Figure 11). On the whole, we observed that both methods perform comparably, with each showing slight advantages for different cell types and metrics. In some cases, we observed both metrics to worsen at high guidance strengths for both DG-PFG and DirFM-CFG. We hypothesize that this behavior may be due to the generative models focusing on regions of the sequence space

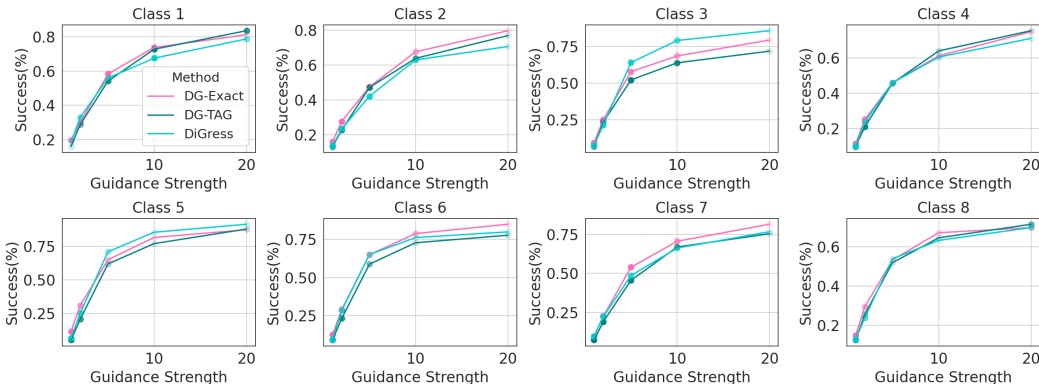

Figure 12: Alternative metrics on classifier-guided cell type conditioned enhancer generation on eight cell type classes. We defined a sampled sequence as a *success* if the oracle classifier predicts the target class as the most likely class for the sequence. For each method, we sampled 1,000 sequences with increasing guidance strength and calculated the percentage of successful samples. DG-Exact, DG-TAG and DiGress all used guidance strength $\gamma \in \{1, 2, 5, 10, 20\}$. Intensity of color of the circle markers indicate the average pairwise hamming distance of the sampled sequences, where darker is higher diversity. All methods have average pairwise hamming distances between 340 and 380 for all classes and guidance factors.

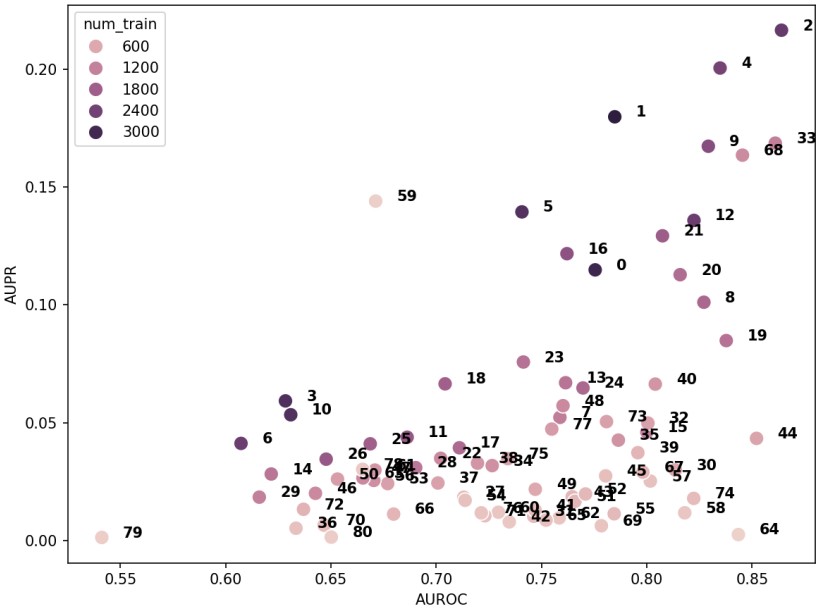

Figure 13: Test set performance of the oracle classifier used for enhancer generation evaluation. The oracle is the same as the one used in Stärk et al. (2024). Each class is colored by the number of samples in the training set. Per-class AUROC and AUPR are computed with the one-vs-rest approach, where each class is iteratively treated as the positive class and all other classes are treated as the negative.

that deviate from the data distribution. Alternatively, this could be due to the model having difficulty sampling from the correct temperature-annealed distributions. We leave investigations of this issue for future research.

Table 3: Recovery on the PDB test set provided by Dauparas et al. (2022).

| Method | Recovery |
|---|---|
| ProteinMPNN ($T = 0.5$) | 45.37% |
| ProteinMPNN ($T = 0.2$) | 47.56% |
| ProteinMPNN ($T = 0.1$) | 47.92% |
| ProteinMPNN ($T = 0.01$) | 47.99% |
| FMIF ($T = 0.1$) | **49.22%** |

## F.4 STABILITY-CONDITIONED PROTEIN GENERATIVE MODELS

### F.4.1 FMIF TRAINING

We trained a flow-based inverse folding closely based on the ProteinMPNN architecture (Dauparas et al., 2022). Specifically, we used the ProteinMPNN architecture with 4 encoder and 4 decoder layers, and we removed the autoregressive mask. We trained with a backbone noise of 0.1 using 30 nearest-neighbors to construct the graph and kept all other hyperparameters the same as the ProteinMPNN architecture. FMIF was trained using the flow-matching training objective (Campbell et al., 2024) with a masking noise process. The model was trained using the training set of the PDB dataset proposed in Dauparas et al. (2022). The model was trained for 400 epochs with the Adam optimizer and a learning rate of 0.001 (Kingma & Ba, 2015). To sample from the trained model we used Euler integration with a step size of 0.01 and a stochasticity level, $\eta$, of 1. To compare our model with ProteinMPNN, we re-trained ProteinMPNN with the same amount of backbone noise using their provided training scripts. Table 3 shows the sequence recovery on the PDB test set provided by Dauparas et al. (2022). FMIF shows improved recovery over ProteinMPNN. All training was done on one RTX 6000A GPU.

### F.4.2 STABILITY PREDICTION

The regression model used to predict $\Delta\Delta G$ given a protein sequence and structure was built using a similar architecture to FMIF, but with a few modifications. Specifically, the final layer is mean-pooled and then mapped with an MLP consisting of a $128 \rightarrow 128$ linear layer followed by a ReLU activation function before being mapped with a linear layer to a single scalar.

We trained our model using the data provided by Tsuboyama et al. (2023), specifically the file (Tsuboyama2023_Dataset2_Dataset3_20230416.csv). We pre-processed the data to remove any sequences that had a difference greater than 0.5 in the average $\Delta G$ measurement between replicates. We also mapped all sequences with measurements that were outside the dynamic range of the experiment ($> 5$ and $< 1$) to the nearest measurable value (5 or 1). We created a validation set by clustering all of the data based on the wild-type using the WT_cluster column. During training we pass every sequence through our stability prediction model along with their corresponding wild-type sequences. To force the model to learn $\Delta\Delta G$, a mean-squared error loss is computed from the difference in predicted folding free-energies between a sequence and the wild-type and the true difference in folding free-energies between the sequence and the wild-type. Models were trained using AdamW with a weight-decay of 0.0001 and a learning rate of 0.0001. We observed much better generalization on the validation set if the subset of the model weights before the mean-pooling was initialized at the corresponding FMIF model weights. During training we used the average Spearman's correlation among each validation cluster to decide the optimal number of epochs. Our best model had an average Spearman's correlation of 0.80, demonstrating an ability to generalize to novel structures and sequences (Figure 14). The final stability model was then trained on the full dataset. All models were trained on a single RTX 6000A GPU.

### F.4.3 STABILITY GUIDANCE

To perform guidance, for each of the evaluated proteins, we created a protein-specific noisy classifier to predict $p(\Delta\Delta G \geq 0 \,|\, \mathbf{x_t}, \mathbf{c})$. The classifier was constructed using the same architecture as the stability regression model but with a final sigmoid activation function. Model weights were initialized at the weights of the stability regression model. The model was trained for 30 epochs using only the data for the particular protein that we wish to perform guidance on. The model was trained using

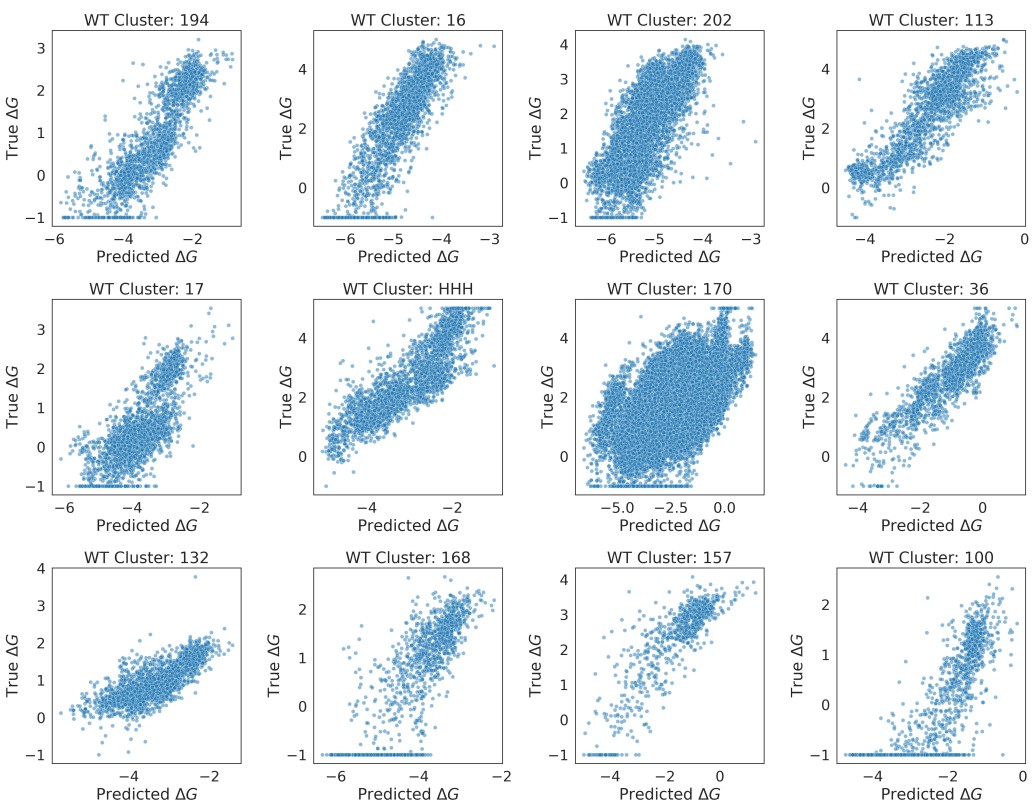

Figure 14: True versus predicted $\Delta G$ predictions for 12 clusters from the validation set.

a negative log-likelihood loss and the Adam optimizer with a learning rate of $0.0001$. We initially observed a large discrepancy between the noisy classifier and the actual clean stability regression model. To help mitigate this, we found it helpful to further train the noisy classifier on samples generated from the guided model and labeled with the clean regression model. Specifically, we generated 1,000 new sequences by sampling using FMIF-DG-TAG with a guide strength of $0.1$. Each of these sequences was then labeled with the clean regression model and added to the original training set of labeled sequences. The noisy classifier was then trained for an additional 30 epochs. Guidance was performed using the approximate guide-adjusted rates. As with the non-guided FMIF model, we used Euler integration with a step size of $0.01$ and a stochasticity level of $1$.

### F.4.4 BASELINES

To evaluate DiGress, we first trained a discrete-time discrete diffusion inverse folding model. This model was trained using the same architecture and training hyperparameters as FMIF. Consistent with the number of euler-integration steps used for sampling from FMIF and FMIF-DG, we used $100$ discrete time points for training the DiGress diffusion model. The noisy classifier also used the same architecture and training procedure as was used for FMIF-DG-Exact and FMIF-DG-TAG.

To generate sequences from ProteinMPNN we used the generation scripts provided in the Protein-MPNN GitHub repository.[9] We used the pre-trained weights for the model trained with a backbone noise of $0.1$ (the same noise used in FMIF) provided in the GitHub repository.

### F.4.5 METRICS

All generated sequences were folded using a local version of ColabFold (Mirdita et al., 2022; Jumper et al., 2021). To mitigate any bias, we use the *sequence only* mode which does not use structural templates or multiple sequence alignments. We defined success as sequences that were generated with RMSD $\leq 2$Å and $\Delta\Delta G \geq 0$. Diversity was computed as the average pairwise Hamming distance among the generated sequences.

---

[9]https://github.com/dauparas/ProteinMPNN

### F.4.6 DETAILED RESULTS

Below, we show detailed results for each of the eight proteins that we evaluated (Tables 4-11).

Table 4: Inverse-folding guided results for protein 5KPH evaluated by protein stability, folding into correct structure, and diversity. We defined success as sequences that were generated with RMSD $\leq 2\text{Å}$ and $\Delta\Delta G \geq 0$. Diversity was computed as the average pairwise Hamming distance among the generated sequences. $P_\theta(\mathbf{x} \mid \mathbf{c})$ Temp is the temperature of the baseline structure-conditioned model without stability guidance, while parenthetical $\gamma$ denotes the guidance strength. ProteinMPNN performs inverse-folding without any stability awareness, whereas FMIF refers to our own unguided inverse folding model without any stability awareness. DiGress preforms (approximate) predictor-guidance on a discrete-time diffusion inverse folding model. FMIF-DG-TAG refers to our TAG-approximated predictor-guidance, whereas FMIF-DG-Exact refers to our exact predictor-guidance, both applied to FMIF.

| Method | $P_\theta(\mathbf{x} \mid \mathbf{c})$ Temp | $\Delta\Delta G > 0$ % ($\uparrow$) | RMSD $\leq 2\text{Å}$ % ($\uparrow$) | Diversity ($\uparrow$) | Success ($\uparrow$) |
|---|---|---|---|---|---|
| ProteinMPNN | 1.0 | 0% | 47% | 53.07 | 0% |
|  | 0.1 | 1% | 100% | 27.98 | 1% |
|  | 0.01 | 0% | 100% | 27.42 | 0% |
| FMIF | 1.0 | 0% | 75% | 50.29 | 0% |
|  | 0.1 | 2% | 100% | 23.13 | 2% |
|  | 0.01 | 4% | 100% | 21.82 | 4% |
| DiGress ($\gamma = 1$) | 0.1 | 0% | 100% | 18.65 | 0% |
| DiGress ($\gamma = 10$) | 0.1 | 0% | 100% | 16.76 | 0% |
| DiGress ($\gamma = 100$) | 0.1 | 0% | 100% | 22.54 | 0% |
| FMIF-DG-TAG ($\gamma = 1$) | 0.1 | 9% | 100% | 24.15 | 9% |
| FMIF-DG-TAG ($\gamma = 10$) | 0.1 | 52% | 100% | 23.26 | 52% |
| FMIF-DG-TAG ($\gamma = 100$) | 0.1 | 94% | 100% | 20.82 | 94% |
| FMIF-DG-Exact ($\gamma = 1$) | 0.1 | 10% | 100% | 24.66 | 10% |
| FMIF-DG-Exact ($\gamma = 10$) | 0.1 | 58% | 100% | 25.78 | 58% |
| FMIF-DG-Exact ($\gamma = 100$) | 0.1 | 100% | 100% | 19.72 | 100% |

Table 5: Inverse-folding guided results for protein 1F0M evaluated by protein stability, folding into correct structure, and diversity. We defined success as sequences that were generated with RMSD $\leq 2\text{Å}$ and $\Delta\Delta G \geq 0$. Diversity was computed as the average pairwise Hamming distance among the generated sequences. $P_\theta(\mathbf{x} \mid \mathbf{c})$ Temp is the temperature of the baseline structure-conditioned model without stability guidance, while parenthetical $\gamma$ denotes the guidance strength. ProteinMPNN performs inverse-folding without any stability awareness, whereas FMIF refers to our own unguided inverse folding model without any stability awareness. DiGress preforms (approximate) predictor-guidance on a discrete-time diffusion inverse folding model. FMIF-DG-TAG refers to our TAG-approximated predictor-guidance, whereas FMIF-DG-Exact refers to our exact predictor-guidance, both applied to FMIF.

| Method | $P_\theta(\mathbf{x} \mid \mathbf{c})$ Temp | $\Delta\Delta G > 0$ % ($\uparrow$) | RMSD $\leq 2\text{Å}$ % ($\uparrow$) | Diversity ($\uparrow$) | Success ($\uparrow$) |
|---|---|---|---|---|---|
| ProteinMPNN | 1.0 | 0% | 15% | 45.17 | 0% |
|  | 0.1 | 14% | 95% | 17.70 | 14% |
|  | 0.01 | 18% | 97% | 16.53 | 17% |
| FMIF | 1.0 | 0% | 24% | 44.95 | 0% |
|  | 0.1 | 77% | 100% | 12.44 | 77% |
|  | 0.01 | 77% | 100% | 10.30 | 77% |
| DiGress ($\gamma = 1$) | 0.1 | 36% | 95% | 12.40 | 34% |
| DiGress ($\gamma = 10$) | 0.1 | 66% | 100% | 11.11 | 66% |
| DiGress ($\gamma = 100$) | 0.1 | 71% | 100% | 11.81 | 71% |
| FMIF-DG-TAG ($\gamma = 1$) | 0.1 | 79% | 100% | 12.19 | 79% |
| FMIF-DG-TAG ($\gamma = 10$) | 0.1 | 96% | 100% | 11.91 | 96% |
| FMIF-DG-TAG ($\gamma = 100$) | 0.1 | 69% | 100% | 8.98 | 69% |
| FMIF-DG-Exact ($\gamma = 1$) | 0.1 | 74% | 100% | 12.53 | 74% |
| FMIF-DG-Exact ($\gamma = 10$) | 0.1 | 97% | 100% | 12.36 | 97% |
| FMIF-DG-Exact ($\gamma = 100$) | 0.1 | 100% | 100% | 10.13 | 100% |

Table 6: Inverse-folding guided results for protein 6M3N evaluated by protein stability, folding into correct structure, and diversity. We defined success as sequences that were generated with RMSD $\leq 2$Å and $\Delta\Delta G \geq 0$. Diversity was computed as the average pairwise Hamming distance among the generated sequences. $P_\theta(\mathbf{x}\,|\,\mathbf{c})$ Temp is the temperature of the baseline structure-conditioned model without stability guidance, while parenthetical $\gamma$ denotes the guidance strength. ProteinMPNN performs inverse-folding without any stability awareness, whereas FMIF refers to our own unguided inverse folding model without any stability awareness. DiGress preforms (approximate) predictor-guidance on a discrete-time diffusion inverse folding model. FMIF-DG-TAG refers to our TAG-approximated predictor-guidance, whereas FMIF-DG-Exact refers to our exact predictor-guidance, both applied to FMIF.

| Method | $P_\theta(\mathbf{x}\,|\,\mathbf{c})$ Temp | $\Delta\Delta G > 0$ % ($\uparrow$) | RMSD $\leq 2$Å % ($\uparrow$) | Diversity ($\uparrow$) | Success ($\uparrow$) |
|---|---|---|---|---|---|
| ProteinMPNN | 1.0 | 0% | 9% | 41.03 | 0% |
| | 0.1 | 92% | 99% | 14.04 | 91% |
| | 0.01 | 90% | 99% | 13.66 | 89% |
| FMIF | 1.0 | 9% | 30% | 39.97 | 7% |
| | 0.1 | 100% | 99% | 12.16 | 99% |
| | 0.01 | 100% | 99% | 12.48 | 99% |
| DiGress ($\gamma = 1$) | 0.1 | 97% | 94% | 8.93 | 91% |
| DiGress ($\gamma = 10$) | 0.1 | 100% | 100% | 8.41 | 100% |
| DiGress ($\gamma = 100$) | 0.1 | 100% | 99% | 9.98 | 99% |
| FMIF-DG-TAG ($\gamma = 1$) | 0.1 | 100% | 100% | 12.43 | 100% |
| FMIF-DG-TAG ($\gamma = 10$) | 0.1 | 100% | 99% | 13.01 | 99% |
| FMIF-DG-TAG ($\gamma = 100$) | 0.1 | 100% | 100% | 12.11 | 100% |
| FMIF-DG-Exact ($\gamma = 1$) | 0.1 | 100% | 97% | 12.29 | 97% |
| FMIF-DG-Exact ($\gamma = 10$) | 0.1 | 100% | 99% | 11.96 | 99% |
| FMIF-DG-Exact ($\gamma = 100$) | 0.1 | 99% | 43% | 25.48 | 42% |

Table 7: Inverse-folding guided results for protein 1O6X evaluated by protein stability, folding into correct structure, and diversity. We defined success as sequences that were generated with RMSD $\leq 2$Å and $\Delta\Delta G \geq 0$. Diversity was computed as the average pairwise Hamming distance among the generated sequences. $P_\theta(\mathbf{x}\,|\,\mathbf{c})$ Temp is the temperature of the baseline structure-conditioned model without stability guidance, while parenthetical $\gamma$ denotes the guidance strength. ProteinMPNN performs inverse-folding without any stability awareness, whereas FMIF refers to our own unguided inverse folding model without any stability awareness. DiGress preforms (approximate) predictor-guidance on a discrete-time diffusion inverse folding model. FMIF-DG-TAG refers to our TAG-approximated predictor-guidance, whereas FMIF-DG-Exact refers to our exact predictor-guidance, both applied to FMIF.

| Method | $P_\theta(\mathbf{x}\,|\,\mathbf{c})$ Temp | $\Delta\Delta G > 0$ % ($\uparrow$) | RMSD $\leq 2$Å % ($\uparrow$) | Diversity ($\uparrow$) | Success ($\uparrow$) |
|---|---|---|---|---|---|
| ProteinMPNN | 1.0 | 0% | 26% | 49.47 | 0% |
| | 0.1 | 28% | 90% | 24.33 | 24% |
| | 0.01 | 31% | 85% | 23.96 | 27% |
| FMIF | 1.0 | 0% | 34% | 47.66 | 0% |
| | 0.1 | 56% | 98% | 18.60 | 54% |
| | 0.01 | 48% | 99% | 16.82 | 47% |
| DiGress ($\gamma = 1$) | 0.1 | 1% | 100% | 14.47 | 1% |
| DiGress ($\gamma = 10$) | 0.1 | 0% | 100% | 11.49 | 0% |
| DiGress ($\gamma = 100$) | 0.1 | 16% | 100% | 14.84 | 16% |
| FMIF-DG-TAG ($\gamma = 1$) | 0.1 | 61% | 100% | 17.58 | 61% |
| FMIF-DG-TAG ($\gamma = 10$) | 0.1 | 83% | 99% | 18.47 | 83% |
| FMIF-DG-TAG ($\gamma = 100$) | 0.1 | 96% | 97% | 15.34 | 93% |
| FMIF-DG-Exact ($\gamma = 1$) | 0.1 | 66% | 98% | 19.30 | 64% |
| FMIF-DG-Exact ($\gamma = 10$) | 0.1 | 81% | 100% | 16.94 | 81% |
| FMIF-DG-Exact ($\gamma = 100$) | 0.1 | 99% | 98% | 20.97 | 97% |

Table 8: Inverse-folding guided results for protein 2JT1 evaluated by protein stability, folding into correct structure, and diversity. We defined success as sequences that were generated with RMSD $\leq 2$Å and $\Delta\Delta G \geq 0$. Diversity was computed as the average pairwise Hamming distance among the generated sequences. $P_\theta(\mathbf{x} \mid \mathbf{c})$ Temp is the temperature of the baseline structure-conditioned model without stability guidance, while parenthetical $\gamma$ denotes the guidance strength. ProteinMPNN performs inverse-folding without any stability awareness, whereas FMIF refers to our own unguided inverse folding model without any stability awareness. DiGress preforms (approximate) predictor-guidance on a discrete-time diffusion inverse folding model. FMIF-DG-TAG refers to our TAG-approximated predictor-guidance, whereas FMIF-DG-Exact refers to our exact predictor-guidance, both applied to FMIF.

| Method | $P_\theta(\mathbf{x} \mid \mathbf{c})$ Temp | $\Delta\Delta G > 0$ % ($\uparrow$) | RMSD $\leq 2$Å % ($\uparrow$) | Diversity ($\uparrow$) | Success ($\uparrow$) |
|---|---|---|---|---|---|
| ProteinMPNN | 1.0 | 0% | 13% | 47.15 | 0% |
| | 0.1 | 8% | 88% | 17.29 | 6% |
| | 0.01 | 6% | 81% | 17.31 | 2% |
| FMIF | 1.0 | 2% | 14% | 45.11 | 1% |
| | 0.1 | 14% | 88% | 14.33 | 13% |
| | 0.01 | 10% | 89% | 13.27 | 9% |
| DiGress ($\gamma = 1$) | 0.1 | 11% | 96% | 16.23 | 11% |
| DiGress ($\gamma = 10$) | 0.1 | 14% | 98% | 14.17 | 14% |
| DiGress ($\gamma = 100$) | 0.1 | 46% | 100% | 12.58 | 46% |
| FMIF-DG-TAG ($\gamma = 1$) | 0.1 | 16% | 92% | 14.58 | 16% |
| FMIF-DG-TAG ($\gamma = 10$) | 0.1 | 26% | 89% | 11.98 | 21% |
| FMIF-DG-TAG ($\gamma = 100$) | 0.1 | 85% | 75% | 13.30 | 63% |
| FMIF-DG-Exact ($\gamma = 1$) | 0.1 | 21% | 90% | 14.26 | 20% |
| FMIF-DG-Exact ($\gamma = 10$) | 0.1 | 26% | 97% | 10.51 | 26% |
| FMIF-DG-Exact ($\gamma = 100$) | 0.1 | 91% | 77% | 17.31 | 68% |

Table 9: Inverse-folding guided results for protein 2JN4 evaluated by protein stability, folding into correct structure, and diversity. We defined success as sequences that were generated with RMSD $\leq 2$Å and $\Delta\Delta G \geq 0$. Diversity was computed as the average pairwise Hamming distance among the generated sequences. $P_\theta(\mathbf{x} \mid \mathbf{c})$ Temp is the temperature of the baseline structure-conditioned model without stability guidance, while parenthetical $\gamma$ denotes the guidance strength. ProteinMPNN performs inverse-folding without any stability awareness, whereas FMIF refers to our own unguided inverse folding model without any stability awareness. DiGress preforms (approximate) predictor-guidance on a discrete-time diffusion inverse folding model. FMIF-DG-TAG refers to our TAG-approximated predictor-guidance, whereas FMIF-DG-Exact refers to our exact predictor-guidance, both applied to FMIF.

| Method | $P_\theta(\mathbf{x} \mid \mathbf{c})$ Temp | $\Delta\Delta G > 0$ % ($\uparrow$) | RMSD $\leq 2$Å % ($\uparrow$) | Diversity ($\uparrow$) | Success ($\uparrow$) |
|---|---|---|---|---|---|
| ProteinMPNN | 1.0 | 0% | 1% | 41.87 | 0% |
| | 0.1 | 0% | 32% | 15.77 | 0% |
| | 0.01 | 1% | 36% | 14.76 | 0% |
| FMIF | 1.0 | 0% | 0% | 40.44 | 0% |
| | 0.1 | 20% | 40% | 14.06 | 8% |
| | 0.01 | 15% | 32% | 13.03 | 6% |
| DiGress ($\gamma = 1$) | 0.1 | 0% | 56% | 11.98 | 0% |
| DiGress ($\gamma = 10$) | 0.1 | 9% | 80% | 10.94 | 8% |
| DiGress ($\gamma = 100$) | 0.1 | 4% | 56% | 11.84 | 4% |
| FMIF-DG-TAG ($\gamma = 1$) | 0.1 | 20% | 29% | 13.21 | 8% |
| FMIF-DG-TAG ($\gamma = 10$) | 0.1 | 45% | 26% | 10.06 | 17% |
| FMIF-DG-TAG ($\gamma = 100$) | 0.1 | 20% | 21% | 10.28 | 6% |
| FMIF-DG-Exact ($\gamma = 1$) | 0.1 | 21% | 35% | 13.18 | 10% |
| FMIF-DG-Exact ($\gamma = 10$) | 0.1 | 60% | 69% | 10.06 | 50% |
| FMIF-DG-Exact ($\gamma = 100$) | 0.1 | 78% | 39% | 6.64 | 28% |

Table 10: Inverse-folding guided results for protein 6ACV evaluated by protein stability, folding into correct structure, and diversity. We defined success as sequences that were generated with RMSD $\leq 2\text{Å}$ and $\Delta\Delta G \geq 0$. Diversity was computed as the average pairwise Hamming distance among the generated sequences. $P_\theta(\mathbf{x} \,|\, \mathbf{c})$ Temp is the temperature of the baseline structure-conditioned model without stability guidance, while parenthetical $\gamma$ denotes the guidance strength. ProteinMPNN performs inverse-folding without any stability awareness, whereas FMIF refers to our own unguided inverse folding model without any stability awareness. DiGress preforms (approximate) predictor-guidance on a discrete-time diffusion inverse folding model. FMIF-DG-TAG refers to our TAG-approximated predictor-guidance, whereas FMIF-DG-Exact refers to our exact predictor-guidance, both applied to FMIF.

| Method | $P_\theta(\mathbf{x} \,|\, \mathbf{c})$ Temp | $\Delta\Delta G > 0$ % ($\uparrow$) | RMSD $\leq 2\text{Å}$ % ($\uparrow$) | Diversity ($\uparrow$) | Success ($\uparrow$) |
|---|---|---|---|---|---|
| ProteinMPNN | 1.0 | 0% | 1% | 44.73 | 0% |
|  | 0.1 | 2% | 12% | 15.79 | 2% |
|  | 0.01 | 2% | 14% | 14.55 | 1% |
| FMIF | 1.0 | 0% | 0% | 45.08 | 0% |
|  | 0.1 | 10% | 36% | 14.01 | 6% |
|  | 0.01 | 18% | 45% | 13.67 | 13% |
| DiGress ($\gamma = 1$) | 0.1 | 0% | 17% | 15.74 | 0% |
| DiGress ($\gamma = 10$) | 0.1 | 0% | 5% | 13.18 | 0% |
| DiGress ($\gamma = 100$) | 0.1 | 45% | 30% | 14.53 | 20% |
| FMIF-DG-TAG ($\gamma = 1$) | 0.1 | 28% | 43% | 13.27 | 15% |
| FMIF-DG-TAG ($\gamma = 10$) | 0.1 | 84% | 59% | 9.12 | 54% |
| FMIF-DG-TAG ($\gamma = 100$) | 0.1 | 97% | 75% | 6.66 | 74% |
| FMIF-DG-Exact ($\gamma = 1$) | 0.1 | 26% | 39% | 13.97 | 15% |
| FMIF-DG-Exact ($\gamma = 10$) | 0.1 | 71% | 55% | 9.93 | 44% |
| FMIF-DG-Exact ($\gamma = 100$) | 0.1 | 69% | 60% | 9.46 | 46% |

Table 11: Inverse-folding guided results for protein 2K5N evaluated by protein stability, folding into correct structure, and diversity. We defined success as sequences that were generated with RMSD $\leq 2\text{Å}$ and $\Delta\Delta G \geq 0$. Diversity was computed as the average pairwise Hamming distance among the generated sequences. $P_\theta(\mathbf{x} \,|\, \mathbf{c})$ Temp is the temperature of the baseline structure-conditioned model without stability guidance, while parenthetical $\gamma$ denotes the guidance strength. ProteinMPNN performs inverse-folding without any stability awareness, whereas FMIF refers to our own unguided inverse folding model without any stability awareness. DiGress preforms (approximate) predictor-guidance on a discrete-time diffusion inverse folding model. FMIF-DG-TAG refers to our TAG-approximated predictor-guidance, whereas FMIF-DG-Exact refers to our exact predictor-guidance, both applied to FMIF.

| Method | $P_\theta(\mathbf{x} \,|\, \mathbf{c})$ Temp | $\Delta\Delta G > 0$ % ($\uparrow$) | RMSD $\leq 2\text{Å}$ % ($\uparrow$) | Diversity ($\uparrow$) | Success ($\uparrow$) |
|---|---|---|---|---|---|
| ProteinMPNN | 1.0 | 0% | 3% | 47.98 | 0% |
|  | 0.1 | 0% | 63% | 25.26 | 0% |
|  | 0.01 | 0% | 70% | 24.90 | 0% |
| FMIF | 1.0 | 0% | 6% | 45.55 | 0% |
|  | 0.1 | 3% | 76% | 14.89 | 2% |
|  | 0.01 | 1% | 72% | 13.64 | 0% |
| DiGress ($\gamma = 1$) | 0.1 | 0% | 94% | 14.73 | 0% |
| DiGress ($\gamma = 10$) | 0.1 | 0% | 99% | 12.38 | 0% |
| DiGress ($\gamma = 100$) | 0.1 | 0% | 47% | 14.13 | 0% |
| FMIF-DG-TAG ($\gamma = 1$) | 0.1 | 7% | 85% | 14.84 | 6% |
| FMIF-DG-TAG ($\gamma = 10$) | 0.1 | 9% | 82% | 13.31 | 7% |
| FMIF-DG-TAG ($\gamma = 100$) | 0.1 | 12% | 70% | 10.26 | 8% |
| FMIF-DG-Exact ($\gamma = 1$) | 0.1 | 1% | 77% | 14.83 | 1% |
| FMIF-DG-Exact ($\gamma = 10$) | 0.1 | 9% | 82% | 13.54 | 8% |
| FMIF-DG-Exact ($\gamma = 100$) | 0.1 | 36% | 31% | 12.62 | 13% |

