# OpenReview forum: "Unlocking Guidance for Discrete State-Space Diffusion and Flow Models"
_ICLR.cc/2025/Conference — ICLR 2025 Poster_

### Official Review · Reviewer_MZ28 · 2024-10-30

**Soundness:** 3
**Presentation:** 3
**Contribution:** 3
**Rating:** 8
**Confidence:** 4

**Summary:**

The paper presents a method for guiding diffusion models and flow models on discrete spaces. The method is constructed using continuous time Markov chain models to represent the discrete changes with jumps. A guidance scheme can be constructed by guiding the rate of the time-continuous process. The model is evaluated with experiments on small molecules, DNA and protein sequences.

**Strengths:**

- the paper targets an important and difficult problem
- the use of time continuous processes to enable guidance seems well-founded
- the paper is nicely presented
- the method is evaluated on relevant datasets

**Weaknesses:**

- while the method is presented as working for both diffusion and flow models, I believe the experiments are only carried out using flow matching (as stated on line 297)

**Questions:**

- you write that the probability of jumps in two or more dimensions simultaneously is zero. But how does it work in the numerical setting where the continuous time is discretized into finite time intervals, e.g. in the Euler integration? I guess here two dimensions can jump simultaneously. Does this cause problems?
- can you comment more on the issue with diffusion models? (that you only use flow matching in the experiments) Is this a weakness of the model

---

> ### Author Response · Authors · 2024-11-16
> **Response**
>
> Thank you for your thoughtful comments and questions. We address them below.
>
> > while the method is presented as working for both diffusion and flow models, I believe the experiments are only carried out using flow matching (as stated on line 297)
>
> Indeed, our approach is general and works for both discrete state-space diffusion and flow models built on CTMCs. It was no doubt easy to miss, but to demonstrate that Discrete Guidance readily applies to diffusion models, we also guided a pre-trained discrete diffusion model to produce class-conditional images (Appendix F.1). As we were focusing our manuscript on scientific examples, we relegated this to the Appendix, but included it specifically because it leveraged an already-trained diffusion model, with our method, to do guidance. The reason that we only used flow matching models in the experiments in the main paper is that we need to train our own unconditional denoising models, and found flow matching training to be more stable than diffusion.
>
> > you write that the probability of jumps in two or more dimensions simultaneously is zero. But how does it work in the numerical setting where the continuous time is discretized into finite time intervals, e.g. in the Euler integration? I guess here two dimensions can jump simultaneously. Does this cause problems?
>
> Thanks for this insightful question. When the sampling is discretized, e.g. as in Euler integration, it is the case that more than one dimension can jump within a discrete time interval, potentially resulting in discretization errors. In practice, one could make the time steps in Euler integration small enough so that only one dimension can transition at any point in time. Furthermore, one can also use exact sampling methods, e.g. Gillespie's Algorithm, which might require more sampling steps (lines 2025-1030, Gillespie [1], Campbell et al. [2]). In our empirical investigations, we have found that Euler integration worked reasonably well, similar to what Campbell et al. [3] observed, but there might exist situations where exact sampling methods are necessary.
>
> > can you comment more on the issue with diffusion models? (that you only use flow matching in the experiments) Is this a weakness of the model
>
> This was addressed in our earlier response “Indeed, our approach is general and …”.
>
>
> [1] Gillespie. “Exact stochastic simulation of coupled chemical reactions.” The Journal of Physical Chemistry (1977)
>
> [2] Campbell et al. “A Continuous Time Framework for Discrete Denoising Models.” NeurIPS (2022)
>
> [3] Campbell et al. "Generative flows on discrete state-spaces: Enabling multimodal flows with applications to protein co-design." ICML (2024).

---

> > ### Comment · Reviewer_MZ28 · 2024-11-19
> >
> > Thank you for the response. I maintain my score. I am still a bit worried about the time discretization. The time steps can of course be small, but I am not sure it is clear that the method will actually converge to the continuous time solution when you assume dimensions cannot jump simultaneously in the finite time intervals. If you can prove this, it would be great of course.

---

> > > ### Author Response · Authors · 2024-11-19
> > > **Response to comment**
> > >
> > > > Thank you for the response. I maintain my score. I am still a bit worried about the time discretization. The time steps can of course be small, but I am not sure it is clear that the method will actually converge to the continuous time solution when you assume dimensions cannot jump simultaneously in the finite time intervals. If you can prove this, it would be great of course.
> > >
> > > Thank you for your engagement with this question. We agree this is important to clarify and hope that our updated reply here can help.
> > >
> > > To the best of our knowledge, with Euler integration, even when taking smaller and smaller time steps there will always be some non-zero probability of having more than one jump in any fixed time interval. Our understanding is that errors from Euler integration not only affect our approach, but all unconditional continuous-time discrete flow-matching and diffusion models. These models only provide exact (unconditional) samples when exact numerical integration is used.
> > >
> > > However, Gillespie’s algorithm [1] is a well-established approach to exactly integrate CTMC dynamics without discretization error. Our method can be easily used without modification with the Gillespie integration method to produce exact samples. Gillespie’s Algorithm can simulate the CTMC exactly by alternating between sampling a holding time and sampling the next state (that only differs in one dimension from the previous state), where both the holding time and the state transition probabilities depend on the rates. We ended up using Euler integration for two reasons: (1) Euler is simpler to implement and (2) Euler allowed us to tradeoff speed for integration error by controlling the time step.
> > >
> > > [1] Gillespie. “Exact stochastic simulation of coupled chemical reactions.” The Journal of Physical Chemistry (1977)

---

> > > > ### Comment · Reviewer_MZ28 · 2024-11-19
> > > >
> > > > Great, thank you. The possible use of Gillespie's algorithm is to me a satisfactory answer.

---

### Official Review · Reviewer_wQnx · 2024-10-31

**Soundness:** 3
**Presentation:** 2
**Contribution:** 3
**Rating:** 6
**Confidence:** 3

**Summary:**

The authors propose a methodology to apply conditional guidance to generative models that are discrete and based on Continuous-time Markov Chains.
When assuming single state transitions in the Markov chain, Discrete Guidance is shown to be tractable since it does not necessitate to sum over all the chain states but only on the sparse set of reachable states.
Besides, a heuristic acceleration called Taylor-approximated guidance is propose to further limit the number of neural evaluations.
Three different examples are presented demonstrating the feasibility and efficiency of the approach and present comparison with competitive methods.

**Strengths:**

The paper introduces a principle mechanism for guidance in a discrete state flow-based generative model.

Experiments demonstrate the feasibility of the approach in three different settings (and small images Appendix F.1).

Reproducibility is ensured by providing source code.

Overall I find that the paper is too dense for the ICLR format and would benefit from a more in depth review. Yet the approach is certainly of interest for the community.

**Weaknesses:**

The term coined "discrete guidance" seems too general since there is already another competing approach called DiGress (Vignac et al. ICLR 2023) and also Dirichlet FM, sometimes with similar performance
(eg 423 "As predictor-free guidance for DirFM-CFG and DG-PFG behaved comparably (Appendix F.3.5)")
In that respect, the empirical results being relatively close to DiGress, the title of the paper is also a bit problematic  in my opinion.

The mathematical presentation in the main paper is a too quick summary in my opinion.
Equation (3) is only valid for tilde x different than x, and flow conservation is used for setting R gamma(x,x) (Equation 25). This should be included in the main text.
I read Appendix C and was a bit confused by the presentation.
At first I was surprised with key differences: In Equation (14) the numerator depends on both x tilde and x while in Equations (2) and (3) it only depends on the state x tilde. The simplification is discussed later in Section C.6, but I would have like to read this just below Equation (14).

216: The text suggests gamma is between 0 and 1, but it is used with very high values (2, 10, 100) later on. I would suggests to discuss the range of gamma right away.

In the experiments, the value of gamma used for each experiment is not reported clearly.

The presentation of the experiments is also very dense.
Somehow the conditioning mechanisms are not very clear.
One practical issue with classifier and/or classifier-free guidance is the need to eihter train a classifier on noisy data or to train a conditional generative model.
Here the effort needed for retraining is not clear, eg
419 For Discrete Guidance, we trained an unconditional flow-matching model, then used predictor guidance with either exact (DG-Exact) or Taylor-approximated guidance (DG-TAG).
How was the predictor trained ? Is there any example that is training-free ? (see question below)


Minor corrections / suggestions:
109 The text suggests that the gradient of the predictor is easily accessible of continuous DM but this is not the case and based on heuristics (eg using a classifier or assuming a Gaussian form for the distribution).
140 Why "However" ?
145 Say right away that the rate function is positive but can take negative value when x tilde equals x. Also precise the meaning of the Kroenecker delta symbol.
214: This footnote is already useful for p. 3
412: *an* conditional flow-matching model
897 Cl’ement Vignac (accent issue)

**Questions:**

"Originally, for DirFM-CG we used the same guidance strengths γ ∈ {1, 3, 6, 10, 20} reported by Stark et al. (2024)." Why use the same guidance ? Is the formalism similar ?

There exists training-free guidance strategies for solving imaging inverse problems using diffusion models, eg:

Diffusion Posterior Sampling for General Noisy Inverse Problems
Hyungjin Chung, Jeongsol Kim, Michael Thompson Mccann, Marc Louis Klasky, Jong Chul Ye
ICLR 2023

Pseudoinverse-Guided Diffusion Models for Inverse Problems
Jiaming Song, Arash Vahdat, Morteza Mardani, Jan Kautz
ICLR 2023

Are training-free guidance achievable for discrete diffusion/flow models?

---

> ### Author Response · Authors · 2024-11-16
> **Response part 1 out of 3**
>
> Thank you for your thoughtful comments and questions. We address them below.
>
> > The term coined "discrete guidance" seems too general since there is already another competing approach called DiGress (Vignac et al. ICLR 2023) and also Dirichlet FM, sometimes with similar performance (eg 423 "As predictor-free guidance for DirFM-CFG and DG-PFG behaved comparably (Appendix F.3.5)") In that respect, the empirical results being relatively close to DiGress, the title of the paper is also a bit problematic in my opinion.
>
> Thank you for the feedback. We certainly do not want to appear to be overclaiming our contribution at the expense of others. We coined our method “discrete guidance” because our approach is the first method that rigorously derives an exact, tractable and general method for guiding discrete diffusion and flow-matching models with a predictor. In contrast, DiGress is an approximate method for performing guidance that does not have any theoretical guarantees and is only applicable to discrete time discrete state-space diffusion models and can not be applied to flow models. Correspondingly, we show in our paper that our approach yields a significant improvement over DiGress in two out of three of the experiments (small-molecules and proteins).
> Dirichlet FM performs guidance on the continuous simplex using previously established techniques for guidance with continuous state-spaces. Thus, Dirichlet FM is a form of continuous-space guidance. Moreover, this approach has clear empirical drawbacks when performing classifier guidance, as we show in our DNA experiments. Thus, we believe we have appropriately situated our work to warrant the term “discrete guidance”. Having said that, if it is truly a sticking point, we are open to changing the title to something that you think is more appropriate.
>
> > The mathematical presentation in the main paper is a too quick summary in my opinion.
>
> Indeed, a more detailed exposition of the background and derivations in the main text would make it more self-contained; due to space constraints we had included such a detailed introduction in Appendix B. What we didn’t do well was to make connections between these sections and the main text more explicit, which we believe would help alleviate the issue, and will do so in our new version.
>
> > Equation (3) is only valid for tilde x different than x, and flow conservation is used for setting R gamma(x,x) (Equation 25). This should be included in the main text.
>
> Agreed, thank you for pointing it out. We will add this information in the revised version of the manuscript.
>
> > I read Appendix C and was a bit confused by the presentation. At first I was surprised with key differences: In Equation (14) the numerator depends on both x tilde and x while in Equations (2) and (3) it only depends on the state x tilde. The simplification is discussed later in Section C.6, but I would have like to read this just below Equation (14).
>
> Great suggestion. We will move section “Appendix C.6 Simplifying Predictor Guidance in CTMCs” right after section “Appendix C.3 Predictor Guidance in CTMCs” (thereby making it section C.4) for the revised version of the manuscript.
>
> 216: The text suggests gamma is between 0 and 1, but it is used with very high values (2, 10, 100) later on. I would suggests to discuss the range of gamma right away.
>
> Good point. Indeed, gamma can be any positive real number and we will change the sentence in lines 203-204 accordingly in the revised manuscript.
>
> > In the experiments, the value of gamma used for each experiment is not reported clearly.
>
> To clarify, we had indicated the value of gamma used for each experiment in the main text. For the small-molecule experiments, the value of gamma is described on lines 404-406. For the DNA experiments, the values of gammas are described on lines 439-440. For the protein examples, the values of gamma used are on line 478. Note that we explain in the caption of Figure 4 (lines 504-505) that the results in the figure are the maximum success values over the three reported gamma values. Please let us know if you still find it unclear.
>
>
> > The presentation of the experiments is also very dense. Somehow the conditioning mechanisms are not very clear.
>
> Would you mind clarifying how you think the experimental section could be improved? Perhaps you mean that this section could benefit from more detailed descriptions? We believe that more detailed descriptions of the experiments would help to make it more accessible. However, due to space constraints, we had to include the comprehensive details of each experiment in Appendix F.
>
> Regarding the conditioning mechanisms, perhaps you mean that it is unclear whether we are applying predictor or predicter-free guidance in each experiment? This was not very clear; we will modify the text to clearly state for each experiment whether predictor-guidance or predictor-free guidance was used.

---

> ### Author Response · Authors · 2024-11-16
> **Response part 2 out of 3**
>
> > One practical issue with classifier and/or classifier-free guidance is the need to either train a classifier on noisy data or to train a conditional generative model. Here the effort needed for retraining is not clear, eg 419 For Discrete Guidance, we trained an unconditional flow-matching model, then used predictor guidance with either exact (DG-Exact) or Taylor-approximated guidance (DG-TAG). How was the predictor trained? Is there any example that is training-free ? (see question below)
>
> Indeed, it was an oversight not to have mentioned training of the predictive model on noisy data more explicitly in the main text. We had discussed one of the standard procedures for doing so (Song et al. [1], Dhariwal et al. [2]) at the end of “Appendix C.6 Simplifying Predictor Guidance in CTMCs”. Improving upon these procedures is an interesting problem that warrants stand-alone investigation, such as by Klarner et al. [3]. We had also discussed the variations of this training procedure that we made to yield more useful predictive models: we found it helpful to train the noisy predictor on samples labeled by the clean predictor (lines 2593-2595). We detailed which procedure was used for training the predictors in each experiment in the corresponding sections in the Appendix (e.g. Appendix F.2.2 for molecules, Appendix F.3.2 for DNA, and Appendix F.4.3 for proteins). Unfortunately, we forgot to link to these sections or discuss the issue at all in the main text, which we will now remedy.
>
> > 109 The text suggests that the gradient of the predictor is easily accessible of continuous DM but this is not the case and based on heuristics (eg using a classifier or assuming a Gaussian form for the distribution).
>
> We agree that this characterization of guidance in continuous state-space is potentially misleading and will change the wording in the manuscript to reflect this.
>
> > 140 Why "However" ?
>
> Removed!
>
> > 145 Say right away that the rate function is positive but can take negative value when x tilde equals x. Also precise the meaning of the Kroenecker delta symbol.
>
> Good point – we will clarify this in the text.
>
> > 214: This footnote is already useful for p. 3
>
> We agree and will move the footnote to page 3.
>
> > 421: an conditional flow-matching model 897 Cl’ement Vignac (accent issue)
>
> Thank you! These typos will be corrected in the revised version of the manuscript.
>
> > "Originally, for DirFM-CG we used the same guidance strengths γ ∈ {1, 3, 6, 10, 20} reported by Stark et al. (2024)." Why use the same guidance strengths? Is the formalism similar?
>
> To clarify, DirFM-CG is a method proposed in Stark et al. [4]. Originally, we used the same guidance strengths as Stark et al. to reproduce the reported results in that work and to ensure a fair and consistent comparison with DirFM-CG, since we are evaluating on the same task using exactly the same metrics as Stark et al. Since we found DirFM-CG to perform quite poorly, to give that method the benefit of the doubt, we then extended the range of guidance strengths for DirFM-CG in an attempt to improve its performance beyond those reported by the authors in their paper (lines 440-442), so as to ensure as fair a comparison as possible with other methods. The results in Figure 3 were produced with these additional guidance strengths. The exact values for the additional guidance strengths we used were stated in Appendix F.3.3 (lines 2431-2432). We systematically expanded the range until DirFM-CG became numerically unstable and produced NaNs (lines 2433-2434). We hypothesize that the numerical issue might be related to the need of DirFM-CG to project the classifier score function onto the tangent plane of the simplex to perform guidance within the simplex.

---

> > ### Comment · Reviewer_wQnx · 2024-11-19
> >
> > I acknowledge that I have read the response by the authors and I thank them for the various clarifications.
> > To answer the direct question:
> > > Regarding the conditioning mechanisms, perhaps you mean that it is unclear whether we are applying predictor or predicter-free guidance in each experiment?
> > Yes.

---

> > > ### Author Response · Authors · 2024-11-19
> > > **Updates in manuscript**
> > >
> > > Thank you for carefully reading our responses and for engaging with the discussion. We have now uploaded a revised pdf incorporating your suggestions, with the changes detailed below. If there is anything else we could clarify or improve to raise your score for the manuscript, please don’t hesitate to let us know.
> > >
> > > > Equation (3) is only valid for tilde x different than x, and flow conservation is used for setting R gamma(x,x) (Equation 25). This should be included in the main text.
> > >
> > > We updated the manuscript in Section 4, paragraphs 5 (line numbers 207-208) and 6 (line numbers 212-213).
> > >
> > > > I read Appendix C and was a bit confused by the presentation. At first I was surprised with key differences: In Equation (14) the numerator depends on both x tilde and x while in Equations (2) and (3) it only depends on the state x tilde. The simplification is discussed later in Section C.6, but I would have like to read this just below Equation (14).
> > >
> > > In the Appendix, we moved "C.6. Simplifying predictor guidance in CTMCs" after "C.3. Predictor guidance in CTMCs" and split Section C.6. into two individual sections for more clarity.
> > >
> > > >216: The text suggests gamma is between 0 and 1, but it is used with very high values (2, 10, 100) later on. I would suggests to discuss the range of gamma right away.
> > >
> > > We clarified the range of gamma Section 4, paragraph 4 (line numbers 207-208) and paragraph 6 (line numbers 221-223).
> > >
> > > > The presentation of the experiments is also very dense. Somehow the conditioning mechanisms are not very clear.
> > >
> > > We have clarified each of the experimental sections to make sure the conditioning mechanism is clear. For the small molecule section we modified line numbers 368-369. For the protein section we modified line 461.
> > >
> > > > One practical issue with classifier and/or classifier-free guidance is the need to either train a classifier on noisy data or to train a conditional generative model... How was the predictor trained?
> > >
> > > We modified the main text in the first paragraph of Section 6 (line numbers 308-310) to make this more clear.
> > >
> > > > 140 Why "However" ?
> > >
> > > We removed this.
> > >
> > > > 145 Say right away that the rate function is positive but can take negative value when x tilde equals x. Also precise the meaning of the Kroenecker delta symbol.
> > >
> > > We modified the first paragraph of Section 3 (line numbers 145-157) to make the meaning of the Kronecker delta symbol precise and to explain when rates can be positive and negative.
> > >
> > > > 214: This footnote is already useful for p. 3
> > >
> > > This has been moved to page 3.
> > >
> > > > 421: an conditional flow-matching model 897 Cl’ement Vignac (accent issue)
> > >
> > > This is fixed!
> > >
> > > > Are training-free guidance achievable for discrete diffusion/flow models?
> > >
> > > We added a discussion at the end of Appendix E.2 about training-free guidance!

---

> ### Author Response · Authors · 2024-11-16
> **Response part 3 out of 3**
>
> > There exists training-free guidance strategies for solving imaging inverse problems using diffusion models, eg:
>
> >> Diffusion Posterior Sampling for General Noisy Inverse Problems Hyungjin Chung, Jeongsol Kim, Michael Thompson Mccann, Marc Louis Klasky, Jong Chul Ye ICLR 2023
>
> >> Pseudoinverse-Guided Diffusion Models for Inverse Problems Jiaming Song, Arash Vahdat, Morteza Mardani, Jan Kautz ICLR 2023
>
> > Are training-free guidance achievable for discrete diffusion/flow models?
>
> Thank you for pointing out the insightful connection between our work and prior work on training-free guidance in continuous state-space diffusion models. While we have not empirically explored any training-free method for predictor guidance, we do believe this is a potentially fruitful area for future research. We will include a more detailed discussion of how our proposed method relates to training-free methods e.g. Chung et al. [5] in the related work section in Appendix E, which we will summarize below.
>
> First, we note that the procedure to compute the guide-adjusted rate matrix in Equation (2) is the same regardless of how one obtains the estimate for $p(y | x, t)$ (to follow the notation in Chung et al., we now denote this distribution as $p(y | x_t)$). As noted in Equation (7) of Chung et al., the true distribution $p(y | x_t)$ is intractable, and the closer one’s approximation is to the true distribution, the closer the generated samples would be to being from the true posterior distribution $p(x | y)$. The contribution of our work lies in showing how to compute the guide-adjusted rate matrix assuming one can reliably estimate $p(y | x_t)$.
>
> Concretely, for continuous state-spaces, one approach for approximating $p(y | x, t)$ is by training a predictor that takes in noisy input, $p^{\phi}(y | x_t)$, where $x_t$ is the “noisy” version of the input $x$ at time step $t$ in the noising process. Alternatively, in Chung et al., one estimates $p(y | x_t)$  as $p(y | \hat{x_0})$, where $\hat{x_0} = E_{x_0 \sim p(x_0 |x_t)} [x_0]$ (Equation 11 in Chung et al). It’s important to note that both training a noisy predictor and the approach in Chung et al. are approximations to the true distribution $p(y | x_t)$. The training procedure of the predictor can be modified to better match the true $p(y | x_t)$, such as those we have discussed above, while the approach in Chung et al. has been shown to perform suboptimally without additional corrections, e.g. via sequential Monte Carlo (SMC) as suggested by Wu et al. [6]. It is not immediately obvious how to extend the training-free approaches in e.g. Chung et al. to discrete state-spaces. For example, one potential issue is that when the input space is discrete, $\hat{x_0} = E_{x_0 \sim p(x_0 |x_t)} [x_0]$  can lie on the probabilistic simplex for one-hot encoded states $x_0$, while $p(y | x_0)$ only takes discrete input (i.e., in the corners of the simplex), although there are potential ways to bypass this issue beyond the scope of our current work.
>
>
> We hope our additional explanations help clarify your questions and mitigate any concern you have. If you feel that your concerns have been sufficiently addressed, would you be willing to consider revising your score accordingly?
>
>
> [1] Song et al. “Score-Based Generative Modeling through Stochastic Differential Equations.” ICLR (2021)
>
> [2] Dhariwal et al. “Diffusion Models Beat GANs on Image Synthesis.” NeurIPS (2021)
>
> [3] Klarner et al. “Context-Guided Diffusion for Out-of-Distribution Molecular and Protein Design.” ICML (2024)
>
> [4] Stark et al. "Dirichlet Flow Matching with Applications to DNA Sequence Design." ICML (2024)
>
> [5] Chung et al. “Diffusion Posterior Sampling for General Noisy Inverse Problems.” ICLR (2023)
>
> [6] Wu et al. “Practical and Asymptotically Exact Conditional Sampling in Diffusion Models.” NeurIPS (2023)

---

### Official Review · Reviewer_ty55 · 2024-10-31

**Soundness:** 3
**Presentation:** 3
**Contribution:** 2
**Rating:** 6
**Confidence:** 3

**Summary:**

This paper presents a technique for guiding the generation of samples from discrete state-space diffusion models. The motivation behind this approach is the guidance mechanism employed in continuous state-space diffusion models. The authors derive the explicit form of both the predictor-guided and predictor-free-guided rate matrices for the discrete diffusion model. Subsequently, they propose a Taylor-approximated guidance as a computationally efficient solution. The effectiveness of the proposed method is demonstrated through its application to various applications, including the guided generation of small molecules, DNA sequences, and protein sequences.

**Strengths:**

- The paper proposes a general method for applying guidance on discrete state-space diffusion models, which constitutes a novel contribution to the field.
The authors provide a comprehensive derivation of the predictor-guided and predictor-free-guided rate matrices for the discrete diffusion model, accompanied by intuitive explanations of how they compare with the continuous guidance mechanism.
- The proposed method is demonstrated through its successful application to a diverse range of applications, including the guided generation of small molecules, DNA sequences, and protein sequences.

**Weaknesses:**

- The paper omits an introduction to the forward and backward processes of the discrete state-space diffusion models. Additionally, the rationale behind the training of the rate matrix $R_t(x,\tilde x)$ is not explicitly provided. I recommend that the authors commence with a more general introduction to the discrete state-space diffusion models, followed by a detailed explanation of the guidance mechanism in the continuous case, with a clear comparison between the two.
- Furthermore, it would be beneficial to include more algorithm details of the training of both the predictor-guided and predictor-free-guided rate matrices. This information should be provided in addition to the code available in the supplementary material, as it may be less accessible and understandable.

**Questions:**

- It is unclear how the predictor-free-guided rate matrix differs from the predictor-guided rate matrix in Equation (4), which is simply substituting the original expression of the predictor-guided rate matrix in Equation (2) into Equation (4). To the best of my understanding, the classifier-free guidance in the continuous case eliminates the classifier-like likelihood $p_t(y|x_t)$ from the score function, but it needs to be evident how the predictor-free-guidance in the discrete case accomplishes the same objective Could the authors provide further clarification on this matter?
- Could you elaborate on how the Taylor-approximated guidance operates in practice? Is it that instead of $p^\phi(y|x,t)$, we only store $\nabla_x \log p^\phi(y|x,t)$? What if there is no discernible order of the discrete states? How can we embed the discrete states into a continuous space for the Taylor approximation?

---

> ### Author Response · Authors · 2024-11-16
> **Response part 1 out of 3**
>
> Thank you for your thoughtful comments and questions. We address them below.
>
> > The paper omits an introduction to the forward and backward processes of the discrete state-space diffusion models. Additionally, the rationale behind the training of the rate matrix
>  is not explicitly provided. I recommend that the authors commence with a more general introduction to the discrete state-space diffusion models, followed by a detailed explanation of the guidance mechanism in the continuous case, with a clear comparison between the two.
>
> Indeed, we agree that including a more self-contained introduction about the forward and backward processes of discrete state-space diffusion models and the training of the rate matrix in the main text would make the paper more readable. Due to space constraints, we had included a detailed introduction to CTMCs in Appendix B, but omitted the important background regarding the forward and backward processes of the discrete state-space diffusion models and the rationale behind the training of the rate matrix, which we will now include.
>
> Regarding comparison between the guidance mechanism in the continuous and discrete cases, we had included a detailed discussion in Appendix E.2, which we will revise for improved clarity. Briefly, we started by reviewing the derivation from Sohl-Dickstein et al. [1] for constructing a general conditional reverse diffusion process in a discrete-time, continuous state-space diffusion framework and the approximations they introduced. We then showed how these approximations are made exact in the continuous-time, score-based diffusion framework of Song et al. [2]. We then turned to the discrete state-space settings and pointed out both analogy and distinction between Discrete Guidance and its continuous-space counterpart. It would be great if you had any further specific feedback that you believe would be helpful for the reader.
>
> > Furthermore, it would be beneficial to include more algorithm details of the training of both the predictor-guided and predictor-free-guided rate matrices. This information should be provided in addition to the code available in the supplementary material, as it may be less accessible and understandable.
>
> Thank you for this suggestion! We agree that a more clear explanation of the main algorithm for both training and guided sampling would improve readability of the paper. We will revise the manuscript to summarize the PyTorch code in Appendix D into a more succinct algorithmic description to improve the exposition.
>
> We also want to clarify that the training of the unconditional models (in the case of predictor guidance) and conditional models (in the case of predictor-free guidance) follow established procedures detailed in prior work (e.g. in Section 3.1.2 in Campbell et al. [3]). We will make this more clear in the main text on line 299 by adding “...in all three experiments, following the training procedure in Campell et al.”
>
> In the case of predictor guidance, one also needs to train a time-dependent noisy predictor that approximates $p(y | x_t, t)$. There are standard procedures to train these predictors (Song et al. [2], Dhariwal et al. [4]) and we explain one of these at the end of “Appendix C.6 Simplifying Predictor Guidance in CTMCs”, but forgot to point to this from the main text, which we will add. We have found that there are ways to modify those procedures to improve guided sampling and have discussed the procedures we used in the corresponding experimental section in the Appendix (e.g. Appendix F.2.2 for molecules, Appendix F.3.2 for DNA, and Appendix F.4.3 for proteins). We also included a more detailed response regarding training of the noisy predictors in our response to a question raised by Reviewer wQnx.

---

> ### Author Response · Authors · 2024-11-16
> **Response part 2 out of 3**
>
> > It is unclear how the predictor-free-guided rate matrix differs from the predictor-guided rate matrix in Equation (4), which is simply substituting the original expression of the predictor-guided rate matrix in Equation (2) into Equation (4). To the best of my understanding, the classifier-free guidance in the continuous case eliminates the classifier-like likelihood
>  from the score function, but it needs to be evident how the predictor-free-guidance in the discrete case accomplishes the same objective Could the authors provide further clarification on this matter?
>
> Apologies for the confusing way this was written. Equation (4) does not actually depend on any predictor distribution, $p(y | x, t)$, as does Equation (2) and (3), while our phrasing made it sound like perhaps it did—we will clarify the wording. What Equation (4) does require, is the conditional rate matrix $R_t(x, \tilde{x} | y)$, obtained by directly training a conditioned denoising model, without any reliance on a predictor model. This result is analogous to how predictor-free guidance is achieved in continuous-space diffusion models, namely, by blending the score of a conditional model trained with a conditional score (no predictor), and an unconditional model (Ho et al. [5]).
>
> > Could you elaborate on how the Taylor-approximated guidance operates in practice? Is it that instead of $p^{\phi}(y | x, t)$, we only store $\nabla_x \log p^{\phi}(y | x, t)$?
>
> Certainly. Suppose we have a trained time-dependent predictor $p^{\phi}(y | x, t)$, for exact predictor guidance, for every sampling step, one needs to evaluate $p^{\phi}(y | x, t)$ on all $D \times (S - 1)$ inputs $\tilde{x}$ that has a hamming-distance of one to the current state $x$, in addition to evaluating the likelihood on $x$, requiring $D \times (S - 1) + 1$ forward passes of the predictor model. In TAG, for every sampling step, one only evaluates $\nabla_x \log p^{\phi}(y | x, t)$ on the current state $x$, which requires one forward and one backward pass of the predictor model. These details were also reported in Listing 2 in Appendix D.1.
>
> > How can we embed the discrete states into a continuous space for the Taylor approximation?
>
> For non-ordinal D-dimensional discrete states, where each dimension can take any of $S$ categories, the one-hot encoded representation of a state is a binary vector in $\mathbb{R}^{D\times S}$. The realization in Grathwohl et al. [6] was that since embedding vectors are generated by multiplying a one-hot-encoded vector with a learnable “embedding matrix” (i.e., the first layer of the neural network), that we can evaluate the neural network on inputs that take on continuous values rather than just $0$ and $1$ in each dimension. This is admittedly a nuanced point, so please let us know if our explanation remains unclear and we will do our best to further clarify.

---

> ### Author Response · Authors · 2024-11-16
> **Response part 3 out of 3**
>
> > What if there is no discernible order of the discrete states?
>
> Indeed, the Taylor approximation that we propose works for both ordinal and non-ordinal discrete data. The surprising finding that this gradient approximation typically provides an accurate approximation in the non-ordinal setting was previously investigated empirically by Grathwohl et al. [6]. Our intuition is that neural network models tend to have an inductive bias that learns smooth functions and therefore have gradients at discrete points that tend to point roughly along a linear interpolation between adjacent discrete values. While it is true that the gradient approximation is likely more accurate when $x$ is ordinal than one-hot, it has been empirically shown to be accurate for purely categorical one-hot-encoded $x$ for the smoothness reasons described above [6-9]. In addition to Grathwohl et al. [6] such Taylor-series approximations have been applied to discrete one-hot variables in much additional work, including applications to text [7], proteins [8], and graphs [9].
>
> Nonetheless, despite the empirical evidence that this approximation works well in practice across a large number of neural network architectures, there are no theoretical guarantees that this first-order Taylor-series approximation will always be accurate. This highlights the particular importance of the development of an exact method in our paper, as opposed to DiGress. In principle, one could also explore methods intended to yield smooth, interpolating neural networks, such as Mixup [10], which may improve the accuracy of the Taylor-series approximation. We and others have so far not found this to be needed.
>
> It seems as though there were no major concerns regarding the soundness and novelty of this work, and we appreciate your recognition that this work constitutes a novel contribution to the field. We hope our additional explanations help clarify your questions and mitigate any concern you may have had. In light of this, would you be willing to consider revising your rating for the manuscript?
>
> [1] Sohl-Dickstein et al. “Deep Unsupervised Learning using Nonequilibrium Thermodynamics.” ICML (2015)
>
> [2] Song et al. “Score-Based Generative Modeling through Stochastic Differential Equations.” ICLR (2021)
>
> [3] Campbell et al. "Generative flows on discrete state-spaces: Enabling multimodal flows with applications to protein co-design." ICML (2024).
>
> [4] Dhariwal et al. “Diffusion Models Beat GANs on Image Synthesis.” NeurIPS (2021)
>
> [5] Ho et al. “Classifier-free Diffusion Guidance.” NeurIPS Workshop (2022)
>
> [6] Grathwohl et al. "Oops I Took a Gradient: Scalable Sampling for Discrete Distributions." ICML (2021)
>
> [7] Zhang et al. "A langevin-like sampler for discrete distributions." ICML (2022)
>
> [8] Kirjner et al. "Improving protein optimization with smoothed fitness landscapes." ICLR (2023)
>
> [9] Vignac et al. "DiGress: Discrete Denoising diffusion for graph generation." ICLR (2023)
>
> [10] Zhang et al. "mixup: Beyond Empirical Risk Minimization." ICLR (2018)

---

> > ### Author Response · Authors · 2024-11-19
> > **Updates in manuscript**
> >
> > We have now uploaded a revised pdf incorporating your suggestions, with the changes detailed below. If you have other suggestions that you would like to see incorporated, please don’t hesitate to let us know.
> >
> > > The paper omits an introduction to the forward and backward processes of the discrete state-space diffusion models. Additionally, the rationale behind the training of the rate matrix
> >  is not explicitly provided. I recommend that the authors commence with a more general introduction to the discrete state-space diffusion models, followed by a detailed explanation of the guidance mechanism in the continuous case, with a clear comparison between the two.
> >
> > We have now updated Appendix B (lines 1017-1029) to include a more detailed description of the forward and backward processes of discrete state-space diffusion and flow models, as well as the training of the rate matrix. It would be great if you have any further specific feedback that you believe would be helpful for the reader, and we would be happy to try to incorporate them.
> >
> > > Furthermore, it would be beneficial to include more algorithm details of the training of both the predictor-guided and predictor-free-guided rate matrices. This information should be provided in addition to the code available in the supplementary material, as it may be less accessible and understandable.
> >
> > We have now updated Appendix D to include algorithm descriptions in addition to the PyTorch code (Algorithm 1-3). We also added a line to clarify that the training of the unconditional models (in the case of predictor guidance) and conditional models (in the case of predictor-free guidance) follow established procedures detailed in prior work (lines 300-301). We have also updated lines 308-310 to point to the sections in the Appendix where we discussed the training of the predictors used in predictor guidance (Appendix C.5 for an overview, and Appendix F.2-4 for task-specific training procedures).
> >
> > > It is unclear how the predictor-free-guided rate matrix differs from the predictor-guided rate matrix in Equation (4), which is simply substituting the original expression of the predictor-guided rate matrix in Equation (2) into Equation (4). To the best of my understanding, the classifier-free guidance in the continuous case eliminates the classifier-like likelihood
> >  from the score function, but it needs to be evident how the predictor-free-guidance in the discrete case accomplishes the same objective Could the authors provide further clarification on this matter?
> >
> > We have now clarified this in Section 4 in lines 217-219.

---

### Official Review · Reviewer_t8PN · 2024-11-02

**Soundness:** 3
**Presentation:** 3
**Contribution:** 2
**Rating:** 6
**Confidence:** 3

**Summary:**

The author introduces a guidance framework for discrete generation models. This approach relies on utilizing continuous-time Markov processes over discrete state Spaces, which frees up the computationally processability of sampling from the desired lead distribution. More importantly, the authors demonstrate the effects of the proposed discrete guidance framework on a range of applications, including guided generation of small molecules, DNA sequences, and protein sequences.

**Strengths:**

The authors of this paper propose a guidance framework for generating models (diffusion models, flow matching models) on discrete state Spaces. This guidance strategy is applicable to a wide range of generation models of discrete state Spaces achieved through CTMC, and has been applied to the generation of guidance conditions in many fields such as small molecules, DNA sequences, and protein sequences. In addition, this paper has a reasonable logical structure and clear expression.

**Weaknesses:**

While the authors have demonstrated that this guiding framework is empirically effective, more precise theoretical guarantees, such as error analysis, are lacking, and further research into potential tradeoffs between efficiency and accuracy will be of interest to the community. In addition, the article is more inclined to show the effect of guidance and ignore the quality of generation, so it is difficult for readers to judge the practicality of this guidance framework. Finally, judging from the process of theoretical derivation, the technical route of this work seems to be weak in innovation.

**Questions:**

The authors mention that the proposed framework is suitable for a wide range of generation models implemented through  the CTMC, but this guidance framework is only applied to diffusion models and flow matching models. Can you elaborate on how to apply the framework to other generation models?

---

> ### Author Response · Authors · 2024-11-16
> **Response part 1 out of 3**
>
> Thank you for your thoughtful comments and questions. We address them below.
>
> > While the authors have demonstrated that this guiding framework is empirically effective, more precise theoretical guarantees, such 	as error analysis, are lacking, and further research into potential tradeoffs between efficiency and accuracy will be of interest to the community.
>
> We agree that theoretical guarantees are important. Indeed, a major contribution of our paper was to derive for the first time an exact, tractable method for applying guidance to diffusion and flow matching models on discrete state spaces. This is in contrast to other works that only provide heuristic / approximate approaches for guidance [1]. In particular, we prove that our guidance equations are guaranteed to sample from the correct, desired conditional distribution (Appendix C).
>
> The theoretical validity of our approach was mentioned by Reviewer ty55, who said "The authors provide a comprehensive derivation of the predictor-guided and predictor-free-guided rate matrices for the discrete diffusion model, accompanied by intuitive explanations of how they compare with the continuous guidance mechanism." Similarly, Reviewer wQnx mentioned that “the paper introduces a principled mechanism for guidance in a discrete state flow-based generative model.”
>
> Regarding error analysis, would you mind clarifying what type of analysis you would like to see? We note that, while our derivation is exact, in practice, error can accumulate due to (1) numerical integration of the CTMC and (2) the fact that the trained time-dependent predictors may not match the true distribution $p(y | x_t, t)$. We will revise the manuscript to make this point more explicit. These same errors arise in diffusion and flow-matching models on real-valued state-spaces, where this issue has, as far as we know, not been well-characterized but where these models have proven tremendously successful in practice. Similarly, in our experiments, we see that despite these potential issues, our conditional sampling is useful in practice. If you have a suggestion for a specific type of error analysis, we’d be happy to try our best to include it.

---

> ### Author Response · Authors · 2024-11-16
> **Response part 2 out of 3**
>
> > In addition, the article is more inclined to show the effect of guidance and ignore the quality of generation, so it is difficult for readers to judge the practicality of this guidance framework.
>
> Our apologies that the quality of generation was not clearly communicated. Indeed, our intention with all of the empirical experiments was to evaluate the quality of the samples generated by the guidance procedure. We next clarify this point, and will update the manuscript accordingly.
>
> Samples obtained from a correct and effective guidance method should match the specified desired conditional distribution that is being targeted. The better a guidance method, the better the match will be. As the target distribution is unknown, directly quantifying the match is difficult, and there is no single established method for quantifying the degree of match, or more generally evaluating conditional generative models. We assessed the quality of the generated samples along two axes: (1) how well the samples satisfy the property specification being conditioning on and (2) how diverse the samples are to ensure that our methods are not simply collapsing onto a small set of samples that satisfy the desired property. We next explain in more detail.
>
> In our small-molecule experiments, we evaluated the quality of our generated samples by checking to what extent the samples contained the desired property, as measured by mean absolute error from the specified desired property value. In the appendix we also evaluated  diversity, measured with Tanimoto similarity. In addition, we included visual examples of the generated molecules for a large range of property values (Figure 8 and 9).
>
> In our DNA experiments, we follow the sampling quality metrics proposed in the paper on which those experiments are anchored, that of Stark et al [2]. Namely, Stark et al. used the probability of the desired target class under the oracle classifier model to check for samples having the desired property value. They also used the metric of Frechet Biological Distance (FBD), which measures the similarity between the generated samples and the training data for a given cell type, using a hidden layer representation of the oracle classifier, as a proxy for measuring both the quality and diversity of the generated samples. We used these same evaluation metrics.
>
> In our protein experiments, we checked for samples having the two desired property values: (1)  the sequence folds into a desired structure, and (2) the sequence is at least as stable as wild-type, $\Delta \Delta G$ ≥ 0. The first criteria was quantified by folding the sequences with AlphaFold2 and then measuring the root mean squared deviation (RMSD) to the desired conditioning structure. This is a commonly used metric to assess the quality of samples generated from inverse folding models. The stability criterion was measured by evaluating whether the generated sequences have a predicted stability greater than the wild-type sequence with the trained stability regression model. We also measured diversity using the average pairwise hamming distance between the generated sequences (results in ​​Appendix F.4.6).
>
> Please let us know if there are more specific metrics you would like us to include?

---

> ### Author Response · Authors · 2024-11-16
> **Response part 3 out of 3**
>
> > Finally, judging from the process of theoretical derivation, the technical route of this work seems to be weak in innovation.
>
> Prior to this work, there was no theoretically rigorous method for applying guidance to discrete state-space diffusion and flow-matching models, despite the large desire to have such a method, suggesting that the community at large did not know how to derive our results. Our primary innovation was showing (in Appendix C) that in the continuous time-limit of the discrete diffusion or flow-matching process, guidance becomes tractable. Intuitively, this is due to the fact that in the continuous-time limit, there is zero probability that two dimensions change at the same instantaneous point in time [3], making the normalizing constant in Bayes rule tractable – an important insight that the community had not yet had.
>
>
> > The authors mention that the proposed framework is suitable for a wide range of generation models implemented through the CTMC, but this guidance framework is only applied to diffusion models and flow matching models. Can you elaborate on how to apply the framework to other generation models?
>
> This is a good point. While in practice we have only instantiated our method for diffusion and flow matching, conceptually, our framework works only at the level of rate matrices of a CTMC for which transitions are independent across dimensions. It does not matter (for our framework) how these rate matrices have been determined (e.g., within a continuous-time discrete space diffusion or flow-model framework, or some other approach not yet invented). This means that our framework can guide rates determined by any generative method based on CTMCs with independent transitions across dimensions. Nevertheless, we are not yet aware of any other models, and for clarity, will edit the manuscript to reflect this so that the reader is not puzzled about what these other methods are.
>
>
> We hope our additional explanations help clarify your questions and mitigate any concern you have. If you feel that your concerns have been sufficiently addressed, would you be willing to consider revising your score accordingly?
>
> [1] Vignac et al. "DiGress: Discrete Denoising diffusion for graph generation." ICLR (2023)
>
> [2] Stark et al. "Dirichlet Flow Matching with Applications to DNA Sequence Design." ICML (2024)
>
> [3] Campbell et al. “A Continuous Time Framework for Discrete Denoising Models.” NeurIPS (2022)

---

> > ### Author Response · Authors · 2024-11-19
> > **Updates in manuscript**
> >
> > We have now uploaded a revised pdf incorporating your suggestions. The changes that we have made are as follows:
> >
> > > The authors mention that the proposed framework is suitable for a wide range of generation models implemented through the CTMC, but this guidance framework is only applied to diffusion models and flow matching models. Can you elaborate on how to apply the framework to other generation models?
> >
> > We have updated the last paragraph of Section 4 (line 225) to point to the section in the Appendix where we discuss this (Appendix C.9). We have also clarified the text in Appendix C.9 (line numbers 1378-1385).
> >
> > If you have other suggestions that you would like to see incorporated, please don’t hesitate to let us know.

---

### Author Response · Authors · 2024-11-16
**Global comment to reviewers**

We thank all the reviewers for their thoughtful comments. We have addressed each reviewer’s comments in the individual responses below. We will soon upload a revised pdf reflecting the changes that we had mentioned in our responses, but we want to post the responses here first to initiate the discussion with the reviewers.

---

> ### Author Response · Authors · 2024-11-19
> **Updated manuscript**
>
> We have now updated the pdf document to incorporate the revisions that we have mentioned in our responses. The changes are shown in red in the pdf. We have also added individual responses to each reviewer specifying the changes that are specific to each reviewer’s comment.

---

### Meta-Review · Area_Chair_ywvz · 2024-12-17

**Metareview:**

This paper introduces a principled method for applying guidance to discrete state-space diffusion and flow models using continuous-time Markov processes. The approach is theoretically grounded, computationally tractable, and broadly applicable to both discrete diffusion and flow-matching models. The method's effectiveness is demonstrated through experiments on small-molecule generation, DNA sequence generation, and protein sequence generation.

The proposed method is the first work to rigorously derive an exact, tractable, and general method for guiding discrete diffusion and flow-matching models with a predictor. In contrast, most competing approaches rely on approximate methods for guidance, which lack theoretical guarantees and/or are limited to discrete-time, discrete-state-space diffusion models. Furthermore, most of the competing approaches cannot be directly applied to flow models. The proposed approach overcomes these limitations, offering a mathematically sound and practical solution for guiding generative processes in discrete state-spaces.

The panel of reviewers recommend acceptance of this paper: the work addresses an important gap in the field by extending guidance to discrete state-spaces; this type of models are, for example, essential for applications in natural sciences such as molecular and biological sequence generation.

**Additional Comments On Reviewer Discussion:**

During the rebuttal phase, the authors provided a number of clarifications. They clarified the relationship between the proposed method and competing approaches such as DiGress and Dirichlet Flow, highlighting its advantages and differences. Additionally, they provided details on how to discretize the continuous-time discrete-state Markov process (Eg. Gillespie also, etc..), also improved the paper's presentation by fixing bad notation choices and updating Appendix D to include algorithm descriptions for better reproducibility. Furthermore, they elaborated on how the Taylor-approximated guidance operates in practice.

---

### Decision · Program_Chairs · 2025-01-22

Accept (Poster)